# A generalized neural tangent kernel for surrogate gradient learning

**Luke Eilers**[*]
Department of Physiology, University of Bern, Switzerland
Institute for Applied Mathematics, University of Bonn, Germany
`luke.eilers@unibe.ch`

**Raoul-Martin Memmesheimer**
Institute of Genetics, University of Bonn, Germany
`rm.memmesheimer@uni-bonn.de`

**Sven Goedeke**
Bernstein Center Freiburg, University of Freiburg, Germany
Institute of Genetics, University of Bonn, Germany
`sven.goedeke@bcf.uni-freiburg.de`

## Abstract

State-of-the-art neural network training methods depend on the gradient of the network function. Therefore, they cannot be applied to networks whose activation functions do not have useful derivatives, such as binary and discrete-time spiking neural networks. To overcome this problem, the activation function's derivative is commonly substituted with a surrogate derivative, giving rise to surrogate gradient learning (SGL). This method works well in practice but lacks theoretical foundation.

The neural tangent kernel (NTK) has proven successful in the analysis of gradient descent. Here, we provide a generalization of the NTK, which we call the surrogate gradient NTK, that enables the analysis of SGL. First, we study a naive extension of the NTK to activation functions with jumps, demonstrating that gradient descent for such activation functions is also ill-posed in the infinite-width limit. To address this problem, we generalize the NTK to gradient descent with surrogate derivatives, i.e., SGL. We carefully define this generalization and expand the existing key theorems on the NTK with mathematical rigor. Further, we illustrate our findings with numerical experiments. Finally, we numerically compare SGL in networks with sign activation function and finite width to kernel regression with the surrogate gradient NTK; the results confirm that the surrogate gradient NTK provides a good characterization of SGL.

## 1 Introduction

Artificial neural networks (ANNs) originate from the biologically inspired perceptron [Rosenblatt, 1958]. While the perceptron has a binary output that is faithful to the all-or-none behavior of spiking neurons in the nervous system, most activation functions considered nowadays for ANNs are smooth (like the logistic function) or at least semi-differentiable (like the ReLU function). Differentiable network functions enable the learning of network weights with methods that leverage the gradient of the network function with respect to the network weights like backpropagation [Rumelhart et al.,

---

[*]Corresponding author

38th Conference on Neural Information Processing Systems (NeurIPS 2024).

1986]. These methods are very successful [LeCun et al., 2015], but cannot be used without well-defined gradients.

This is a problem when considering more biologically plausible ANNs, which are typically used in computational neuroscience to understand how the networks of spiking neurons in our nervous system work. These include spiking neural networks (SNNs). Motivated by the energy efficiency of our brain, SNNs and similar networks, such as binary neural networks (BNNs), are considered in the context of neuromorphic computing [Merolla et al., 2014]. Both discrete-time SNNs and BNNs have in common that their activation functions do not have useful derivatives, which renders standard gradient-descent training impossible [Neftci et al., 2019, Taherkhani et al., 2020, Tavanaei et al., 2019, Roy et al., 2019, Pfeiffer and Pfeil, 2018]. For the scope of this paper, these activation functions can be thought of as step-like functions like the sign function, in which cases the derivative vanishes almost everywhere and is thus no longer informative about the shape of the activation function.

Surrogate gradient learning resolves this issue by providing the missing information about the activation function in the form of a surrogate derivative [Hinton, 2012, Bengio et al., 2013, Zenke and Ganguli, 2018]. As a result, the gradient-based methods for classical ANNs can be leveraged with great success [Zenke and Vogels, 2021]. However, a theoretical basis underpinning the intuition is missing and it is often unclear which surrogate derivative should be chosen for a particular network. For a review focusing on surrogate gradient learning methods, which we are most interested in, see Neftci et al. [2019]. For a comprehensive review of other learning methods for SNNs, we refer to Taherkhani et al. [2020], Tavanaei et al. [2019], and Eshraghian et al. [2021].

The neural tangent kernel (NTK) introduced by Jacot et al. [2018] allows to formulate gradient descent as a kernel method. Just as ANNs with randomly initialized weights converge under certain conditions to Gaussian processes (GPs) in the infinite-width limit [Matthews et al., 2018, Lee et al., 2018], the NTK converges at initialization and during training to a deterministic kernel in the same limit. Moreover, the NTK then describes how the network function changes under gradient descent in the infinite-width regime. This led to both a better theoretical understanding of gradient descent and practical applications to neural network training; see Section 1.2 for more details.

## 1.1 Contribution

We study the NTK for networks with sign function as activation function. As the NTK theory is not directly applicable due to an ill-defined derivative, we consider the NTK for a sequence of activation functions that approximates the sign function and then derive a principled way of generalizing the NTK to gain more theoretical insight into surrogate gradient learning. Our contributions are as follows:

- We provide a clear definition of the infinite-width limit in Section 2.2, capturing the different notions used in the literature due to the different choices of rates at which the layer widths increase. This definition is used consistently in all mathematical statements and the respective proofs.

- In Section 2.3, we demonstrate that the direct extension of the NTK for the sign activation function using an approximating sequence is not well-defined due to the divergence of the kernel's diagonal. This illustrates, from the NTK perspective, that gradient descent is ill-defined for activation functions with jumps and how this problem will be mitigated by surrogate gradient learning. Moreover, we connect this divergence to results by Radhakrishnan et al. [2023] in Theorem 2.3 and show that the direct extension of the NTK can be seen as a well-defined kernel for classification.

- We define a generalized version of the NTK in Section 2.4 using so-called quasi-Jacobian matrices and prove the convergence to a deterministic, in general asymmetric, kernel in the infinite width limit at initialization in Theorem 2.4. Using the generalized NTK, we formulate surrogate gradient learning in terms of the generalized NTK for networks with differentiable activation functions. This novel NTK is named the SG-NTK and we prove its convergence to a deterministic kernel during training in Theorem 2.5.

- For both the diverging direct extension of the NTK and the SG-NTK with sign activation function and arbitrary surrogate derivative, we derive exact analytical expressions in sections D.1, D.2 and E.2. In particular, we identify the terms that emerge from SGL and that prevent divergence.

- In Section 3, we illustrate our findings, in particular Theorem 2.4 and Theorem 2.5, using simulations. Numerical experiments show that the distribution of networks trained with SGL shows agreement with the distribution given by the SG-NTK.

Mathematically precise versions of all statements can be found in the appendix, which is self-contained to ensure rigor and a consistent notation throughout all theorems and proofs.

## 1.2 Related work

**The neural tangent kernel.** The convergence of randomly initialized ANNs in the infinite-width limit to Gaussian processes under appropriate scaling of the weights has first been described by Neal [1996] for a single hidden layer and has been extended to multiple hidden layers and other network architectures like CNNs [Matthews et al., 2018, Garriga-Alonso et al., 2018, Yang, 2019b, Lee et al., 2018]. The NTK was introduced by Jacot et al. [2018] with first results on its convergence at network initialization and during training. Theoretical results on the implication of the convergence of NTKs on the behaviour of trained wide ANNs were given by Lee et al. [2019], Arora et al. [2019], Allen-Zhu et al. [2019]. In Section C.2, we review central theorems on the NTK to enable a clear comparison to our theoretical results.

The NTK has been generalized to all kinds of ANN standard architectures Yang [2020] such as CNNs Arora et al. [2019] and attention layers Hron et al. [2020]. In particular, it has been generalized to RNNs Alemohammad et al. [2020], which are particularly interesting in light of SNNs due to their shared temporal dimension. Bordelon and Pehlevan [2022] derive a generalization of the NTK called effective NTK for different learning rules using an approach similar to ours. Note that all of these extensions require well-defined gradients.

The ability of overparameterized neural networks to converge during training and to generalize can be explained using the NTK [Allen-Zhu et al., 2019, Bordelon et al., 2020, Bietti and Mairal, 2019, Du et al., 2019]. The NTK has also been used in more applied areas such as neural architecture search [Chen et al., 2021] and dataset distillation [Nguyen et al., 2021].

**Surrogate gradient learning.** In the context of computational neuroscience, the idea of replacing the derivative of the output of a spiking neuron with a surrogate derivative was introduced by Bohte [2011]. To deal with the temporal component in SNNs or more generally RNNs, the resulting gradient is usually combined with backpropagation through time (BPTT). Prominent examples of surrogate gradient approaches include SuperSpike by Zenke and Ganguli [2018] and a number of works with different surrogate derivatives [Wu et al., 2018, 2019, Shrestha and Orchard, 2018, Bellec et al., 2018, Esser et al., 2016, Woźniak et al., 2020]. In the general ANN literature, the method is better known as straight-through estimation (STE) and was introduced by Hinton [2012] and by Bengio et al. [2013] in more detail. It was successfully applied by Hubara et al. [2016] and Cai et al. [2017].

Surrogate gradient learning or STE is only heuristically motivated and it is hence desirable to derive a theoretical basis. The influence of different surrogate derivatives on the method has been analyzed through systematic numerical simulations Zenke and Vogels [2021], revealing that the particular shape has a minor impact compared to the scale. In a more theoretical work, Yin et al. [2019] analyzed the convergence of STE for a Heaviside activation function with three different surrogate derivatives and found that the descent directions of the respective surrogate gradients reduce the population loss when the clipped ReLU function is used as surrogate derivative. Gygax and Zenke [2024] examine how SGL is connected to smoothed probabilistic models [Bengio et al., 2013, Neftci et al., 2019, Jang et al., 2019] and to stochastic automatic differentiation [Arya et al., 2022]. In particular, they consider SGL for differentiable activation functions, as we do in our derivation of the SG-NTK.

## 2 Theoretical results

### 2.1 Notation and NTK parametrization

We consider multilayer perceptrons with so-called neural tangent kernel parametrization. For a network with depth $L$, layer width $n_l$ for $0 \leq l \leq L$, activation function $\sigma$, weight matrices $W^{(l)} \in \mathbb{R}^{n_l \times n_{l-1}}$, and biases $b^{(l)} \in \mathbb{R}^{n_l}$, the preactivations with NTK parametrization are given by

$$h^{(1)}(x) = \frac{\sigma_w}{\sqrt{n_0}} W^{(1)} x + \sigma_b \, b^{(1)},$$

$$h^{(l+1)}(x) = \frac{\sigma_w}{\sqrt{n_l}} W^{(l+1)} \sigma\left(h^{(l)}(x)\right) + \sigma_b \, b^{(l+1)} \quad \text{for } l = 1, \ldots, L-1,$$

where $\sigma_w > 0$, $\sigma_b \geq 0$, and we initialize $W_{ij}^{(l)}, b_i^{(l)} \overset{\text{iid}}{\sim} \mathcal{N}(0,1)$ for all $i,j$. The NTK parametrization differs from the standard parametrization by a rescaling factor of $1/\sqrt{n_l}$ in layer $l+1$. The network function is then given by $f(\,\cdot\,) = h^{(L)}(\,\cdot\,) \colon \mathbb{R}^{n_0} \to \mathbb{R}^{n_L}$ and notably the last layer is a preactivation layer. We denote the total number of weights by $P$. More details can be found in Section C.1.

**Notation (see Remark C.1 and C.2)** By default, we will interpret any vector as a column vector, i.e., we identify $\mathbb{R}^n$ with $\mathbb{R}^{n \times 1}$. This is the case even when writing $x = (x_1, \ldots, x_n) \in \mathbb{R}^n$ for handier notation. Row vectors will be indicated within calculations using the transpose operator, $x^\mathsf{T}$. For a function $f \colon \mathbb{R}^{n_1} \to \mathbb{R}^{n_2}$ and $\mathcal{X} = (x_1, \ldots, x_d) \in \mathbb{R}^{d \cdot n_1}$, we define the vector $f(\mathcal{X}) := (f(x_1), \cdots, f(x_d)) \in \mathbb{R}^{d \cdot n_2}$. For $n_2 = 1$, we denote the gradient of $f$ by $\nabla f(x) \in \mathbb{R}^{n_1}$ for all $x \in \mathbb{R}^{n_1}$. If $n_2 > 1$, we denote the Jacobian matrix of $f$ by $Jf \colon \mathbb{R}^{n_1} \to \mathbb{R}^{n_2 \times n_1}$. A subscript of the form $J_\theta f(x)$ denotes Jacobian matrices with respect to a subset of variables. We write $f(n) \asymp g(n)$ to denote that $f(n)$ and $g(n)$ are asymptotically proportional, i.e., $f(n) \sim Kg(n)$ for some constant $K \neq 0$.

## 2.2 The infinite-width limit

We will consider neural networks in the limit of infinitely many hidden layer neurons, i.e., $n_l \to \infty$ for all $1 \leq l \leq L-1$. We will see later, when paraphrasing existing results from the literature, that different ways of taking the number of hidden neurons to infinity can be found. To formalize these notions, we use the definition of a width function from Matthews et al. [2018] with slight modifications:

**Definition 2.1** (Width function). *For every layer $l = 0, \ldots, L$ and any $m \in \mathbb{N}$, the number of neurons in that layer is given by $r_l(m)$, and we call $r_l \colon \mathbb{N} \to \mathbb{N}$ the width function of layer $l$. We say that a width function $r_l$ is strictly increasing if $r_l(m) < r_l(m+1)$ for all $m \geq 1$. We set*

$$\mathcal{R}_L := \left\{ (r_l)_{l=1}^{L-1} \mid r_l \text{ is a strictly increasing width function for all } 0 < l < L \right\},$$

*the set of collections of strictly increasing width functions for network depth $L$.*

Every element of $\mathcal{R}_L$ provides a way to take the widths of the hidden layers to infinity by setting $n_l = r_l(m)$ for any $1 \leq l < L$ and considering $m \to \infty$. The notions of infinite-width limits used in the literature will now correspond to classes $R \subseteq \mathcal{R}_L$ for which the respective limiting statements hold. This is captured in the following definition.

**Definition 2.2** (Types of infinite-width limits). *Consider a statement $\mathcal{S}$ of the form "Let an ANN have depth $L$ and network layer widths defined by $n_0$, $n_L$, and $n_l := r_l(m)$ for $1 \leq l < L$ and some $(r_l)_{l=1}^{L-1} \in \mathcal{R}_L$. Then, for fixed $n_0$ and any $n_L$, the statement $\mathcal{P}$ holds as $m \to \infty$." We also write the statement $\mathcal{S}$ as $\mathcal{S}(r)$.*

(i) *We say that such a statement $\mathcal{S}$ holds strongly, if $\mathcal{S}(r)$ holds for any $r \in \mathcal{R}_L$. This can be interpreted as requiring that the statement holds as $\min_{1 \leq l < L}(n_l) \to \infty$. We will also write "$\mathcal{P}$ holds as $n_1, \ldots, n_{L-1} \to \infty$ strongly".*

(ii) *We say that such a statement $\mathcal{S}$ holds for $(n_l)_{1 \leq l \leq L-1} \asymp n$, if $\mathcal{S}$ holds for all $r \in \mathcal{R}_L$ with $r_l(m) \asymp m$ for all $1 \leq l < L$. This means that $\mathcal{S}(r)$ holds for all $r \in \mathcal{R}_L$ such that $r_p(m)/r_q(m) \to \alpha_{p,q} \in (0, \infty)$ as $m \to \infty$. We will also write "$\mathcal{P}$ holds as $(n_l)_{1 \leq l < L} \asymp n$".*

(iii) *We say that such a statement $\mathcal{S}$ holds weakly, if $\mathcal{S}$ holds for at least one $r \in \mathcal{R}_L$. We will also write "$\mathcal{P}$ holds as $n_1, \ldots, n_{L-1} \to \infty$ weakly". This type of infinite-width limit is tightly connected to the sequential infinite-width limit.*

**Remark 2.1** (Connection between weak and sequential infinite-width limits). *In the sequential infinite-width limit, meaning that $n_1 \to \infty, \ldots, n_{L-1} \to \infty$ sequentially, the layer widths are not strictly finite. This is opposed to applications, where layer widths may be large but finite. Hence, the infinite-width limits using width functions as explained above are more meaningful in practice. For a sequential limit $\lim_{n_1 \to \infty} \lim_{n_2 \to \infty} f(n_1, n_2) = f^*$, we can find functions $n_1(m)$ and $n_2(m)$ such that $\lim_{m \to \infty} f(n_1(m), n_2(m)) = f^*$. However, the rate at which $n_1(m), n_2(m)$ diverge as $m \to \infty$ cannot generally be controlled. Since the weak infinite-width limit allows for arbitrary rates, any sequential infinite-width limit can hence be turned into a weak infinite-width limit as defined in Definition 2.2 (iii).*

We use Definition 2.2 to paraphrase the convergence of ANNs to GPs in the infinite-width limit:

**Theorem 2.1** (Theorem 4 from Matthews et al. [2018])**.** *Any network function $f$ of depth $L$ defined as in Section 2.1 with continuous activation function $\sigma$ that satisfies the linear envelope property, i.e., there exist $c, m \geq 0$ with $|\sigma(u)| \leq c + m|u|$ for all $u \in \mathbb{R}$, converges in distribution as $n_1, \ldots, n_{L-1} \to \infty$ strongly to a multidimensional Gaussian process $(X_j)_{j=1}^{n_L}$ for any fixed countable input set $(x_i)_{i=1}^{\infty}$. It holds $X_j \overset{iid}{\sim} \mathcal{N}(0, \Sigma^{(L)})$ where the covariance function $\Sigma^{(L)}$ is recursively given by*

$$\Sigma^{(1)}(x, x') = \frac{\sigma_w^2}{n_0}\langle x, x'\rangle + \sigma_b^2, \quad \Sigma^{(L)}(x, x') = \sigma_w^2 \, \mathbb{E}_{g\sim\mathcal{N}(0,\Sigma^{(L-1)})}[\sigma(g(x))\,\sigma(g(x'))] + \sigma_b^2. \quad (1)$$

We also write $\Sigma^{(L)} = \Sigma_\sigma$. The theorem states that the distribution of the network function, which is given by its randomly initialized weights, approaches the distribution of independent GPs as the hidden layer widths increase. Hence, a large-width network will approximately be a realization of the respective GPs. Note that this result can be generalized to non-continuous activation functions without well-defined derivatives that fulfil the linear envelope property, like $\sigma(z) = \text{sign}(z)$. We provide a rigorous proof of this kind of generalization for the case $n_1, \ldots, n_{L-1} \to \infty$ weakly in Theorem E.3. $\Sigma_\sigma$ remains well-defined in this case since the expectation in Equation (1) does. When approximating the sign function with scaled error function, i.e., $\sigma(z) = \text{erf}_m(z) := \text{erf}(m \cdot z)$, it holds that $\lim_{m\to\infty} \Sigma_{\text{erf}_m} = \Sigma_{\text{sign}}$ due to the dominated convergence theorem.

## 2.3 Direct extension of the neural tangent kernel in the infinite-width limit

We consider a dataset $\mathcal{D} = (\mathcal{X}, \mathcal{Y})$ with $\mathcal{X} = (x_1, \ldots, x_d) \in \mathbb{R}^{d \cdot n_0}$ and $\mathcal{Y} = (y_1, \ldots, y_d) \in \mathbb{R}^{d \cdot n_L}$. To solve the regression problem, i.e., to find weights $\theta \in \mathbb{R}^P$ such that $f(x_i; \theta) = y_i$ for all $i = 1, \ldots, d$, we apply gradient descent in continuous time, also know as gradient flow, with learning rate $\eta$ and loss function $\mathcal{L} \colon \mathbb{R}^{d \cdot n_L} \times \mathbb{R}^{d \cdot n_L} \to \mathbb{R}$. This means that, using the chain rule, the learning rule can then be written as

$$\frac{\mathrm{d}}{\mathrm{d}t}\theta_t = -\eta \, \nabla_\theta \mathcal{L}(f(\mathcal{X}; \theta_t); \mathcal{Y}) = -\eta \, J_\theta f(\mathcal{X}; \theta_t)^\mathsf{T} \, \nabla_{f(\mathcal{X}; \theta_t)} \mathcal{L}(f(\mathcal{X}; \theta_t); \mathcal{Y}). \quad (2)$$

To derive the NTK, we observe that the network function then evolves according to

$$\frac{\mathrm{d}}{\mathrm{d}t}f(x; \theta_t) = J_\theta f(x; \theta_t)\, \frac{\mathrm{d}}{\mathrm{d}t}\theta_t \overset{(2)}{=} -\eta \, J_\theta f(x; \theta_t) J_\theta f(\mathcal{X}; \theta_t)^\mathsf{T} \, \nabla_{f(\mathcal{X}; \theta_t)} \mathcal{L}(f(\mathcal{X}; \theta_t); \mathcal{Y}) \quad (3)$$

$$=: -\eta \, \hat{\Theta}_t(x, \mathcal{X}) \nabla_{f(\mathcal{X}; \theta_t)} \mathcal{L}(f(\mathcal{X}; \theta_t); \mathcal{Y}),$$

where we implicitly defined the empirical neural tangent kernel as $\hat{\Theta}(x, x') := J_\theta f(x; \theta) J_\theta f(x'; \theta)^\mathsf{T}$, which depends on the current weights $\theta = \theta_t$. This means that the learning dynamics of the network functions during training are given by a kernel whose entries are the scalar products between the gradients of the network's output neuron activity, $\hat{\Theta}_{ij}(x, x') = \langle \nabla_\theta f_i(x; \theta), \nabla_\theta f_j(x'; \theta)\rangle$. The key result on the NTK is that the empirical NTK converges in the infinite-width limit to a constant kernel, the analytic NTK, at initialization, $\theta = \theta_0$, and during training, $\theta = \theta_t$:

**Theorem 2.2** (Theorem 1 from Jacot et al. [2018] for general $\sigma_w > 0$)**.** *For any network function of depth $L$ defined as in Section 2.1 with Lipschitz continuous activation function $\sigma$, $\hat{\Theta} =: \hat{\Theta}^{(L)}$ converges in probability to a constant kernel $\Theta^{(L)} \otimes \mathrm{I}_{n_L}$ as $n_1, \ldots, n_{L-1} \to \infty$ weakly. This means that for all $x, x' \in \mathbb{R}^{n_0}$ and $1 \leq i, j \leq n_L$, it holds $\hat{\Theta}_{ij}^{(L)}(x, x') \to \delta_{ij} \, \Theta^{(L)}(x, x')$ in probability, where $\delta_{ij}$ denotes the Kronecker delta. We call $\Theta^{(L)}$ the analytic neural tangent kernel of the network, which is recursively given by*

$$\Theta^{(1)}(x, x') = \Sigma^{(1)}(x, x'), \quad \Theta^{(L)}(x, x') = \Sigma^{(L)}(x, x') + \Theta^{(L-1)}(x, x') \cdot \dot{\Sigma}^{(L)}(x, x'),$$

*where $\Sigma^{(l)}$ are defined as in Theorem 2.1 and we define*

$$\dot{\Sigma}^{(L)}(x, x') = \sigma_w^2 \, \mathbb{E}_{g\sim\mathcal{N}(0,\Sigma^{(L-1)})}\left[\dot{\sigma}(g(x))\,\dot{\sigma}(g(y))\right]. \quad (4)$$

We also write $\Theta^{(L)} = \Theta_\sigma$. If $\theta_t$ are the weights during gradient flow learning at time $t \geq 0$ as before, Theorem 2.2 shows that $\hat{\Theta}_t^{(L)} \to \Theta^{(L)} \otimes \mathrm{I}_{n_L}$ for $t = 0$ in the infinite-width limit. This reveals

that the gradients of the output neurons, $\nabla_\theta f_i(x; \theta)$, are mutually orthogonal. Under additional assumptions this convergence also holds for the whole training duration, $t > 0$, see Theorem 2 of Jacot et al. [2018] for the case $n_1, \dots, n_{L-1} \to \infty$ weakly, and Chapter G of Lee et al. [2019] for $(n_l)_{1 \le l < L} \asymp n$. Hence, the kernel that describes the learning dynamics stays constant in the infinite-width limit. This implies that the distribution of the network function during training also converges to GPs [Lee et al., 2019, Theorem 2.2]. Then, any output neuron after training has mean $m(x) = \Theta^{(L)}(x, \mathcal{X}) \Theta^{(L)}(\mathcal{X}, \mathcal{X})^{-1} \mathcal{Y}$ under the assumption of a mean squared error (MSE) loss, see Section C.2.1. The expression for the mean is equivalent to kernel regression with the NTK.

The above results show that gradient flow for networks with randomly initialized weights is characterized by the analytic NTK $\Theta^{(L)}$ in the infinite-width limit. Since we are interested in gradient flow for networks with activation functions inadequate for gradient descent training, we want to know whether the analytic NTK can be extended to such activation functions. We see that the derivative of the activation function does not have to be defined point-wise for Equation (4). In particular, activation functions with distributional derivatives, e.g., $\frac{\mathrm{d}}{\mathrm{d}z} \mathrm{sign}(z) = 2\,\delta(z)$ can be taken into consideration, where $\delta$ denotes the Dirac delta distribution. If we again approximate the sign function with scaled error functions $\mathrm{erf}_m, m \in \mathbb{N}$, it holds

$$\mathbb{E}_{g \sim \mathcal{N}(0, \Sigma_{\mathrm{erf}_m})} \left[ \dot{\mathrm{erf}}_m(g(x))\, \dot{\mathrm{erf}}_m(g(y)) \right] \xrightarrow{m \to \infty} \mathbb{E}_{g \sim \mathcal{N}(0, \Sigma_{\mathrm{sign}})} \left[ 2\,\delta(g(x)) \cdot 2\,\delta(g(y)) \right], \quad (5)$$

in case the right-hand side exists. A rigorous analysis of this limit can be found in Section D.1. Heuristically, a simple observation suffices: if $x = y$, we have a one-dimensional integral over two delta distributions, which yields infinity. On the other hand, if $x \ne y$ and $\Sigma_{\mathrm{sign}}$ is non-degenerate, a two-dimensional integral over two delta distributions yields a finite value. We derive analytic expressions for $\lim_{m \to \infty} \Theta_{\mathrm{erf}_m} =: \Theta_{\mathrm{sign}}$ in Lemma D.3. We call this kernel singular, because $\Theta_{\mathrm{sign}}(x, x) = \infty$ and $\Theta_{\mathrm{sign}}(x, y) \in \mathbb{R}$ if $x \ne y$. By the same reasoning, this divergence occurs for any activation function with jumps. Note that for the mean given by kernel regression, this implies $m(x_i) = y_i$ and $m(x) = 0$ if $x$ is not a training point. Intuitively, this is because the network is initialized with zero mean and different input points are uncorrelated under the singular kernel, so only the training points are learned. A limit kernel with this property also arises if the activation function is fixed but the depth of the network goes to infinity as was shown by Radhakrishnan et al. [2023]. We adopt the ideas of their proof to show that the sign of $m$ is still useful for classification:

**Theorem 2.3** (Inspired by Lemma 5 of Radhakrishnan et al. [2023]; see Theorem D.4). *Let $\sigma_b^2 > 0$ or let all $x_i \in \mathbb{R}^{n_0}$ be pairwise non-parallel. Let $L \ge 2$ and $x_i \in \mathcal{S}_R^{n_0 - 1}$ for all $i = 1, \dots, d$, where $\mathcal{S}_R^{n_0 - 1} \subseteq \mathbb{R}^{n_0}$ is the sphere of radius $R$. Assuming that $\Theta_\infty^{(L)}(x, \mathcal{X}) \mathcal{Y} \ne 0$ for almost all $x \in \mathcal{S}_R^{n_0 - 1}$, it holds*

$$\lim_{m \to \infty} \mathrm{sign}\left( \Theta_m^{(L)}(x, \mathcal{X}) \Theta_m^{(L)}(\mathcal{X}, \mathcal{X})^{-1} \mathcal{Y} \right) = \mathrm{sign}\left( \Theta_\infty^{(L)}(x, \mathcal{X}) \mathcal{Y} \right) \quad \textit{a.e. on } \mathcal{S}_R^{n_0 - 1}.$$

The estimator $\Theta_\infty^{(L)}(x, \mathcal{X}) \mathcal{Y} = \sum_{i=1}^n \Theta_\infty^{(L)}(x, x_i)\, y_i$ has the form of a so-called Nadaraya-Watson estimator and is well-defined for singular kernels such as $\Theta^{(L)}$. If we assume a classification problem in the sense that $\mathcal{Y} \subseteq \{-1, 1\}^n$, it holds $\mathrm{sign}\big(\Theta_\infty^{(L)}(x_i, \mathcal{X}) \mathcal{Y}\big) = \Theta_\infty^{(L)}(x_i, \mathcal{X}) \mathcal{Y}$ at training point $x_i$.

## 2.4 Generalization of the neural tangent kernel and application to surrogate gradient learning

The above singularity of the limit kernel can be avoided by considering surrogate gradient learning instead of gradient descent. First, we introduce a generalization of the NTK that later leads to the surrogate gradient NTK.

By definition and originally due to the chain rule, the empirical NTK consists of two Jacobian matrices of the network function with respect to the weights. The Jacobian matrix can be thought of as a recursive formula, $J_\theta f(x; \theta) = G(x, \theta; \sigma, \dot{\sigma})$, which is given by the input $x$ and the architecture of the network, including the activation function $\sigma$ and its derivative $\dot{\sigma}$. This formula can be modified to define a quasi-Jacobian matrix, $J^{\sigma_1, \tilde{\sigma}_1}(x; \theta) := G(x, \theta; \sigma_1, \tilde{\sigma}_1)$, where $\tilde{\sigma}_1$ does not have to be the derivative of $\sigma_1$. Analogous to the definition of the empirical NTK we define the empirical generalized NTK to be $\hat{I}(x, x') := J^{\sigma_1, \tilde{\sigma}_1}(x; \theta)\, J^{\sigma_2, \tilde{\sigma}_2}(x'; \theta)^\mathsf{T}$, where $\sigma_1, \tilde{\sigma}_1, \sigma_2, \tilde{\sigma}_2$ are specified in the following theorem.

**Theorem 2.4** (Generalized version of Theorem 1 by Jacot et al. [2018]; see Theorem E.4). *For activation functions $\sigma_1, \sigma_2$ and so-called surrogate derivatives $\tilde{\sigma}_1, \tilde{\sigma}_2$ such that $\sigma_1, \sigma_2, \tilde{\sigma}_1,$ and $\tilde{\sigma}_2$ satisfy the linear envelope property and are continuous except for finitely many jump points, denote the empirical generalized neural tangent kernel*

$$\hat{I}^{(L)}(x, x') = J^{(L),\sigma_1,\tilde{\sigma}_1}(x; \theta) \, J^{(L),\sigma_2,\tilde{\sigma}_2}(x'; \theta)^{\intercal} \quad \text{for } x, x' \in \mathbb{R}^{n_0},$$

*as before. Then, for any $x, x' \in \mathbb{R}^{n_0}$ and $1 \leq i, j \leq n_L$, it holds $\hat{I}_{ij}^{(L)}(x, x') \xrightarrow{\mathcal{P}} \delta_{ij} I^{(L)}(x, x')$, as $n_1, \ldots, n_{L-1} \to \infty$ weakly. We call $I^{(L)}$ the analytic generalized neural tangent kernel, which is recursively given by*

$$I^{(1)}(x, x') = \Sigma_{1,2}^{(1)}(x, x'), \ I^{(L)}(x, x') = \Sigma_{1,2}^{(L)}(x, x') + I^{(L-1)}(x, x') \cdot \tilde{\Sigma}_{1,2}^{(L)}(x, x') \text{ for } L \geq 2, \text{ with}$$

$$\Sigma_{1,2}^{(L)}(x, x') = \sigma_w^2 \, \mathbb{E}[\sigma_1(Z_1) \, \sigma_2(Z_2)] + \sigma_b^2 \ \text{for } L \geq 2 \ \text{and} \ \Sigma_{1,2}(x, x') = \frac{\sigma_w^2}{n_0}\langle x, x' \rangle + \sigma_b^2, \text{ where}$$

$$\tilde{\Sigma}_{1,2}^{(L)}(x, x') = \sigma_w^2 \, \mathbb{E}[\tilde{\sigma}_1(Z_1) \, \tilde{\sigma}_2(Z_2)] \ \text{ and } \ (Z_1, Z_2) \sim \mathcal{N}\left(0, \begin{pmatrix} \Sigma_1^{(L-1)}(x,x) & \Sigma_{1,2}^{(L-1)}(x,x') \\ \Sigma_{1,2}^{(L-1)}(x,x') & \Sigma_2^{(L-1)}(x',x') \end{pmatrix}\right).$$

$\Sigma_1$ and $\Sigma_2$ denote the covariance functions of the Gaussian processes that arise from network functions $f_1, f_2$ with activation functions $\sigma_1, \sigma_2$, respectively, in the infinite-width limit. The covariance between these two Gaussian processes is denoted by $\Sigma_{1,2}$. This covariance function is asymmetric in the sense that $\text{Cov}[f_1(x_1), f_2(x_2)] \neq \text{Cov}[f_1(x_2), f_2(x_1)]$ in general.

We show in Theorem 2.4 that the generalized empirical NTK tends to a generalized analytic NTK at initialization in an infinite-width limit. Now, the SGL rule can be written using the generalized NTK, similar to Equation (3):

$$\frac{\mathrm{d}}{\mathrm{d}t} f(x; \theta_t) = J_\theta f(x; \theta_t) \, \frac{\mathrm{d}}{\mathrm{d}t} \theta_t = -\eta \, J_\theta f(x; \theta_t) J^{\sigma,\tilde{\sigma}}(x; \theta_t)^{\intercal} \, \nabla_{f(\mathcal{X};\theta_t)} \mathcal{L}(f(\mathcal{X}; \theta_t); \mathcal{Y}) \quad (6)$$

$$=: -\eta \, \hat{I}_t(x, \mathcal{X}) \, \nabla_{f(\mathcal{X};\theta_t)} \mathcal{L}(f(\mathcal{X}; \theta_t); \mathcal{Y}) \quad (7)$$

Here, we chose $\sigma_1 = \sigma_2 = \sigma$, $\tilde{\sigma}_1 = \dot{\sigma}$ and $\tilde{\sigma}_2 = \tilde{\sigma}$ in the previous definition of the generalized NTK. This we call the surrogate gradient NTK (SG-NTK) with activation function $\sigma$ and surrogate derivative $\tilde{\sigma}$. Compared to the classical NTK, one of the true Jacobian matrices is replaced by the quasi-Jacobian matrix, since the learning rule is SGL instead of gradient flow. Note that we assume that the derivative of the activation function, $\dot{\sigma}$, exists. Theorem 2.4 shows convergence at time $t = 0$. We extend this convergence to $t > 0$ in the following theorem:

**Theorem 2.5** (Based on Theorem G.2 from Lee et al. [2019]; see Theorem E.5). *Let $\sigma, \dot{\sigma}, \tilde{\sigma}$ as before, all Lipschitz continuous, and $\dot{\sigma}, \tilde{\sigma}$ bounded. Let $f_t$ be a network function with depth $L$ initialized as in Section 2.1 and trained with MSE loss and SGL with surrogate derivative $\tilde{\sigma}$. Assume that the generalized NTK converges in probability to the analytic generalized NTK of Theorem 2.4, $\hat{I}^{(L)} \to I^{(L)} \otimes \mathrm{I}_{n_L}$, as $(n_l)_{l=1}^{L-1} \asymp n$. Furthermore, assume that the smallest and largest eigenvalue of the symmetrization of $I^{(L)}(\mathcal{X}, \mathcal{X})$, $S^{(L)} := \left( I^{(L)}(\mathcal{X}, \mathcal{X}) + I^{(L)}(\mathcal{X}, \mathcal{X})^{\intercal} \right)/2$, are given by $0 < \lambda_{\min} \leq \lambda_{\max} < \infty$ and that the learning rate is given by $\eta > 0$. Then, for any $\delta > 0$ there exist $R > 0, N \in \mathbb{N}$ and $K > 1$ such that for every $n \geq N$, the following holds with probability at least $1 - \delta$ over random initialization:*

$$\sup_{t \in [0,\infty)} \left\| \hat{I}_t^{(L)}(\mathcal{X}, \mathcal{X}) - I^{(L)}(\mathcal{X}, \mathcal{X}) \right\|_F \leq \frac{6K^3 R}{\lambda_{\min}} n^{-\frac{1}{2}}, \ \text{ where } \| \cdot \|_F \text{ denotes the Frobenius norm.}$$

We also write $I^{(L)} = I_{\sigma,\tilde{\sigma}}$ or simply $I_\sigma$. The explicit analytic expression of the SG-NTK is derived in Section E.2 for activation function $\sigma = \text{erf}_m$, $m \in \mathbb{N}$ and surrogate derivative $\tilde{\sigma} = \text{erf}$ as well as general surrogate derivatives. $I_{\text{erf}_m,\tilde{\sigma}}$ converges as $m \to \infty$, because compared to Equation (5) we obtain $\mathbb{E}[\dot{\text{erf}}_m(g(x)) \, \tilde{\sigma}(g(y))] \to \mathbb{E}[2\delta(g(x)) \, \tilde{\sigma}(g(y))]$ and the delta distribution yields a finite value. In Section E.2, we show this rigorously and derive the analytic expressions. Hence, we define $I_{\text{sign},\tilde{\sigma}} := \lim_{m \to \infty} I_{\text{erf}_m,\tilde{\sigma}}$.

For any $m \in \mathbb{N}$ we can consider the SGL dynamics given by Equation (7) in the infinite-width limit to obtain

$$\frac{\mathrm{d}}{\mathrm{d}t} f(x; \theta_t) = -\eta \, I_{\text{erf}_m}(x, \mathcal{X}) \, \nabla_{f(\mathcal{X};\theta_t)} \mathcal{L}(f(\mathcal{X}; \theta_t); \mathcal{Y}).$$

With MSE error, this equation is solved by a GP with mean $m(x) = I_{\mathrm{erf}_m}(x, \mathcal{X})I_{\mathrm{erf}_m}(\mathcal{X}, \mathcal{X})^{-1}\mathcal{Y}$ for $t \to \infty$ and it is natural to assume that the networks trained with SGL converge in distribution to this GP in the infinite-width limit, analogous to the standard NTK case [Lee et al., 2019, Theorem 2.2]. However, the empirical counterpart to $I_{\mathrm{sign}}$ does not exist as we cannot formulate an empirical SG-NTK for the sign activation function due to the missing Jacobian matrix, compare Equation (6). We suggest that even in this case the network trained with SGL will converge in distribution to the GP given by $I_{\mathrm{sign}}$, since SGL with activation function $\mathrm{erf}_m$ will approach SGL with activation function sign as $m \to \infty$ and the GP given by $I_{\mathrm{erf}_m}$ will approach the GP given by $I_{\mathrm{sign}}$ as $m \to \infty$. Numerical experiments in Section 3 indicate that this is indeed the case. We conclude that, remarkably, the SG-NTK can be defined for network functions without well-defined gradients and is informative about their learning dynamics under SGL.

## 3 Numerical experiments

We numerically illustrate the divergence of the analytic NTK, $\Theta_{\mathrm{erf}_m}$, shown in Section 2.3 and the convergence of the SG-NTK in the infinite-width limit, $\hat{I}^{(L)} \to I^{(L)}$, at initialization and during training shown in Section 2.4. Simultaneously, we visualize the convergence of the analytic SG-NTK, $I_{\mathrm{erf}_m} \to I_{\mathrm{sign}}$. We consider a regression problem on the unit sphere $\mathcal{S}^1 = \{x \in \mathbb{R}^2 : \|x\| = 1\}$ with $|\mathcal{X}| = 15$ training points, which is shown in Figure B.1, and train 10 fully connected feedforward networks with two hidden layers, and activation function $\mathrm{erf}_m$ for $t = 10000$ time steps and with MSE loss. The NTK only depends on the dot product [Radhakrishnan et al., 2023] and thus the angle between its two arguments, $\Delta\alpha = \sphericalangle(x, x')$. Hence, we plot the NTKs as functions of this angle, where $\Delta\alpha = 0$ corresponds to $x = x'$.

In Figure 1, the empirical and analytic NTKs for the networks described above and trained with gradient descent are plotted for $m \in \{2, 5, 20\}$ and hidden layer widths $n \in \{10, 100, 500, 1000\}$. In addition, the analytic NTK for $m \to \infty$ is plotted. Note that the steep slope of $\mathrm{erf}_m$ for $m = 20$ results in $\mathrm{erf}_m$ being very close to the sign function. For any $m$, we observe that the empirical NTKs converge to the analytic NTK both at initialization and after training as NTK theory states. Figure B.3 illustrates this further by displaying the mean squared errors between the empirical NTKs and the respective analytic NTK. The convergence slows down for larger $m$. Further, the plots confirm that the analytic NTKs diverge as $m \to \infty$ if and only if $\Delta\alpha = 0$. To show this more clearly, we scaled the y-axis with the inverse hyperbolic sine (asinh), which is approximately linear for small absolute values and logarithmic for large absolute values.

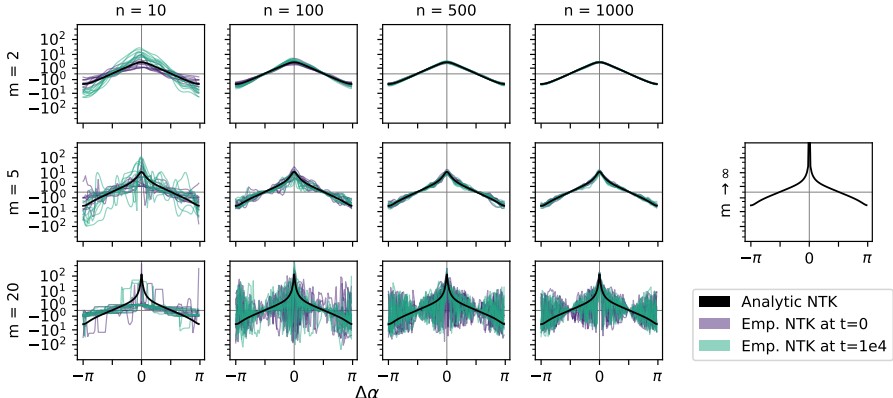

Figure 1: We plot empirical and analytic NTKs of 10 networks for different hidden layer widths $n$ and activation functions $\mathrm{erf}_m$. The kernels are plotted at initialization and after gradient descent training with $t = 1e4$ time steps, learning rate $\eta = 0.1$, and MSE error. The y-axis is asinh-scaled.

For Figure 2, we use the same setup as before, but train the networks using SGL with surrogate derivative $\tilde{\sigma} = \dot{\mathrm{erf}}$, and compare the empirical and analytic SG-NTKs instead of NTKs. We observe that the empirical SG-NTKs converge to the analytic SG-NTK as $n \to \infty$ both at initialization and after training in accordance with Theorem 2.4 and Theorem 2.5. Figure B.4 illustrates this further by displaying the mean squared errors between empirical SG-NTKs and respective analytic SG-NTK.

Moreover, we observe that the analytic SG-NTKs indeed converge to a finite limit as $m \to \infty$, as shown in Section 2.4.

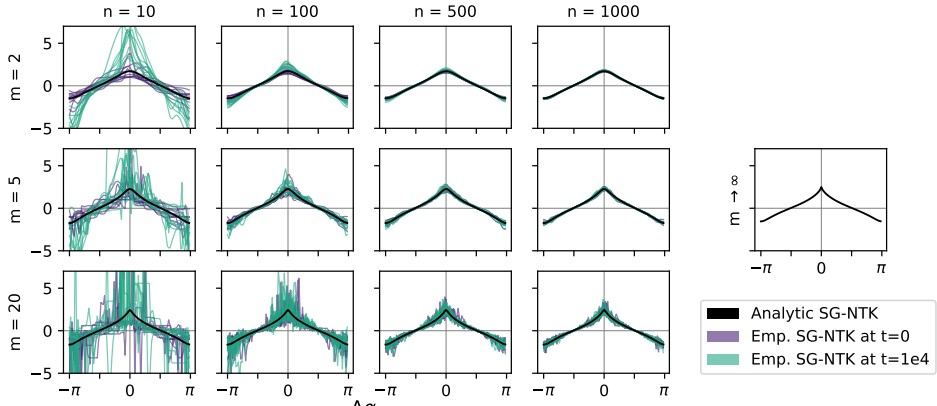

Figure 2: We plot empirical and analytic SG-NTKs of ten networks for different hidden layer widths $n$ and activation functions $\mathrm{erf}_m$. The kernels are plotted at initialization and after surrogate gradient learning with $t = 1\mathrm{e}4$ time steps, learning rate $\eta = 0.1$, MSE error, and surrogate derivative $\tilde{\sigma} = \dot{\mathrm{erf}}$.

Finally, we consider SGL for networks with the same architecture and training objective as before, but with sign activation function, which can be seen as the case $m \to \infty$ of the setups above. We examine whether the distribution of network functions trained with SGL agrees with the distribution of the GP given by the limit kernel $I_{\mathrm{sign}}$. Specifically, we compare 500 networks trained with SGL for $t = 30000$ time steps, which represent the distribution of the network function after training, to the mean and confidence band of the GP. The mean of the GP is given by kernel regression using the SG-NTK, $m(x) = I_{\mathrm{sign}}(x, \mathcal{X})I_{\mathrm{sign}}(\mathcal{X}, \mathcal{X})^{-1}\mathcal{Y}$, and the confidence band is given by $m(x) \pm 2\sigma_{\mathrm{GP}}(x)$, where $\sigma_{\mathrm{GP}}(x)$ is the standard deviation at $x$. We observe in Figure 3a that the mean of the trained networks is close to the GP's mean for network width $n = 500$ and that most networks lie within the confidence band. The mean of the networks differs from the kernel regression using the kernel $\Sigma_{\mathrm{sign}}$. Figure 3b shows that this agreement between SGL and the SG-NTK already exists for a network width of $n = 100$, demonstrating that the SG-NTK predicts the SGL dynamics of networks with moderate width. Note that the variance in the networks' output and the confidence band can be reduced (see Arora et al. [2019] and Section A).

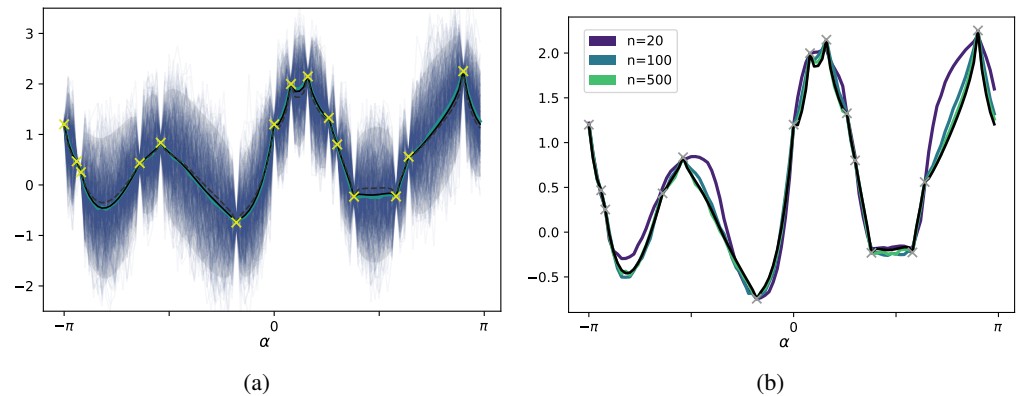

(a)  (b)

Figure 3: Comparison of SGL learning in networks with different hidden layer widths with SG-NTK predictions. **(a)** 500 networks (blue) with sign activation function and hidden layer widths $n = 500$ trained with SGL using the surrogate derivative $\tilde{\sigma} = \dot{\mathrm{erf}}$ for $t = 3\mathrm{e}4$ time steps plotted together with their mean (cyan), the SG-NTK-GP's mean (black) and confidence band (grey), and the $\Sigma_{\mathrm{sign}}$ kernel regression (dashed). Training points are indicated with crosses. **(b)** The mean of ensembles of 500 networks is plotted as in (a) for different hidden layer widths.

# 4    Conclusion

Gradient descent training is not applicable to networks with sign activation function. In the present study, we have first shown that this is even true for the infinite-width limit in the sense that the NTK diverges to a singular kernel. We found that this singular kernel still has interesting properties and allows for classification, but it is unusable for regression.

We then studied SGL, which is applicable to networks with sign activation function. We defined a generalized version of the NTK that can be applied to SGL and derived a novel SG-NTK. We proved that the convergence of the NTK in the infinite-width limit extends to the SG-NTK, both at initialization and during training. Strikingly, we were able to derive an SG-NTK for the sign activation function, $I_{\text{sign}}$, by approximating the sign function with error functions. We suggest that this SG-NTK predicts the learning dynamics of SGL, and support this claim with heuristic arguments and numerical simulations.

A limitation of our work is that due to the considered NTK framework, our results are naturally only applicable to sufficiently wide networks with random initialization. Further, we only prove the convergence of the SG-NTK during training for activation functions with well-defined derivatives. More rigorous analysis should be carried out on how the connection between SGL and the SG-NTK carries over to activation functions with jumps, as shown by our simulations.

Our derivation of the SG-NTK opens a novel path towards addressing the many unanswered questions regarding the training of binary networks, in particular regarding the class of functions that SGL learns for wide networks and how that class differs for different activation functions and surrogate derivatives.

## Acknowledgments and Disclosure of Funding

We thank Andreas Eberle for helpful discussions. We thank the German Federal Ministry of Education and Research (BMBF) for support via the Bernstein Network (Bernstein Award 2014, 01GQ1710).

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

# A   Additional remarks on the numerical experiments

**Weight initialization and implementation.**    All networks are initialized with $\sigma_w = 1, \sigma_b = 0.1$
For the implementation of the NTK and SG-NTK we use the JAX package [Bradbury et al., 2018]
and Neural Tangents package [Novak et al., 2020, 2022, Han et al., 2022, Sohl-Dickstein et al., 2020,
Hron et al., 2020] with modifications. Computations were done using an Intel Core i7-1355U CPU
and 16 GB RAM. The simulations for Figure 1 and Figure 2 took two hours each. The simulations
for Figure 3 took 12 hours.

**Variance reduction trick.**    The variance in the outputs of the networks trained with gradient
descent or SGL and the confidence band given the NTK or SG-NTK respectively can be reduced by
multiplying the weights of the last layer with a constant $\kappa < 1$ at initialization [Arora et al., 2019].
This is explained in detail at the end of Section C.2.1. We can see from Figure B.2 that the agreement
between the distribution of the trained networks and the distribution given by the SG-NTK still holds;
however, network width and training time have to be increased.

# B   Additional figures

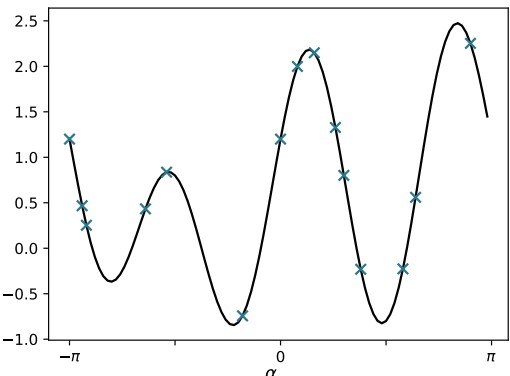

Figure B.1: Target function and training points for the numerical experiments, $f(x, y) = 4xy^2 - 0.8x^3 + 1.2y^2 - 0.8x^2 y$.

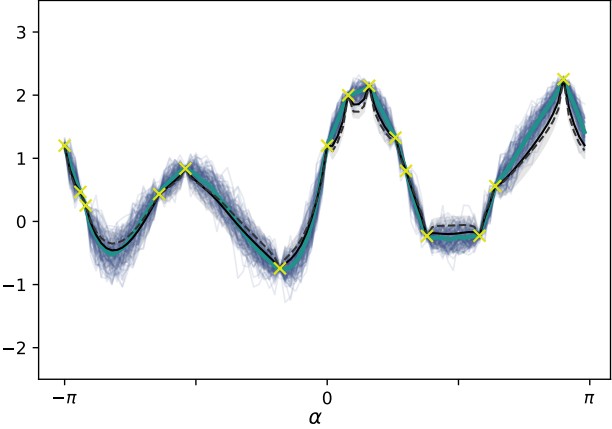

Figure B.2: Network functions of 100 networks with sign activation function and hidden layer widths
$n = 500$ trained with SGL using the surrogate derivative $\tilde{\sigma} = \mathrm{erf}$ for $t = 2\mathrm{e}4$ time steps plotted
together with the SG-NTK-GP's mean and confidence band. The parameters of the last layer have
been multiplied with $\kappa = 0.2$ at initialization.

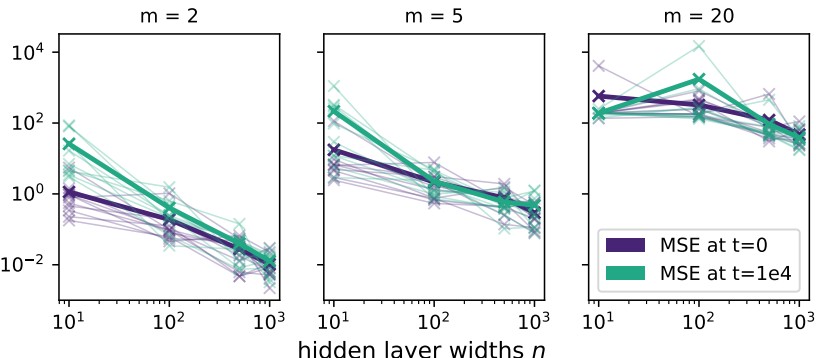

Figure B.3: Mean squared errors between empirical NTKs and analytic NTKs in Figure 1 for all 10 networks (thin lines) and averaged over the 10 networks (thick lines).

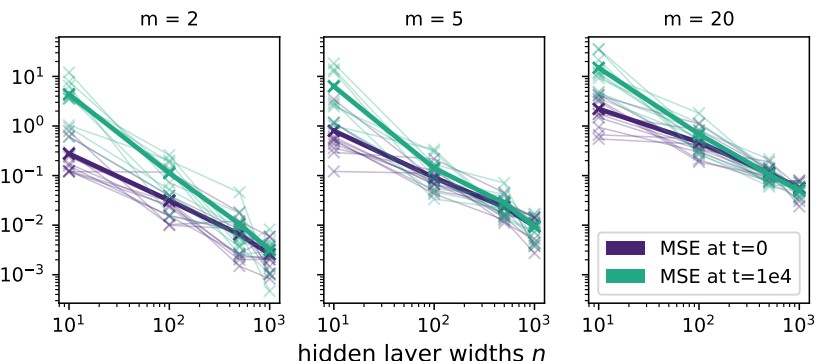

Figure B.4: Mean squared errors between empirical SG-NTKs and analytic SG-NTKs in Figure 2 for all 10 networks (thin lines) and averaged over the 10 networks (thick lines).

## C  Neural tangent kernel theory

### C.1  Introduction of the neural tangent kernel

We begin by defining an artificial neural network with a parameterization suitable for considering the limit of infinitely many hidden neurons. This parameterization is called *neural tangent kernel parameterization* and differs from standard parameterization of multilayer perceptrons by a rescaling factor of $1/\sqrt{n_l}$ in layer $l + 1$, where $n_l$ is the width of layer $l$. We follow the slightly more general definition in [Lee et al., 2019, Equation (1)]. A discussion of this kind of parameterization can be found in [Jacot et al., 2018, Remark 1].

**Definition C.1** (Artificial neural network with NTK parameterization). *Let $L$ be the depth of the network, $n_k$ for $k = 0, \ldots, L$ the widths of the layers, and $\sigma \colon \mathbb{R} \to \mathbb{R}$ an activation function. We draw network weight matrices $W^{(l)} \in \mathbb{R}^{n_l \times n_{l-1}}$ and biases $b^{(l)} \in \mathbb{R}^{n_l}$ for $l = 1, \ldots, L$ from a probability distribution such that $W_{ij}^{(l)}, b_i^{(l)} \overset{iid}{\sim} \mathcal{N}(0, 1)$. For the parameters, we denote by $\theta^{(l)} = (W^{(l)}, b^{(l)})$ the parameters of layer $l$ and by $\theta^{(1 \colon l)} = (\theta^{(1)}, \theta^{(2)}, \ldots, \theta^{(l)})$ the parameters of layers 1 up to and including $l$. Given some $\sigma_w > 0$ and $\sigma_b \geq 0$ we then define for all $x \in \mathbb{R}^{n_0}$*

$$h^{(1)}\left(x; \theta^{(1)}\right) = \frac{\sigma_w}{\sqrt{n_0}} W^{(1)} x + \sigma_b\, b^{(1)},$$

$$h^{(l+1)}\left(x; \theta^{(1 \colon l+1)}\right) = \frac{\sigma_w}{\sqrt{n_l}} W^{(l+1)} \sigma\left(h^{(l)}\left(x; \theta^{(1 \colon l)}\right)\right) + \sigma_b\, b^{(l+1)} \quad \text{for } l = 1, \ldots, L - 1.$$

Therefore, $h^{(l)}(\,\cdot\,;\theta^{(1:\,l)})$ is a map from $\mathbb{R}^{n_0}$ to $\mathbb{R}^{n_l}$ and we use the short-hand notation $h^{(l)}(x;\theta) = h^{(l)}(x;\theta^{(1:\,l)})$. Finally, we define our network function

$$f(\,\cdot\,;\theta) = h^{(L)}(\,\cdot\,;\theta)\colon \mathbb{R}^{n_0} \to \mathbb{R}^{n_L}.$$

With this definition, the total number of parameters, $P = |\theta|$, is

$$P = \sum_{l=1}^{L} \left|\theta^{(l)}\right| = \sum_{l=1}^{L} n_l(n_{l-1} + 1).$$

Given such a network function with NTK parameterization, we consider a dataset $\mathcal{D} = (\mathcal{X}, \mathcal{Y})$ with $\mathcal{X} = (x_1, \ldots, x_d) \in \mathbb{R}^{d \cdot n_0}$ and $\mathcal{Y} = (y_1, \ldots, y_d) \in \mathbb{R}^{d \cdot n_L}$. First, we want to solve the regression problem, i.e., find parameters $\theta'$ such that $f(x_i; \theta') = y_i$ for all $i = 1, \ldots, d$. Later, we will also consider the classification problem, i.e., we assume $y_i \in \{-1, 1\}$ and want to solve $\text{sign}(f(x_i; \theta')) = y_i$ for all $i = 1, \ldots, d$. Tackling both cases from the regression perspective, we define the so-called loss functional and loss function. The following two definitions are similarly formulated by Jacot et al. [2018].

**Definition C.2** (Loss functional and loss function). *Let $\mathcal{F} = \{g \mid g\colon \mathbb{R}^{n_0} \to \mathbb{R}^{n_L}\}$ and $\mathfrak{L}\colon \mathcal{F} \to \mathbb{R}$ a so-called loss functional. In addition, let $\mathfrak{L}$ be convex, i.e., for all $\lambda \in [0, 1]$ and $g_1, g_2 \in \mathcal{F}$ it holds*

$$\mathfrak{L}(\lambda g_1 + (1 - \lambda)g_2) \leq \lambda \mathfrak{L}(g_1) + (1 - \lambda)\mathfrak{L}(g_2).$$

*We will assume that $\mathfrak{L}$ can be written as*

$$\mathfrak{L}(f) = \frac{1}{d} \sum_{i=1}^{d} \ell(f(x_i; \theta); y_i) =: \mathcal{L}(f(\mathcal{X}); \mathcal{Y}),$$

*so that the so-called loss function $\mathcal{L}(\,\cdot\,; \mathcal{Y})\colon \mathbb{R}^{d \cdot n_L} \to \mathbb{R}$ is differentiable.*

**Remark C.1** (Function evaluation at sets and vector notation). *We want to detail the notation used in Definition C.2. For a function $f\colon \mathbb{R}^{n_0} \to \mathbb{R}^{n_L}$ and $\mathcal{X} = (x_1, \ldots, x_d) \in \mathbb{R}^{d \cdot n_0}$ we define the vector*

$$f(\mathcal{X}) := (f(x_1), \cdots, f(x_d)) \in \mathbb{R}^{d \cdot n_L}.$$

*By default, we will interpret any vector as a column vector, i.e., we identify $\mathbb{R}^n$ with $\mathbb{R}^{n \times 1}$. This is the case even when writing $x = (x_1, \ldots, x_n) \in \mathbb{R}^n$ for handier notation. Row vectors will be indicated within calculations using the transpose operator, $x^\intercal$.*

Let us first consider regular gradient descent learning in continuous time, also known as gradient flow. For this, we assume that our network function is differentiable with respect to its parameters.

**Remark C.2** (Gradient and Jacobian matrix notation). *Let $f\colon \mathbb{R}^n \to \mathbb{R}^m$. For $m = 1$, we denote the gradient by $\nabla f(x) \in \mathbb{R}^n$ for all $x \in \mathbb{R}^n$. If $m > 1$ we denote the Jacobian matrix of $f$ by $Jf\colon \mathbb{R}^n \to \mathbb{R}^{m \times n}$. Therefore, $Jf(x)$ is a $m \times n$ matrix for all $x \in \mathbb{R}^n$. We do not always want to consider the gradient or Jacobian matrix with respect to all variables. We indicate this with subscripts as follows. Let $f\colon \mathbb{R}^n \times \mathbb{R}^P \to \mathbb{R}^m$ and $g_x(\theta) = f(x; \theta)$ for fixed $x \in \mathbb{R}^n$. Then, we write $\nabla_\theta f(x; \theta) := \nabla g_x(\theta)$ and $J_\theta f(x; \theta) := Jg_x(\theta)$. In particular, for a map $f\colon \mathbb{R}^P \to \mathbb{R}$, the gradient with respect to the $j$-th variable, $1 \leq j \leq P$, is a scalar and denoted by $\partial_j f(\theta) := \nabla_{\theta_j} f(\theta)$. This is called the partial derivative of $f$ with respect to the $j$-th variable.*

With this notation, we consider the gradient flow method with learning rate $\eta > 0$ and recall the derivation of the neural tangent kernel. We move the weights in the opposite direction of the gradient of the loss function with respect to the parameters of the network evaluated at the training points:

$$\frac{\mathrm{d}}{\mathrm{d}t}\theta_t = -\eta\,\nabla_\theta \mathcal{L}(f(\mathcal{X}; \theta_t); \mathcal{Y})) = -\eta\, J_\theta f(\mathcal{X}; \theta_t)^\intercal\,\nabla_{f(\mathcal{X}; \theta_t)}\mathcal{L}(f(\mathcal{X}; \theta_t); \mathcal{Y}) \tag{S8}$$

$$= -\eta\,\frac{1}{d}\sum_{i=1}^{d} J_\theta f(x_i; \theta_t)^\intercal\,\nabla_{f(x_i; \theta_t)}\ell(f(x_i; \theta_t); y_i),$$

using the chain rule for the second equality and with $A^\intercal$ denoting the transpose of a matrix $A$. Again using the chain rule, this then implies for any $x \in \mathbb{R}^{n_0}$

$$\frac{\mathrm{d}}{\mathrm{d}t} f(x; \theta_t) = J_\theta f(x; \theta_t) \frac{\mathrm{d}}{\mathrm{d}t} \theta_t \stackrel{(S8)}{=} -\eta \, J_\theta f(x; \theta_t) J_\theta f(\mathcal{X}; \theta_t)^\intercal \nabla_{f(\mathcal{X}; \theta_t)} \mathcal{L}(f(\mathcal{X}; \theta_t); \mathcal{Y}) \qquad \text{(S9)}$$

$$= -\eta \frac{1}{d} \sum_{i=1}^{d} J_\theta f(x; \theta_t) J_\theta f(x_i; \theta_t)^\intercal \, \nabla_{f(x_i; \theta_t)} \ell(f(x_i; \theta_t); y_i). \qquad \text{(S10)}$$

We therefore define the neural tangent kernel as follows:

**Definition C.3** (Neural tangent kernel). *Let f be a network function of depth L as in Definition C.1 with parameters $\theta$, not necessarily drawn randomly. Then, we define the neural tangent kernel (NTK) as:*

$$\hat{\Theta}^{(L)} \colon \mathbb{R}^{n_0} \times \mathbb{R}^{n_0} \to \mathbb{R}^{n_L \times n_L}$$
$$(x, y) \quad \mapsto J_\theta f(x; \theta) J_\theta f(y; \theta)^\intercal.$$

*Therefore, it holds for all $x, y \in \mathbb{R}^{n_0}$ and $1 \le i, j \le n_L$*

$$\hat{\Theta}_{ij}^{(L)}(x, y) = \sum_{p=1}^{P} \partial_{\theta_p} f_i(x; \theta) \, \partial_{\theta_p} f_j(y; \theta).$$

Notice that the NTK depends on the parameters of the networks. It is therefore initialized randomly and varies over the course of the training. With notation $f_t(x) = f(x; \theta_t)$ and $\hat{\Theta}_t^{(L)} = \hat{\Theta}^{(L)}$ for parameters $\theta_t$ at training time $t$ we can now rewrite Equations (S9) and (S10) as follows:

$$\frac{\mathrm{d}}{\mathrm{d}t} f_t(x) = -\eta \hat{\Theta}_t^{(L)}(x, \mathcal{X}) \nabla_{f_t(\mathcal{X})} \mathcal{L}(f_t(\mathcal{X}); \mathcal{Y}) \qquad \text{(S11)}$$

$$= -\eta \frac{1}{d} \sum_{i=1}^{d} \hat{\Theta}_t^{(L)}(x, x_i) \, \nabla_{f_t(x_i)} \ell(f_t(x_i); y_i).$$

We are hence able to express the change of the network function during training in a kernel fashion. Later, we will consider this change of the network function in the infinite-width limit, i.e., $n_1, \ldots, n_{L-1} \to \infty$.

Before doing so, we will generalize the NTK definition in order to apply the NTK to surrogate gradient learning later. In particular, we will break the symmetry of the above definition and generalize the Jacobian matrices to quasi-Jacobian matrices by replacing the derivatives of the activation function by surrogate derivatives. Let us write the recursive formula of the Jacobian matrix of the network function given by the chain rule as $J_\theta f(x; \theta) = G(\sigma; \dot{\sigma}; x; \theta)$, where $\dot{\sigma}$ is the derivative of the activation function $\sigma$. Then, surrogate gradient learning replaces $J_\theta f(x; \theta)$ with $G(\sigma; \tilde{\sigma}; x; \theta)$ for the surrogate derivative $\tilde{\sigma}$ of the activation function $\sigma$. We call this the quasi-Jacobian matrix:

**Definition C.4** (Quasi-Jacobian matrices for neural networks). *Let L be the depth of the network, $n_k$ for $k = 0, \ldots, L$ the width of the layers, $\sigma \colon \mathbb{R} \to \mathbb{R}$ the activation function, and $\tilde{\sigma} \colon \mathbb{R} \to \mathbb{R}$ the so-called surrogate derivative of the activation function. Let f be the network function, $h^{(l)}$, $l = 1, \ldots, L-1$, the intermediate layers as in Definition C.1 and $\theta$ the network parameters. We then define the quasi-Jacobian matrix $J^{(L)}$ of f at point x recursively as follows:*

$$J^{(1)}\left(x; \theta^{(1)}\right) \in \mathbb{R}^{n_1 \times |\theta^{(1)}|} \quad \text{with} \quad J_{k\,\theta_p}^{(1)}\left(x; \theta^{(1)}\right) = \begin{cases} \delta_{ki} \frac{\sigma_w}{\sqrt{n_0}} x_j & \text{if } \theta_p = W_{ij}^{(1)} \\ \delta_{ki} \sigma_b & \text{if } \theta_p = b_i^{(1)} \end{cases} \qquad \text{(S12)}$$

$$J^{(l)}\left(x; \theta^{(1:l)}\right) \in \mathbb{R}^{n_l \times |\theta^{(1:l)}|} \quad \text{for } 2 \le l \le L \text{ with}$$

$$J_{k\,\theta_p}^{(l)}\left(x; \theta^{(1:l)}\right) = \begin{cases} \delta_{ki} \frac{\sigma_w}{\sqrt{n_{l-1}}} \sigma\left(h_j^{(l-1)}(x; \theta)\right) & \text{if } \theta_p = W_{ij}^{(l)} \\ \delta_{ki} \sigma_b & \text{if } \theta_p = b_i^{(l)} \\ \frac{\sigma_w}{\sqrt{n_{l-1}}} \sum_{j=1}^{n_{l-1}} W_{ij}^{(l)} \tilde{\sigma}\left(h_j^{(l-1)}(x; \theta)\right) J_{j\,\theta_p}^{(l-1)}\left(x; \theta^{(1:l-1)}\right) & \text{if } \theta_p \in \theta^{(l-1)}. \end{cases}$$
$$\text{(S13)}$$

**Remark C.3** (Notations for the quasi-Jacobian). *With the above definition of the quasi-Jacobian matrix of the network function $f$ with activation function $\sigma$ and surrogate derivative $\tilde{\sigma}$ we write*

$$J^{(L),\sigma,\tilde{\sigma}}(x;\theta) := J^{(L)}\left(x;\theta^{(L)}\right).$$

*It then holds*

$$J^{(L),\sigma,\dot{\sigma}}(x;\theta) = J_{\theta}f(x;\theta).$$

*For a data set $\mathcal{X}$ instead of a single point $x$, we concatenate the matrices row-wise as before, namely $J^{(L)}(\mathcal{X};\theta) \in \mathbb{R}^{d \cdot n_L \times |\theta|}$.*

**Definition C.5** (The generalized neural tangent kernel). *Let $\sigma_1, \sigma_2$ be activation functions and $\tilde{\sigma}_1, \tilde{\sigma}_2$ the surrogate derivatives respectively. Given a network depth $L$ and parameters $\theta$ we define the generalized neural tangent kernel as:*

$$\hat{I}^{(L)}\colon \mathbb{R}^{n_0} \times \mathbb{R}^{n_0} \to \mathbb{R}^{n_L \times n_L} \tag{S14}$$

$$(x,y) \mapsto J^{(L),\sigma_1,\tilde{\sigma}_1}(x;\theta)\, J^{(L),\sigma_2,\tilde{\sigma}_2}(y;\theta)^{\mathsf{T}}.$$

**Remark C.4.** *The generalized neural tangent kernel agrees with the neural tangent kernel in the case where $\sigma = \sigma_1 = \sigma_2$ and $\dot{\sigma} = \tilde{\sigma}_1 = \tilde{\sigma}_2$.*

## C.2 Notation for the infinite-width limit and review of key theorems for the NTK

In this section we will formulate all important theorems on the NTK that we will need for our later analysis using the introduced notation. Furthermore, we will discuss and remark their proofs, in particular in view of the generalizations that will be proved in Section E.1.

**Convergence of networks to Gaussian processes in the infinite-width limit.** We will consider neural networks in the limit of infinitely many hidden layer neurons. The fact that such networks converge to Gaussian processes was first mentioned by Neal [1996]. We follow and present the formulations of Matthews et al. [2018] for the general mathematical statement. First, we formalize the limit of infinitely many hidden neurons.

**Definition C.6** (Width function, as in Definition 3 of Matthews et al. [2018] with modifications). *For every layer $l = 0, \ldots, L$ and any $m \in \mathbb{N}$, the number of neurons at that layer is given by $r_l(m)$, and we call $r_l\colon \mathbb{N} \to \mathbb{N}$ the width function of layer $l$. We say that a width function $r_l$ is strictly increasing if $r_l(m) < r_l(m+1)$ for all $m \geq 1$. We set*

$$\mathcal{R}_L := \left\{ (r_l)_{l=1}^{L-1} \mid r_l \text{ is a strictly increasing width function for all } 1 \leq l < L \right\},$$

*the set of collections of strictly increasing width functions for network depth $L$.*

Every element of $\mathcal{R}_L$ provides a way to take the widths of the hidden layers to infinity by setting $n_0$ and $n_L$ to some constant, setting $n_l = r_l(m)$ for any $1 \leq l < L$ and considering $m \to \infty$. To formally define for which ways of taking the widths of hidden layers to infinity a statement holds, we can now state the set $R \subseteq \mathcal{R}_L$ such that the statement holds for widths given by any $r \in R$ as $m \to \infty$. Clearly, the claim "The statement holds for all $r \in R_1$." is stronger than "The statement holds for all $r \in R_2$." if $R_2 \subseteq R_1$. On the basis of these considerations, we define three types of infinite-width limits using the previous definition in order to structure the different types of limits in this thesis as well as in the literature.

**Definition C.7** (Types of infinite-width limits). *Consider a statement $\mathcal{S}$ of the form "Let an ANN have depth $L$ and network layer widths defined by $n_0, n_L$ and $n_l := r_l(m)$ for $1 \leq l < L$ and some $(r_l)_{l=1}^{L-1} \in \mathcal{R}_L$. Then, for fixed $n_0$ and any $n_L$, the statement $\mathcal{P}$ holds as $m \to \infty$." We also write the statement $\mathcal{S}$ as $\mathcal{S}(r)$.*

- *(i) We say that such a statement $\mathcal{S}$ holds strongly, if $\mathcal{S}(r)$ holds for any $r \in \mathcal{R}_L$. This can be interpreted as requiring that the statement holds as $\min_{1 \leq l < L}(n_l) \to \infty$. We will also write "$\mathcal{P}$ holds as $n_1, \ldots, n_{L-1} \to \infty$ strongly".*

- *(ii) We say that such a statement $\mathcal{S}$ holds for $(n_l)_{1 \leq l \leq L-1} \asymp n$, if $\mathcal{S}$ holds for all $r \in \mathcal{R}_L$ with $r_l(m) \asymp m$ for all $1 \leq l < L$. In other words $\mathcal{S}(r)$ holds for all $r \in \mathcal{R}_L$ such that $r_p(m)/r_q(m) \to \alpha_{p,q} \in (0, \infty)$ as $m \to \infty$. We will also write "$\mathcal{P}$ holds as $(n_l)_{1 \leq l < L} \asymp n$".*

*(iii) We say that such a statement $\mathcal{S}$ holds weakly, if $\mathcal{S}$ holds for at least one $r \in \mathcal{R}_L$. This can be read as requiring that the statement holds as $n_1 \to \infty, \ldots, n_{L-1} \to \infty$ sequentially. We will also write "$\mathcal{P}$ holds as $n_1, \ldots, n_{L-1} \to \infty$ weakly".*

**Theorem C.1** (Theorem 4 from Matthews et al. [2018]). *Any network function $f$ of depth $L$ defined as in Definition C.1 with continuous activation function $\sigma$ that satisfies the linear envelope property, i.e., there exist $c, m \geq 0$ with*

$$|\sigma(u)| \leq c + m|u| \quad \forall u \in \mathbb{R},$$

*converges in distribution as $n_1, \ldots, n_{L-1} \to \infty$ strongly to a multidimensional Gaussian process $(X_j)_{j=1}^{n_L}$ for any fixed countable input set $(x_i)_{i=1}^{\infty}$. It holds $X_j \overset{iid}{\sim} \mathcal{N}(0, \Sigma^{(L)})$ where the covariance function $\Sigma^{(L)}$ is recursively given by*

$$\Sigma^{(1)}(x, x') = \frac{\sigma_w^2}{n_0} \langle x, x' \rangle + \sigma_b^2,$$
$$\Sigma^{(L)}(x, x') = \sigma_w^2 \, \mathbb{E}_{g \sim \mathcal{N}(0, \Sigma^{(L-1)})}[\sigma(g(x)) \, \sigma(g(x'))] + \sigma_b^2.$$

**Remark C.5.** *First, a proof of the above theorem can be found in the paper of Matthews et al. [2018]. While it takes a lot of effort to show that the statement holds strongly in the sense of Definition C.7, the weak version of the statement can be proved via induction. This has been done by Jacot et al. [2018] and we will later adapt their proof to show a generalized version, Theorem E.3.*

*Second, in the context of analyzing the network behavior, we are interested in the finite-dimensional distributions first of all, since neural networks are trained and tested on a finite number of data points. From the convergence of the marginal distributions, we can infer the convergence to an stochastic process via the Kolmogorov extension theorem. However, this assumes the product $\sigma$-algebra, which is why Theorem C.1 assumes a fixed countable input set. Matthews et al. [2018] have discussed these formal restrictions in more detail (Chapter 2.2). If one does not want to be restricted to a countable index set, one could, for example, consider the condition by Prokhorov [1956, Theorem 2.1]. A similar approach was taken by Bracale et al. [2021], which applied the Kolmogorov-Chentsov criterion [Kallenberg, 2021, Theorem 4.23].*

*Finally, note that the theorem assumes continuity of the activation function. In the proof of Matthews et al. [2018] this is only used in order to apply the continuous mapping theorem. However, it is sufficient for the limiting process to attain possible points of discontinuity with probability zero for the continuous mapping theorem to be applicable. The theorem is thus also valid for activation functions that are continuous except at finitely many jump points, such as step-like activation functions.*

**Convergence of the NTK at initialization in the infinite-width limit.** Jacot et al. [2018] showed that the previously defined empirical NTK converges to a deterministic limit, which we will call the analytic NTK.

**Theorem C.2** (Theorem 1 from Jacot et al. [2018], slightly generalized). *For any network function of depth $L$ defined as in Definition C.1 with Lipschitz continuous activation function $\sigma$, the empirical neural tangent kernel $\hat{\Theta}^{(L)}$ converges in probability to a constant kernel $\Theta^{(L)} \otimes \mathrm{I}_{n_L}$ as $n_1, \ldots, n_{L-1} \to \infty$ weakly. For all $x, x' \in \mathbb{R}^{n_0}$ and $1 \leq i, j \leq n_L$, it holds*

$$\hat{\Theta}_{ij}^{(L)}(x, x') \xrightarrow{\mathcal{P}} \delta_{ij} \, \Theta^{(L)}(x, x'),$$

*which we also write as*

$$\hat{\Theta}^{(L)} \xrightarrow{\mathcal{P}} \Theta^{(L)} \otimes \mathrm{I}_{n_L}.$$

*We call $\Theta^{(L)}$ the analytic neural tangent kernel of the network, which is recursively given by*

$$\Theta^{(1)}(x, x') = \Sigma^{(1)}(x, x')$$
$$\Theta^{(L)}(x, x') = \Sigma^{(L)}(x, x') + \Theta^{(L-1)}(x, x') \cdot \dot{\Sigma}^{(L)}(x, x'),$$

*where $\Sigma^{(l)}$ are defined as in Theorem C.1 and we define*

$$\dot{\Sigma}^{(L)}(x, x') = \sigma_w^2 \, \mathbb{E}_{g \sim \mathcal{N}(0, \Sigma^{(L-1)})} \left[ \dot{\sigma}(g(x)) \, \dot{\sigma}(g(x')) \right].$$

Compared to Theorem 1 of Jacot et al. [2018], the statement is slightly generalized in the sense that it allows for arbitrary $\sigma_w > 0$. The arguments in the proof work the same way.

**Remark C.6** (Versions of Theorem C.2 in the literature). *A proof of this theorem for $(n_l)_{1\leq l<L} \gtrsim n$ is given by Yang [2019a] and his proof is also referenced by Lee et al. [2019]. However, the proof is given in terms of so-called tensor programs and therefore harder to follow. For the ReLU activation function, a proof for $n_1,\ldots,n_{L-1} \to \infty$ strongly is provided by Arora et al. [2019, Theorem 3.1]. We will later prove a version of this theorem for the generalized NTK.*

**Convergence of the NTK during training in the infinite-width limit.** Not only does the NTK converge to a constant kernel in the infinite-width limit, even the kernel during training, $\hat{\Theta}_t^{(L)}$, converges to this constant kernel. This was also discovered by Jacot et al. [2018].

**Theorem C.3** (Theorem 2 by Jacot et al. [2018]). *Assume any network function of depth L defined as in Definition C.1 with Lipschitz continuous activation function $\sigma$, twice differentiable with bounded second derivative, and trained with gradient flow as in Equation S10. Let $T > 0$ such that*

$$\int_0^T \|\nabla \ell(f_t(\,\cdot\,); f^*(\,\cdot\,))\|_{p_{\mathrm{emp}}} \, \mathrm{d}t = \int_0^T \sqrt{d} \left\| \nabla_{f_t(\mathcal{X})} \mathcal{L}(f_t(\mathcal{X}); f^*(\mathcal{X})) \right\|_2 \, \mathrm{d}t \in \mathcal{O}_p(1), \qquad (*)$$

*where $X \in \mathcal{O}_p(1)$ denotes that $X$ is stochastically bounded. Then, as $n_1,\ldots,n_{L-1} \to \infty$ weakly, the empirical NTK $\hat{\Theta}_t^{(L)}$ converges in probability to the analytic NTK $\Theta^{(L)} \otimes \mathrm{I}_{n_L}$ in probability uniformly for $t \in [0, T]$. We therefore write*

$$\hat{\Theta}_t^{(L)} \xrightarrow{\mathcal{P}} \Theta^{(L)} \otimes \mathrm{I}_{n_L}.$$

**Remark C.7** (Versions of Theorem C.3 in the literature). *The proof of Jacot et al. [2018] relies heavily on a function space perspective. Since this formulation tends to lack mathematical rigor, we will rely on the proof of the theorem for the case $(n_l)_{1\leq l<L} \gtrsim n$ given by Lee et al. [2019, Chapter G]. In particular, the first inequality of (S51) in Theorem G.2 of Lee et al. [2019] implies the condition ($*$). Furthermore, a different approach to proving the above statement for the case $(n_l)_{1\leq l<L} \gtrsim n$ using the Hessian matrix of the network function was taken by Liu et al. [2020, Proposition 2.3, Theorem 3.2]. A partial proof of a version of this theorem for the generalized NTK will be given later. Only an auxiliary lemma remains to be proved.*

### C.2.1 Gradient flow in the infinite-width limit

Given the results of the previous section, we can formulate an infinite width version of Equation (S10) by replacing the empirical with the analytic NTK. This allows us to analyze the learning dynamics of networks in the infinite-width limit, which yields connections to kernel methods and reproducing kernel Hilbert spaces. We then discuss how far the resulting functions and solutions in the infinite-width limit deviate from the finite width networks. This is essential to evaluate to what extend the results in the infinite-width limit can inform us about the behavior of gradient flow in the finite-width networks. First, we state the infinite-width version of Equation (S11) using Theorem C.3,

$$\frac{\mathrm{d}}{\mathrm{d}t} f_t(x) = -\eta \left( \Theta^{(L)} \otimes \mathrm{I}_{n_L} \right)(x, \mathcal{X}) \, \nabla_{f_t(\mathcal{X})} \mathcal{L}(f_t(\mathcal{X}); \mathcal{Y})$$

$$= -\eta \frac{1}{d} \sum_{i=1}^d \Theta^{(L)}(x, x_i) \cdot \mathrm{I}_{n_L} \, \nabla \ell(f_t(x_i); y_i)$$

$$= -\eta \sum_{i=1}^d \Theta^{(L)}(x, x_i) \frac{1}{d} \nabla \ell(f_t(x_i); y_i) \qquad (S15)$$

$$= -\eta \, \Theta^{(L)}(x, \mathcal{X}) \frac{1}{d} \left[ \nabla \ell(f_t(x_1); y_1), \ldots, \nabla \ell(f_t(x_d); y_d) \right]^\mathsf{T}$$

$$=: -\eta \, \Theta^{(L)}(x, \mathcal{X}) \, \nabla_{f_t(\mathcal{X})} \mathcal{L}(f_t(\mathcal{X}); \mathcal{Y}), \qquad (S16)$$

where in the last line we interpret $\nabla_{f_t(\mathcal{X})} \mathcal{L}(f_t(\mathcal{X}); \mathcal{Y}) \in \mathbb{R}^{d \times n_L}$ as a matrix of size $d \times n_L$ with entries $\left[ \nabla \mathcal{L}_{f_t(\mathcal{X})}(f_t(\mathcal{X}); \mathcal{Y}) \right]_{i\,j} = 1/d \cdot \partial_j \ell(f_t(x_i); y_i)$. Recall that $\partial_j \ell(f_t(x_i); y_i)$ is the partial derivative of $\ell(\,\cdot\,; y_i)$ with respect to its $j$-th entry, i.e., with respect to $f_{t,j}(x_i)$. Note that the last line is a row vector, which we can identify as a column vector. The fact that the NTK is now time-independent and non-random has two interesting implications:

- Equation (S16) is now an differential equation that can be solved explicitly or numerically for certain loss functions.
- According to Equation (S15), the time derivative of $f_t(x)$ can now be expressed element-wise as a linear combination of functions of the type $\Theta^{(L)}(\,\cdot\,,\tilde{x})\colon \mathbb{R}^{n_0} \to \mathbb{R}$. For an arbitrary symmetric and positive definite kernel $k(\,\cdot\,,\,\cdot\,)$, the completion of the linear span of functions of this type is called the reproducing kernel Hilbert space (RKHS) of $k$. Assuming that the solution of Equation (S15) is an element of the RKHS of $\Theta^{(L)}$, one can ask what the space looks like.

The ODE of Equation (S16) has already been considered by Jacot et al. [2018, Chapter 5] and Lee et al. [2019, Chapter 2.2], and we will follow the observations made there. To do this, we will assume the mean squared error (MSE) loss,

$$\mathcal{L}(\tilde{\mathcal{Y}};\mathcal{Y}) = \frac{1}{2}\|\tilde{\mathcal{Y}} - \mathcal{Y}\|_2^2,$$

implying $\nabla_{f_t(\mathcal{X})}\mathcal{L}(f_t(\mathcal{X});\mathcal{Y}) = f_t(\mathcal{X}) - \mathcal{Y}$, where $f_t(\mathcal{X})$ and $\mathcal{Y}$ are again interpreted as matrices of dimension $d \times n_L$. This gives us the following ODE

$$\frac{\mathrm{d}}{\mathrm{d}t}f_t(x) = -\eta\,\Theta^{(L)}(x,\mathcal{X})\,(f_t(\mathcal{X}) - \mathcal{Y})\,.$$

Now, for simplicity, we denote $\Theta(x,y) := \Theta^{(L)}(x,y)$ and $\Theta := \Theta(\mathcal{X},\mathcal{X})$. Furthermore, we consider an arbitrary set of test points $\mathcal{X}_T$. The solution of the ODE is then given by

$$f_t(\mathcal{X}_T) = \mu_t(\mathcal{X}_T) + \gamma_t(\mathcal{X}_T) \quad \text{for}$$
$$\mu_t(\mathcal{X}_T) = \Theta(\mathcal{X}_T,\mathcal{X})\Theta^{-1}\left(\mathrm{I}_d - e^{-\eta\Theta t}\right)\mathcal{Y} \quad \text{and}$$
$$\gamma_t(\mathcal{X}_T) = f_0(\mathcal{X}_T) - \Theta(\mathcal{X}_T,\mathcal{X})\Theta^{-1}\left(\mathrm{I}_d - e^{-\eta\Theta t}\right)f_0(\mathcal{X})\,.$$

Recall that by Theorem C.1, the components $f_{0,j}$ are independent and identically distributed Gaussian processes with mean zero and covariance function $\Sigma := \Sigma^{(L)}$. Hence, $\gamma_t$ has mean zero and the mean of $f_t$ is given by $\mu_t$. By looking at the components of $f_t$,

$$f_{t,j}(\mathcal{X}_T) = f_{0,j}(\mathcal{X}_T) - \Theta(\mathcal{X}_T,\mathcal{X})\Theta^{-1}\left(\mathrm{I}_d - e^{-\eta\Theta t}\right)(f_{0,j}(\mathcal{X}_T) - \mathcal{Y}_j)\,,$$

we can conclude that they are independent and identically distributed as well. One can show that the components are indeed Gaussian processes again with mean $\mu_t$. Using $\gamma_t$ we can also compute the covariance matrix for our arbitrary set of test points $\mathcal{X}_T$,

$$\begin{aligned}
\Gamma_t(\mathcal{X}_T,\mathcal{X}_T) &:= \mathbb{E}\left[\gamma_{t,j}(\mathcal{X}_T)\,\gamma_{t,j}(\mathcal{X}_T)^\intercal\right] = \mathbb{E}\left[f_{0,j}(\mathcal{X}_T)\,f_{0,j}(\mathcal{X}_T)^\intercal\right]\\
&\quad - \mathbb{E}\left[f_{0,j}(\mathcal{X}_T)f_{0,j}(\mathcal{X})^\intercal\left(\mathrm{I}_d - e^{-\eta\Theta t}\right)\Theta^{-1}\Theta(\mathcal{X},\mathcal{X}_T)\right]\\
&\quad - \mathbb{E}\left[\Theta(\mathcal{X}_T,\mathcal{X})\Theta^{-1}\left(\mathrm{I}_d - e^{-\eta\Theta t}\right)f_{0,j}(\mathcal{X})f_{0,j}(\mathcal{X}_T)^\intercal\right]\\
&\quad + \mathbb{E}\left[\Theta(\mathcal{X}_T,\mathcal{X})\Theta^{-1}\left(\mathrm{I}_d - e^{-\eta\Theta t}\right)f_{0,j}(\mathcal{X})f_{0,j}(\mathcal{X})^\intercal\left(\mathrm{I}_d - e^{-\eta\Theta t}\right)\Theta^{-1}\Theta(\mathcal{X},\mathcal{X}_T)\right]\\
&= \Sigma(\mathcal{X}_T,\mathcal{X}_T) - \Sigma(\mathcal{X}_T,\mathcal{X})\left(\mathrm{I}_d - e^{-\eta\Theta t}\right)\Theta^{-1}\Theta(\mathcal{X},\mathcal{X}_T)\\
&\quad - \Theta(\mathcal{X}_T,\mathcal{X})\Theta^{-1}\left(\mathrm{I}_d - e^{-\eta\Theta t}\right)\Sigma(\mathcal{X},\mathcal{X}_T)\\
&\quad + \Theta(\mathcal{X}_T,\mathcal{X})\Theta^{-1}\left(\mathrm{I}_d - e^{-\eta\Theta t}\right)\Sigma(\mathcal{X},\mathcal{X})\left(\mathrm{I}_d - e^{-\eta\Theta t}\right)\Theta^{-1}\Theta(\mathcal{X},\mathcal{X}_T)\,.
\end{aligned}$$

Assuming that $\Theta$ is positive definite immediately leads to pointwise convergence of the mean and covariance functions. This implies that the gradient flow solution for networks in the infinite-width limit converges to a Gaussian process as $t \to \infty$ with mean function $\mu_\infty$ and covariance function $\Gamma_\infty$ given below. This follows from the weak convergence of the finite-dimensional marginal distributions by Lévy's convergence theorem [Williams, 1991, Section 18.1]. Again, the discussions of Remark C.5 applies. We have

$$\mu_\infty(\mathcal{X}_T) = \Theta(\mathcal{X}_T,\mathcal{X})\Theta^{-1}\mathcal{Y} \quad \text{and} \tag{S17}$$
$$\Gamma_\infty(\mathcal{X}_T,\mathcal{X}_T) = \Sigma(\mathcal{X}_T,\mathcal{X}_T) - \Sigma(\mathcal{X}_T,\mathcal{X})\Theta^{-1}\Theta(\mathcal{X},\mathcal{X}_T)$$
$$- \Theta(\mathcal{X}_T,\mathcal{X})\Theta^{-1}\Sigma(\mathcal{X},\mathcal{X}_T) + \Theta(\mathcal{X}_T,\mathcal{X})\Theta^{-1}\Sigma(\mathcal{X},\mathcal{X})\Theta^{-1}\Theta(\mathcal{X},\mathcal{X}_T)\,. \tag{S18}$$

Lee et al. [2019] state that a network trained with gradient flow will indeed converge in distribution to this Gaussian process as the width goes to infinity:

**Theorem C.4** (Theorem 2.2 from Lee et al. [2019]). *Let the learning rate $\eta < \eta_{\text{critical}}$ for*

$$\eta_{\text{critical}} := 2(\lambda_{\min}(\Theta^{(L)}(\mathcal{X},\mathcal{X})) + \lambda_{\max}(\Theta^{(L)}(\mathcal{X},\mathcal{X})))$$

*with a network function $f_t$ as in Theorem C.3 with hidden layer widths $n_1 = \cdots = n_{L-1} = n$ and restricted to $x \in \mathbb{R}^{n_0}$ with $\|x\|_2 \leq 1$. If $\lambda_{\min}(\Theta^{(L)}(\mathcal{X},\mathcal{X})) > 0$, then the components of $f_t$ converge in distribution to independent, identically distributed Gaussian processes $\mathcal{N}(\mu_t, \Gamma_t)$ as $n \to \infty$ for all $t \in [0,\infty) \cup \{\infty\}$.*

Hence, the result of training a finite-width network with gradient flow for an infinite amount of time will be arbitrarily close in distribution to a Gaussian process with mean function $\mu_\infty$ and covariance function $\Gamma_\infty$, if the width is sufficiently large. Note that by Equations (S17) and (S18) the variance at the training points $\mathcal{X}$ is zero and the mean at the training points is exactly $\mathcal{Y}$.

Since we will focus on the mean, we will first sketch a trick introduced in Chapter 3 of Arora et al. [2019] to make the variance term arbitrarily small. If $f_0$ had mean variance, this would consequently also be the case for all $f_t$ and for the solution $f_\infty := \lim_{t \to \infty} f_t$. This can be achieved by multiplying $f_0$ by a small constant $\kappa > 0$ and considering the network function $g_0 = \kappa f_0$ instead. It then holds

$$\hat{\Theta}_g(x,y) = J_\theta g_0(x;\theta) J_\theta g_0(y;\theta)^\intercal = J_\theta \kappa f_0(x;\theta) J_\theta \kappa f_0(y;\theta)^\intercal = \kappa^2 \hat{\Theta}_f(x,y),$$

and thus we have $\Theta_g = \kappa^2 \Theta_f$. In the infinite-width limit, the derivative of $g_t$ is then given by

$$\frac{\mathrm{d}}{\mathrm{d}t} g_t(x) = -\eta\, \Theta_g(x,\mathcal{X})\,(g_t(\mathcal{X}) - \mathcal{Y})$$
$$= -\eta\, \kappa^2 \Theta_f(x,\mathcal{X})\,(\kappa f_t(\mathcal{X}) - \mathcal{Y}),$$

which implies as before

$$g_t(x) = g_0(x) - \Theta_g(x,\mathcal{X})\Theta_g(\mathcal{X},\mathcal{X})^{-1}\left(\mathrm{I}_d - e^{-\eta\,\Theta_g(\mathcal{X},\mathcal{X})t}\right)(g_0(\mathcal{X}) - \mathcal{Y})$$
$$= \Theta_f(x,\mathcal{X})\Theta_f(\mathcal{X},\mathcal{X})^{-1}\left(\mathrm{I}_d - e^{-\eta\kappa^2\,\Theta_f(\mathcal{X},\mathcal{X})t}\right)\mathcal{Y}$$
$$+ \kappa\left(f_0(x) - \Theta_f(x,\mathcal{X})\Theta_f(\mathcal{X},\mathcal{X})^{-1}\left(\mathrm{I}_d - e^{-\eta\kappa^2\,\Theta_f(\mathcal{X},\mathcal{X})t}\right)f_0(\mathcal{X})\right).$$

Note that the term in the second last line corresponds to the non-random mean of $f$ trained with learning rate $\eta\kappa^2$, and that the term in the last line is random, but can be made arbitrarily small using $\kappa$. We can think of this as a trade-off between learning rate and variance. This justifies why we can focus on the mean in the next section.

To sum up, we are interested in network functions in the infinite-width limit that are trained over time according to

$$\frac{\mathrm{d}}{\mathrm{d}t} f_t(x) = -\eta\, \Theta(x,\mathcal{X})\,(f_t(\mathcal{X}) - \mathcal{Y}) = \sum_{i=1}^{d} \Theta(x,x_i)\,(-\eta\,(f_t(x_i) - y_i)), \tag{S19}$$

where we change from a row vector to a column vector in the last equation. The mean of such network functions after infinite training time is given by

$$f_{\text{NTK}}(x) := \Theta(x,\mathcal{X})\Theta(\mathcal{X},\mathcal{X})^{-1}\mathcal{Y} = \mu_\infty(x). \tag{S20}$$

# D The NTK for sign activation function

The first observation to make in our attempt to apply the neural tangent kernel to networks with the sign function as activation function is that the sign function has a zero derivative almost everywhere. Thus, the derivative of the network function with respect to the network weights is zero for all weights that are not part of the last layer. The case where the weights $\theta^{(1:L-1)}$ are frozen after initialization and only $\theta^{(L)}$ is trained has already been discussed by Lee et al. [2019, Chapter 2.3.1 and Chapter D]. For a network in the infinite-width limit, this approach is equivalent to applying *Gaussian process regression*, i.e., knowing that $f \sim \mathcal{N}\left(0, \Sigma^{(L)}\right)$ for infinite width, one considers $f \mid f(\mathcal{X}) = \mathcal{Y}$. This can be seen by realizing that $\Theta^{(L)} = \Sigma^{(L)}$ if $\dot{\sigma} = 0$ almost everywhere and applying Theorem C.4.

While this is an interesting observation, and the strategy of optimizing only the last layer can also be transferred to finite width networks, we would prefer to train the whole network and not identify the derivative of the sign function with zero, since this discards all information about the jump discontinuities in our networks. An obvious alternative would be to use the distributional derivative of the sign function, which is given by $2\,\delta_0$, where $\delta_0$ denotes the delta distribution. We will see that $\dot{\Sigma}^{(L)}$ still exists when the distributional derivative is substituted into its formula. Alternatively, we can obtain the same expression by approximating the sign function with scaled error functions,

$$\text{erf}_m(z) = \text{erf}(m \cdot z) = \frac{2}{\sqrt{\pi}} \int_0^{m \cdot z} e^{-t^2} \, \text{d}t,$$

and considering the limit $m \to \infty$.

### D.1 The NTK for error activation function

Due to the previous considerations, we begin by deriving the analytic NTK for the error function. Following the notation of Lee et al. [2019], we need to find analytic expressions for the terms

$$\mathcal{T}_m(\Sigma) := \mathbb{E}_{(X,Y)\sim\mathcal{N}(0,\Sigma)}[\text{erf}_m(X)\,\text{erf}_m(Y)] \quad \text{and}$$
$$\dot{\mathcal{T}}_m(\Sigma) := \mathbb{E}_{(X,Y)\sim\mathcal{N}(0,\Sigma)}[\dot{\text{erf}}_m(X)\,\dot{\text{erf}}_m(Y)].$$

Note that by a change of variables we can alternatively consider the terms

$$\mathcal{T}(m^2 \cdot \Sigma) := \mathbb{E}_{(X,Y)\sim\mathcal{N}(0,m^2\cdot\Sigma)}[\text{erf}(X)\,\text{erf}(Y)] = \mathcal{T}_m(\Sigma) \quad \text{and}$$
$$\dot{\mathcal{T}}(m^2 \cdot \Sigma) := \mathbb{E}_{(X,Y)\sim\mathcal{N}(0,m^2\cdot\Sigma)}[\dot{\text{erf}}(X)\,\dot{\text{erf}}(Y)] = \frac{1}{m^2}\dot{\mathcal{T}}_m(\Sigma).$$

For $\Sigma' = \left(\begin{smallmatrix} x\cdot x & x\cdot y \\ x\cdot y & y\cdot y \end{smallmatrix}\right)$, $\mathcal{T}(\Sigma')$ and $\dot{\mathcal{T}}(\Sigma')$ are given in Chapter C of the supplementary material of Lee et al. [2019]. However, we cannot assume that $\Sigma'$ always has this form. While $\dot{\mathcal{T}}$ can be easily calculated, $\mathcal{T}$ is harder to deal with and a reference to Williams [1996, Chapter 3.1] is used. There, the main idea of the proof, how to evaluate a more general expression, is given without further details. We will derive analytic expressions for both terms explicitly.

We start by evaluating $\dot{\mathcal{T}}$. Note that

$$\frac{\text{d}}{\text{d}z}\text{erf}(z) = \frac{2}{\sqrt{\pi}}e^{-z^2} \quad \text{and} \quad \frac{\text{d}}{\text{d}z}\text{erf}_m(z) = \frac{2m}{\sqrt{\pi}}e^{-m^2z^2}.$$

**Lemma D.1.** *Given $U \sim \mathcal{N}(0,\Sigma)$ with invertible covariance matrix $\Sigma \in \mathbb{R}^{d\times d}$ and $x,y \in \mathbb{R}^d$, it holds*

$$\mathbb{E}[\dot{\text{erf}}(U^\intercal x)\,\dot{\text{erf}}(U^\intercal y)] = \frac{4}{\pi}\left((1 + 2x^\intercal\Sigma x)(1 + 2y^\intercal\Sigma y) - (2x^\intercal\Sigma y)^2\right)^{-1/2}. \quad \text{(S21)}$$

*In particular, given $(X,Y) \sim \mathcal{N}(0,\Sigma)$ with invertible covariance matrix $\Sigma \in \mathbb{R}^{2\times 2}$ or with $X = Y$ and singular covariance matrix $\Sigma \in \mathbb{R}^{2\times 2}$, it holds*

$$\dot{\mathcal{T}}(\Sigma) = \mathbb{E}[\dot{\text{erf}}(X)\,\dot{\text{erf}}(Y)] = \frac{4}{\pi}|\text{I}_2 + 2\cdot\Sigma|^{-1/2}, \quad \text{(S22)}$$

*where $|A|$ denotes the determinant of a matrix $A$.*

*Proof.* It holds for $U \sim \mathcal{N}(0,\Sigma)$ with covariance matrix $\Sigma \in \mathbb{R}^{d\times d}$ and $x,y \in \mathbb{R}^d$:

$$\mathbb{E}[\dot{\text{erf}}(U^\intercal x)\,\dot{\text{erf}}(U^\intercal y)] = \int_{\mathbb{R}^d} \frac{1}{(2\pi)^{d/2}|\Sigma|^{1/2}}\left(\frac{2}{\sqrt{\pi}}e^{-(u^\intercal x)^2}\right)\left(\frac{2}{\sqrt{\pi}}e^{-(u^\intercal y)^2}\right)e^{-\frac{1}{2}u^\intercal\Sigma^{-1}u}\,\text{d}u$$

$$\overset{(\star)}{=} \frac{4}{\pi}\int_{\mathbb{R}^d}\frac{1}{(2\pi)^{d/2}}\exp\left(-\frac{1}{2}v^\intercal(\text{I}_d + 2\Sigma^{1/2}xx^\intercal\Sigma^{1/2} + 2\Sigma^{1/2}yy^\intercal\Sigma^{1/2})v\right)\,\text{d}v$$

$$= \frac{4}{\pi}\left|\left(\text{I}_d + 2\Sigma^{1/2}xx^\intercal\Sigma^{1/2} + 2\Sigma^{1/2}yy^\intercal\Sigma^{1/2}\right)^{-1}\right|^{1/2}$$

$$= \frac{4}{\pi}\left|\text{I}_d + 2\Sigma^{1/2}xx^\intercal\Sigma^{1/2} + 2\Sigma^{1/2}yy^\intercal\Sigma^{1/2}\right|^{-1/2},$$

using a change of variable $\Sigma^{1/2}u = v$ for Equation $(\star)$ and using basic properties of the determinant. We can evaluate the determinant in the last line by applying the Sylvester's determinant theorem [Pozrikidis, 2014, (B.1.16)], i.e., $|\mathrm{I}_n + AB^\intercal| = |\mathrm{I}_m + B^\intercal A|$ for any matrices $A, B \in \mathbb{R}^{n \times m}$. We then define $A = B = (\sqrt{2}\Sigma^{1/2}x, \sqrt{2}\Sigma^{1/2}y) \in \mathbb{R}^{d \times 2}$, which yields

$$\left|\mathrm{I}_d + 2\Sigma^{1/2}xx^\intercal\Sigma^{1/2} + 2\Sigma^{1/2}yy^\intercal\Sigma^{1/2}\right| = |\mathrm{I}_d + AB^\intercal|$$

$$= |\mathrm{I}_2 + B^\intercal A| = \left|\begin{pmatrix} 1 + 2x^\intercal\Sigma x & 2x^\intercal\Sigma y \\ 2x^\intercal\Sigma y & 1 + 2y^\intercal\Sigma y \end{pmatrix}\right|.$$

This directly implies Equation (S21). Furthermore, Equation (S22) follows with $\Sigma \in \mathbb{R}^{2 \times 2}$ and $x = \left(\begin{smallmatrix} 1 \\ 0 \end{smallmatrix}\right), y = \left(\begin{smallmatrix} 0 \\ 1 \end{smallmatrix}\right)$ or with $x = y = \left(\begin{smallmatrix} 1 \\ 0 \end{smallmatrix}\right)$ and an arbitrary invertible covariance matrix $\Sigma'$ such that $\Sigma'_{11} = \Sigma_{11}$. $\square$

**Corollary D.1.** *Given $(X, Y) \sim \mathcal{N}(0, \Sigma)$ with invertible covariance matrix $\Sigma$, it holds*

$$\dot{\mathcal{T}}_m(\Sigma) = \mathbb{E}[\dot{\mathrm{erf}}_m(X)\,\dot{\mathrm{erf}}_m(Y)] = \frac{2}{\pi}\left|\Sigma + \mathrm{I}_2/(2m^2)\right|^{-1/2} \xrightarrow{m \to \infty} \frac{2}{\pi}|\Sigma|^{-1/2}. \qquad \text{(S23)}$$

*If $(X, Y) \sim \mathcal{N}(0, \Sigma)$ with $X = Y$ and singular covariance matrix $\Sigma \in \mathbb{R}^{2 \times 2}$, it holds*

$$\dot{\mathcal{T}}_m(\Sigma) \xrightarrow{m \to \infty} \infty.$$

*Proof.*

$$\dot{\mathcal{T}}_m(\Sigma) = m^2\,\dot{\mathcal{T}}(m^2 \cdot \Sigma) \stackrel{\text{Lemma D.1}}{=} \frac{4m^2}{\pi}|\mathrm{I}_2 + 2m^2\Sigma|^{-1/2} = \frac{2m^2}{\pi}\left(4m^4|\mathrm{I}_2/(2m^2) + \Sigma|\right)^{-1/2}$$

$$= \frac{2}{\pi}\left|\Sigma + \mathrm{I}_2/(2m^2)\right|^{-1/2} \xrightarrow{m \to \infty} \begin{cases} \frac{2}{\pi}|\Sigma|^{-1/2} & \text{if } \Sigma \text{ is invertible,} \\ \infty & \text{if } \Sigma \text{ is singular.} \end{cases}$$

$\square$

**Remark D.1.** *As mentioned at the beginning of this chapter, we can get the same result by considering the distributional derivative of the sign function, $2\delta_0$. It holds for $(X, Y) \sim \mathcal{N}(0, \Sigma)$ with invertible covariance matrix $\Sigma$ as before*

$$\mathbb{E}[\delta_0(X)\,\delta_0(Y)] = \int_{\mathbb{R}^2} \frac{1}{2\pi|\Sigma|^{1/2}} 2\delta_0(z_1)\,2\delta_0(z_2)\,e^{-\frac{1}{2}z^\intercal\Sigma^{-1}z}\,\mathrm{d}z = \frac{2}{\pi}|\Sigma|^{-1/2}.$$

*In the case of $X = Y$, the integral is no longer well-defined.*

Next, we consider $\mathcal{T}$ by solving a more general problem, which was formulated in slightly less general form by Williams [1996, Chapter 3.1].

**Lemma D.2.** *Given $U \sim \mathcal{N}(0, \Sigma)$ with invertible covariance matrix $\Sigma \in \mathbb{R}^{d \times d}$ and $x, y \in \mathbb{R}^d$, it holds*

$$V := \mathbb{E}[\mathrm{erf}(U^\intercal x)\,\mathrm{erf}(U^\intercal y)] = \frac{2}{\pi}\arcsin\left(\frac{2\,x^\intercal\Sigma y}{\sqrt{1 + 2\,x^\intercal\Sigma x}\,\sqrt{1 + 2\,y^\intercal\Sigma y}}\right).$$

*In particular, given $(X, Y) \sim \mathcal{N}(0, \Sigma)$ with invertible covariance matrix $\Sigma = \left(\begin{smallmatrix} \Sigma_1 & \Sigma_3 \\ \Sigma_3 & \Sigma_2 \end{smallmatrix}\right) \in \mathbb{R}^{2 \times 2}$ or with $X = Y$ and singular covariance matrix $\Sigma = \left(\begin{smallmatrix} \Sigma_1 & \Sigma_3 \\ \Sigma_3 & \Sigma_2 \end{smallmatrix}\right) \in \mathbb{R}^{2 \times 2}$, $\Sigma_1 = \Sigma_2 = \Sigma_3$, it holds*

$$\mathbb{E}[\mathrm{erf}(X)\,\mathrm{erf}(Y)] = \frac{2}{\pi}\arcsin\left(\frac{2\,\Sigma_3}{\sqrt{1 + 2\,\Sigma_1}\,\sqrt{1 + 2\,\Sigma_2}}\right).$$

*Proof.* We follow the proof idea given by Williams [1996, Chapter 3.1], that is, we define $V(\lambda)$, differentiate the expectation, and integrate by parts. We can then see that $\frac{\mathrm{d}}{\mathrm{d}\lambda}V(\lambda) = (1 - \gamma^2)^{-1/2}\frac{\mathrm{d}\gamma}{\mathrm{d}\lambda}$, which gives the desired $\arcsin$. So, we define

$$V(\lambda) = \mathbb{E}[\mathrm{erf}(\lambda \cdot U^\intercal x)\,\mathrm{erf}(U^\intercal y)] = \int_{\mathbb{R}^d} \frac{1}{(2\pi)^{d/2}\,|\Sigma|^{1/2}}\,\mathrm{erf}(\lambda \cdot u^\intercal x)\,\mathrm{erf}(u^\intercal y)\,e^{-\frac{1}{2}u^\intercal\Sigma^{-1}u}\,\mathrm{d}u,$$

and then differentiate with respect to $\lambda$ on both sides

$$\frac{\mathrm{d}}{\mathrm{d}\lambda} V(\lambda) = \int_{\mathbb{R}^d} \frac{1}{(2\pi)^{d/2} |\Sigma|^{1/2}} \frac{2u^\intercal x}{\sqrt{\pi}} e^{-\lambda^2 (u^\intercal x)^2} \operatorname{erf}(u^\intercal y) \, e^{-\frac{1}{2} u^\intercal \Sigma^{-1} u} \, \mathrm{d}u$$

$$= x^\intercal \int_{\mathbb{R}^d} \frac{1}{(2\pi)^{d/2} |\Sigma|^{1/2}} \frac{2u}{\sqrt{\pi}} e^{-\lambda^2 \cdot u^\intercal x x^\intercal u} \operatorname{erf}(u^\intercal y) \, e^{-\frac{1}{2} u^\intercal \Sigma^{-1} u} \, \mathrm{d}u$$

$$\overset{(\star)}{=} \frac{2 x^\intercal \Sigma^{1/2}}{\sqrt{\pi}} \int_{\mathbb{R}^d} \frac{|\Sigma|^{1/2} v}{(2\pi)^{d/2} |\Sigma|^{1/2}} \operatorname{erf}(v^\intercal \Sigma^{1/2} y) \exp\left( -\frac{1}{2} v^\intercal \left( \mathrm{I}_d + 2\lambda^2 \Sigma^{1/2} x x^\intercal \Sigma^{1/2} \right) v \right) \mathrm{d}v$$

$$= -\frac{2 x^\intercal \Sigma^{1/2}}{\sqrt{\pi} (2\pi)^{d/2}} (\mathrm{I}_d + 2\lambda^2 \Sigma^{1/2} x x^\intercal \Sigma^{1/2})^{-1}$$

$$\times \int_{\mathbb{R}^d} \operatorname{erf}(v^\intercal \Sigma^{1/2} y) \cdot (\mathrm{I}_d + 2\lambda^2 \Sigma^{1/2} x x^\intercal \Sigma^{1/2}) v \cdot \exp\left( -\frac{1}{2} v^\intercal \left( \mathrm{I}_d + 2\lambda^2 \Sigma^{1/2} x x^\intercal \Sigma^{1/2} \right) v \right) \mathrm{d}v$$

$$= \frac{2 x^\intercal \Sigma^{1/2}}{\sqrt{\pi} (2\pi)^{d/2}} (\mathrm{I}_d + 2\lambda^2 \Sigma^{1/2} x x^\intercal \Sigma^{1/2})^{-1}$$

$$\times \int_{\mathbb{R}^d} \exp\left( -\frac{1}{2} v^\intercal \left( \mathrm{I}_d + 2\lambda^2 \Sigma^{1/2} x x^\intercal \Sigma^{1/2} \right) v \right) \cdot \nabla_v \operatorname{erf}(v^\intercal \Sigma^{1/2} y) \, \mathrm{d}v, \tag{S24}$$

using a change of variables $u = \Sigma^{1/2} v$ in Equation $(\star)$ and using partial integration and Gauss' divergence theorem in the last equation. In addition, we used that

$$\nabla_v e^{-\frac{1}{2} v^\intercal \left( \mathrm{I}_d + 2\lambda^2 \Sigma^{1/2} x x^\intercal \Sigma^{1/2} \right) v} = -(\mathrm{I}_d + 2\lambda^2 \Sigma^{1/2} x x^\intercal \Sigma^{1/2}) v \, e^{-\frac{1}{2} v^\intercal \left( \mathrm{I}_d + 2\lambda^2 \Sigma^{1/2} x x^\intercal \Sigma^{1/2} \right) v}$$

and the partial integration rule for scalar functions. To be precise, for differentiable scalar functions $f$ and $g$ it holds

$$\nabla(f \cdot g) = f \cdot \nabla g + g \cdot \nabla f,$$

which then implies the partial integration rule. Furthermore, the left-hand side vanishes in our case due to Gauss' divergence theorem:

$$\left| \int_{\mathbb{R}^d} \nabla_v \left( \operatorname{erf}(v^\intercal \Sigma^{1/2} y) \cdot e^{-\frac{1}{2} v^\intercal \left( \mathrm{I}_d + 2\lambda^2 \Sigma^{1/2} x x^\intercal \Sigma^{1/2} \right) v} \right) \mathrm{d}v \right|$$

$$= \left| \lim_{R \to \infty} \int_{\mathcal{S}^{d-1}(R)} \operatorname{erf}(v^\intercal \Sigma^{1/2} y) \cdot e^{-\frac{1}{2} v^\intercal \left( \mathrm{I}_d + 2\lambda^2 \Sigma^{1/2} x x^\intercal \Sigma^{1/2} \right) v} \, \mathrm{d}v \right|$$

$$\leq \lim_{R \to \infty} \int_{\mathcal{S}^{d-1}(R)} e^{-\frac{1}{2} v^\intercal \left( \mathrm{I}_d + 2\lambda^2 \Sigma^{1/2} x x^\intercal \Sigma^{1/2} \right) v} \, \mathrm{d}v$$

$$\leq \lim_{R \to \infty} S_{d-1}(R) \cdot e^{-\frac{1}{2} R^2 \lambda_{\min}(\mathrm{I}_d + 2\lambda^2 \Sigma^{1/2} x x^\intercal \Sigma^{1/2})} = 0,$$

where $S_{d-1}(R)$ is the surface area of the sphere in $\mathbb{R}^d$ with radius $R$. Continuing with our previous calculations, we see that

$$(S24) = \frac{2 x^\intercal \Sigma^{1/2}}{\sqrt{\pi} (2\pi)^{d/2}} (\mathrm{I}_d + 2\lambda^2 \Sigma^{1/2} x x^\intercal \Sigma^{1/2})^{-1}$$

$$\times \int_{\mathbb{R}^d} e^{-\frac{1}{2} v^\intercal \left( \mathrm{I}_d + 2\lambda^2 \Sigma^{1/2} x x^\intercal \Sigma^{1/2} \right) v} \cdot \Sigma^{1/2} y \frac{2}{\sqrt{\pi}} e^{-(v^\intercal \Sigma^{1/2} y)^2} \, \mathrm{d}v$$

$$= \frac{4}{\pi} \frac{x^\intercal \Sigma^{1/2} (\mathrm{I}_d + 2\lambda^2 \Sigma^{1/2} x x^\intercal \Sigma^{1/2})^{-1} \Sigma^{1/2} y}{(2\pi)^{d/2}} \int_{\mathbb{R}^d} e^{-\frac{1}{2} v^\intercal \left( \mathrm{I}_d + 2\lambda^2 \Sigma^{1/2} x x^\intercal \Sigma^{1/2} + 2\Sigma^{1/2} y y^\intercal \Sigma^{1/2} \right) v} \, \mathrm{d}v.$$

$$\tag{S25}$$

We evaluate the expression outside of the integral in the last line by applying the Sherman-Morrison-Woodbury formula. For $A \in \mathbb{R}^{d \times d}$ and $w_1, w_2 \in \mathbb{R}^d$ it holds [Golub and Van Loan, 1996, (2.4.1)]

$$(C + w_1 w_2^\intercal)^{-1} = C^{-1} - \frac{C^{-1} w_1 w_2^\intercal C^{-1}}{1 + w_2^\intercal C^{-1} w_1}.$$

For $C = I_d$ and $w = w_1 = w_2 = \sqrt{2}\lambda\Sigma^{1/2}x$ this yields

$$(I_d + 2\lambda^2\Sigma^{1/2}xx^\mathsf{T}\Sigma^{1/2})^{-1} = (I_d + ww^\mathsf{T})^{-1} = I_d - \frac{ww^\mathsf{T}}{1 + w^\mathsf{T}w} = I_d - \frac{2\lambda^2\Sigma^{1/2}xx^\mathsf{T}\Sigma^{1/2}}{1 + 2\lambda^2x^\mathsf{T}\Sigma x}.$$

With this we see that

$$x^\mathsf{T}\Sigma^{1/2}(I_d + 2\lambda^2\Sigma^{1/2}xx^\mathsf{T}\Sigma^{1/2})^{-1}\Sigma^{1/2}y = x^\mathsf{T}\Sigma^{1/2}\left(I_d - \frac{2\lambda^2\Sigma^{1/2}xx^\mathsf{T}\Sigma^{1/2}}{1 + 2\lambda^2x^\mathsf{T}\Sigma x}\right)\Sigma^{1/2}y$$

$$= x^\mathsf{T}\Sigma y - \frac{2\lambda^2(x^\mathsf{T}\Sigma x)(x^\mathsf{T}\Sigma y)}{1 + 2\lambda^2x^\mathsf{T}\Sigma x} = x^\mathsf{T}\Sigma y\left(1 - \frac{2\lambda^2x^\mathsf{T}\Sigma x}{1 + 2\lambda^2x^\mathsf{T}\Sigma x}\right) = \frac{x^\mathsf{T}\Sigma y}{1 + 2\lambda^2x^\mathsf{T}\Sigma x}$$

Inserting this into Equation (S25), we obtain

$$(S25) = \frac{2}{\pi}\frac{2x^\mathsf{T}\Sigma y}{1 + 2\lambda^2x^\mathsf{T}\Sigma x}\int_{\mathbb{R}^d}\frac{1}{(2\pi)^{d/2}}e^{-\frac{1}{2}v^\mathsf{T}\left(I_d + 2\lambda^2\Sigma^{1/2}xx^\mathsf{T}\Sigma^{1/2} + 2\Sigma^{1/2}yy^\mathsf{T}\Sigma^{1/2}\right)v}\,\mathrm{d}v$$

$$= \frac{2}{\pi}\frac{2x^\mathsf{T}\Sigma y}{1 + 2\lambda^2x^\mathsf{T}\Sigma x}\left|\left(I_d + 2\lambda^2\Sigma^{1/2}xx^\mathsf{T}\Sigma^{1/2} + 2\Sigma^{1/2}yy^\mathsf{T}\Sigma^{1/2}\right)^{-1}\right|^{1/2}$$

$$= \frac{2}{\pi}\frac{2x^\mathsf{T}\Sigma y}{1 + 2\lambda^2x^\mathsf{T}\Sigma x}\left|I_d + 2\lambda^2\Sigma^{1/2}xx^\mathsf{T}\Sigma^{1/2} + 2\Sigma^{1/2}yy^\mathsf{T}\Sigma^{1/2}\right|^{-1/2}.$$

As in the proof of Lemma D.1, we evaluate this using Sylvester's determinant theorem. With the same notation as before, we can define $A = B = (\sqrt{2}\lambda\Sigma^{1/2}x, \sqrt{2}\Sigma^{1/2}y) \in \mathbb{R}^{d\times 2}$. This yields

$$\left|I_d + 2\lambda^2\Sigma^{1/2}xx^\mathsf{T}\Sigma^{1/2} + 2\Sigma^{1/2}yy^\mathsf{T}\Sigma^{1/2}\right| = |I_d + AB^\mathsf{T}| = |I_2 + B^\mathsf{T}A|$$

$$= \left|\begin{pmatrix}1 + 2\lambda^2x^\mathsf{T}\Sigma x & 2\lambda x^\mathsf{T}\Sigma y \\ 2\lambda x^\mathsf{T}\Sigma y & 1 + 2y^\mathsf{T}\Sigma y\end{pmatrix}\right| = (1 + 2\lambda^2x^\mathsf{T}\Sigma x)(1 + 2y^\mathsf{T}\Sigma y) - 4\lambda^2(x^\mathsf{T}\Sigma y)^2.$$

If we insert this this again, we have so far shown

$$\frac{\mathrm{d}}{\mathrm{d}\lambda}V(\lambda) = \frac{2}{\pi}\frac{2x^\mathsf{T}\Sigma y}{1 + 2\lambda^2x^\mathsf{T}\Sigma x}\left((1 + 2\lambda^2x^\mathsf{T}\Sigma x)(1 + 2y^\mathsf{T}\Sigma y) - 4\lambda^2(x^\mathsf{T}\Sigma y)^2\right)^{-1/2}. \qquad (S26)$$

We now define

$$\gamma(\lambda) := \frac{2\lambda x^\mathsf{T}\Sigma y}{\sqrt{(1 + 2\lambda^2x^\mathsf{T}\Sigma x)(1 + 2y^\mathsf{T}\Sigma y)}},$$

and claim that

$$\frac{2}{\pi}\left(1 - \gamma(\lambda)^2\right)^{-1/2}\frac{\mathrm{d}}{\mathrm{d}\lambda}\gamma(\lambda) = \frac{\mathrm{d}}{\mathrm{d}\lambda}V(\lambda). \qquad (S27)$$

We can find a solution to the claimed equation by finding a function $\tilde{V}$ that satisfies

$$\frac{\mathrm{d}}{\mathrm{d}\gamma(\lambda)}\tilde{V}(\gamma(\lambda)) = \frac{2}{\pi}\left(1 - \gamma(\lambda)^2\right)^{-1/2},$$

and by setting $V(\lambda) := \tilde{V}(\gamma(\lambda))$. This follows from the chain rule. $\tilde{V}$ is thus simply given by $\tilde{V}(\gamma(\lambda)) = \frac{2}{\pi}\arcsin(\gamma(\lambda))$. In particular, this yields

$$V = V(1) = \tilde{V}(\gamma(1)) = \frac{2}{\pi}\arcsin\left(\frac{2x^\mathsf{T}\Sigma y}{\sqrt{(1 + 2x^\mathsf{T}\Sigma x)(1 + 2y^\mathsf{T}\Sigma y)}}\right).$$

It is now left to show Equation (S27) using Equation (S26). First, see that

$$\left(1 - \gamma(\lambda)^2\right)^{-1/2} = \left(1 - \frac{(2\lambda x^\mathsf{T}\Sigma y)^2}{(1 + 2\lambda x^\mathsf{T}\Sigma x)(1 + 2y^\mathsf{T}\Sigma y)}\right)^{-1/2}$$

$$= \left(\frac{(1 + 2\lambda x^\mathsf{T}\Sigma x)(1 + 2y^\mathsf{T}\Sigma y)}{(1 + 2\lambda x^\mathsf{T}\Sigma x)(1 + 2y^\mathsf{T}\Sigma y) - 4\lambda^2(x^\mathsf{T}\Sigma y)^2}\right)^{1/2}.$$

Second, it holds

$$\frac{\mathrm{d}}{\mathrm{d}\lambda}\left[\lambda(1+2\lambda^2 x^{\mathsf{T}}\Sigma x)^{-1/2}\right] = \frac{\mathrm{d}}{\mathrm{d}\lambda}\left[(\lambda^{-2}+2x^{\mathsf{T}}\Sigma x)^{-1/2}\right] = \lambda^{-3}(\lambda^{-2}+2x_{\mathsf{T}}\Sigma x)^{-3/2}$$

$$= (1+2\lambda^2 x^{\mathsf{T}}\Sigma x)^{-3/2} = \frac{1}{\left(\sqrt{1+2\lambda^2 x^{\mathsf{T}}\Sigma x}\right)^3}.$$

With the results of both calculations, we get

$$\frac{2}{\pi}\left(1-\gamma(\lambda)^2\right)^{-1/2}\frac{\mathrm{d}}{\mathrm{d}\lambda}\gamma(\lambda) = \frac{2}{\pi}\left(1-\gamma(\lambda)^2\right)^{-1/2}\frac{2x^{\mathsf{T}}\Sigma y}{\sqrt{1+2y^{\mathsf{T}}\Sigma y}}\frac{\mathrm{d}}{\mathrm{d}\lambda}\left[\lambda(1+2\lambda^2 x^{\mathsf{T}}\Sigma x)^{-1/2}\right]$$

$$= \frac{2}{\pi}\frac{\sqrt{1+2\lambda x^{\mathsf{T}}\Sigma x}\sqrt{1+2y^{\mathsf{T}}\Sigma y}}{\sqrt{(1+2\lambda x^{\mathsf{T}}\Sigma x)(1+2y^{\mathsf{T}}\Sigma y)-4\lambda^2(x^{\mathsf{T}}\Sigma y)^2}}\frac{2x^{\mathsf{T}}\Sigma y}{\sqrt{1+2y^{\mathsf{T}}\Sigma y}}\frac{1}{\left(\sqrt{1+2\lambda^2 x^{\mathsf{T}}\Sigma x}\right)^3}$$

$$= \frac{2}{\pi}\frac{1}{\sqrt{(1+2\lambda x^{\mathsf{T}}\Sigma x)(1+2y^{\mathsf{T}}\Sigma y)-4\lambda^2(x^{\mathsf{T}}\Sigma y)^2}}\frac{2x^{\mathsf{T}}\Sigma y}{1+2\lambda^2 x^{\mathsf{T}}\Sigma x}\overset{(S26)}{=}\frac{\mathrm{d}}{\mathrm{d}\lambda}V(\lambda),$$

which yields Equation (S27) and concludes the proof. The special case $\Sigma \in \mathbb{R}^{2\times 2}$ with invertible covariance matrix or $X = Y$ and singular covariance matrix follows as in the proof of Lemma D.1. $\qquad\square$

**Corollary D.2.** *If* $(X,Y) \sim \mathcal{N}(0,\Sigma)$ *with invertible covariance matrix* $\Sigma = \left(\begin{smallmatrix}\Sigma_1 & \Sigma_3 \\ \Sigma_3 & \Sigma_2\end{smallmatrix}\right) \in \mathbb{R}^{2\times 2}$ *or with* $X = Y$ *and singular covariance matrix* $\Sigma = \left(\begin{smallmatrix}\Sigma_1 & \Sigma_3 \\ \Sigma_3 & \Sigma_2\end{smallmatrix}\right) \in \mathbb{R}^{2\times 2}$, $\Sigma_1 = \Sigma_2 = \Sigma_3$, *it holds*

$$\mathcal{T}_m(\Sigma) = \frac{2}{\pi}\arcsin\left(\frac{\Sigma_3}{\sqrt{\frac{1}{2m^2}+\Sigma_1}\sqrt{\frac{1}{2m^2}+\Sigma_2}}\right) \xrightarrow{m\to\infty} \frac{2}{\pi}\arcsin\left(\frac{\Sigma_3}{\sqrt{\Sigma_1}\sqrt{\Sigma_2}}\right). \qquad (S28)$$

*Proof.* It holds

$$\mathcal{T}_m(\Sigma) = \mathcal{T}(m^2\cdot\Sigma) \overset{\text{Lemma D.2}}{=} \frac{2}{\pi}\arcsin\left(\frac{2m^2\Sigma_3}{\sqrt{1+2m^2\Sigma_1}\sqrt{1+2m^2\Sigma_2}}\right)$$

$$= \frac{2}{\pi}\arcsin\left(\frac{\Sigma_3}{\sqrt{\frac{1}{2m^2}+\Sigma_1}\sqrt{\frac{1}{2m^2}+\Sigma_2}}\right) \xrightarrow{m\to\infty} \frac{2}{\pi}\arcsin\left(\frac{\Sigma_3}{\sqrt{\Sigma_1}\sqrt{\Sigma_2}}\right).$$

$\qquad\square$

We can now use Corollary D.1 and Corollary D.2 to evaluate $\Sigma^{(L)}$, $\dot{\Sigma}^{(L)}$ and $\Theta^{(L)}$ for activation function $\mathrm{erf}_m$, which we denote by $\Sigma_m^{(L)}$, $\dot{\Sigma}_m^{(L)}$, and $\Theta_m^{(L)}$ respectively. We then are interested in the limit $m \to \infty$. First recall that

$$\Sigma^{(1)}(x,x') = \frac{\sigma_w^2}{n_0}\langle x,x'\rangle + \sigma_b^2 \quad \text{and}$$

$$\Sigma^{(L)}(x,x') = \sigma_w^2\,\mathbb{E}_{g\sim\mathcal{N}(0,\Sigma^{(L-1)})}[\sigma(g(x))\,\sigma(g(y))] + \sigma_b^2,$$

by Theorem C.2. Therefore, $\Sigma_m^{(1)}$ is independent of $m$, and it holds for any $x,y \in \mathbb{R}^{n_0}$

$$\Sigma_m^{(1)}(x,y) = \frac{\sigma_w^2}{n_0}\langle x,y\rangle + \sigma_b^2 =: \Sigma_\infty^{(1)}(x,y).$$

Assuming that the limit $\lim_{m\to\infty} \Sigma_m^{(L)}(x,y) =: \Sigma_\infty^{(L)}(x,y)$ exists and that $\Sigma_\infty^{(L)}(x,x) \neq 0$, $\Sigma_\infty^{(L)}(y,y) \neq 0$, we can define

$$\Sigma_m^{(L+1)}(x,y) = \sigma_w^2\, \mathbb{E}_{(X,Y)\sim\mathcal{N}(0,\Sigma_{m;x,y}^{(L)})}[\mathrm{erf}_m(X)\,\mathrm{erf}_m(Y)] + \sigma_b^2 = \sigma_w^2\, \mathcal{T}_m\left(\Sigma_{m;x,y}^{(L)}\right) + \sigma_b^2$$

$$\stackrel{\text{Cor. D.2}}{=} \frac{2\sigma_w^2}{\pi}\arcsin\left(\frac{\Sigma_m^{(L)}(x,y)}{\sqrt{\frac{1}{2m^2}+\Sigma_m^{(L)}(x,x)}\sqrt{\frac{1}{2m^2}+\Sigma_m^{(L)}(y,y)}}\right) + \sigma_b^2$$

$$\xrightarrow{m\to\infty} \frac{2\sigma_w^2}{\pi}\arcsin\left(\frac{\Sigma_\infty^{(L)}(x,y)}{\sqrt{\Sigma_\infty^{(L)}(x,x)}\sqrt{\Sigma_\infty^{(L)}(y,y)}}\right) + \sigma_b^2$$

$$=: \Sigma_\infty^{(L+1)}(x,y). \tag{S29}$$

Hence, it follows via induction and from the continuity of the $\arcsin$ function that $\lim_{m\to\infty} \Sigma_m^{(L)}(x,y) =: \Sigma_\infty^{(L)}(x,y)$ is well defined. We discuss the resulting kernel $\Sigma_\infty^{(L)}$ in Remark D.3. For $\dot{\Sigma}_m^{(L+1)}$, $L \geq 1$, it holds

$$\dot{\Sigma}_m^{(L+1)}(x,y) = \sigma_w^2\, \mathbb{E}_{(X,Y)\sim\mathcal{N}(0,\Sigma_{m;x,y}^{(L)})}[\dot{\mathrm{erf}}_m(X)\,\dot{\mathrm{erf}}_m(Y)] = \sigma_w^2\, \dot{\mathcal{T}}_m\left(\Sigma_{m;x,y}^{(L)}\right)$$

$$\stackrel{\text{Cor. D.1}}{=} \frac{2\sigma_w^2}{\pi}\left|\Sigma_{m;x,y}^{(L)} + \frac{1}{2m^2}\mathrm{I}_2\right|^{-1/2}$$

$$= \frac{2\sigma_w^2}{\pi}\left(\left(\Sigma_m^{(L)}(x,x)+\frac{1}{2m^2}\right)\left(\Sigma_m^{(L)}(y,y)+\frac{1}{2m^2}\right) - \Sigma_m^{(L)}(x,y)^2\right)^{-1/2}. \tag{S30}$$

As we will see later, the limit

$$\lim_{m\to\infty}\dot{\Sigma}_m^{(L)}(x,y) =: \dot{\Sigma}_\infty^{(L)}(x,y) \tag{S31}$$

exists for $x \neq y$ apart from a few exceptions. In the case $x = y$, we obtain

$$\dot{\Sigma}_m^{(L+1)}(x,x) = \frac{2\sigma_w^2}{\pi}\left(\left(\Sigma_m^{(L)}(x,x)+\frac{1}{2m^2}\right)^2 - \Sigma_m^{(L)}(x,x)^2\right)^{-1/2}$$

$$= \frac{2\sigma_w^2}{\pi}\left(\frac{1}{m^2}\Sigma_m^{(L)}(x,x)+\frac{1}{4m^4}\right)^{-1/2} = \frac{2\sigma_w^2}{\pi}m\left(\Sigma_m^{(L)}(x,x)+\frac{1}{4m^2}\right)^{-1/2}$$

$$\sim \frac{2\sigma_w^2}{\pi}m\,\Sigma_\infty^{(L)}(x,x)^{-1/2} \xrightarrow{m\to\infty} \infty, \tag{S32}$$

where $\sim$ denotes asymptotic equality. Therefore, for $x \neq y$, the limit

$$\lim_{m\to\infty}\Theta_m^{(L)}(x,y) =: \Theta_\infty^{(L)}(x,y)$$

exists. However, due to Equation (S32), the NTK diverges for $x = y$ as $m \to \infty$. We will call a kernel with this property a *singular* kernel. We also say that a kernel with this property is *singular along the diagonal*.

**Remark D.2** (Distributional neural tangent kernel). *An alternative conceivable approach to obtain the same kernel $\Theta_\infty^{(L)}$ would have been to consider the distributional Jacobian matrix of the network function with step-like activation function. Whether or not a distributional NTK can be formulated is a question for further research. Here, we only want to point out that the corresponding formulas for the recursive definition of the analytical distributional NTK would then naturally read as follows,*

$$\Theta_\infty^{(1)}(x,x') = \Sigma_\infty^{(1)}(x,x')$$

$$\Theta_\infty^{(L)}(x,x') = \Sigma_\infty^{(L)}(x,x') + \Theta_\infty^{(L-1)}(x,x')\cdot\dot{\Sigma}_\infty^{(L)}(x,x') \quad \text{for } L \geq 2,$$

*where*

$$\Sigma_\infty^{(L)}(x,x') = \sigma_w^2\, \mathbb{E}_{g\sim\mathcal{N}(0,\Sigma_\infty^{(L-1)})}\left[\mathrm{sign}(g(x))\,\mathrm{sign}(g(y))\right] + \sigma_b^2 \quad \text{for } L \geq 2, \tag{S33}$$

$$\dot{\Sigma}_\infty^{(L)}(x,x') = \sigma_w^2\, \mathbb{E}_{g\sim\mathcal{N}(0,\Sigma_\infty^{(L-1)})}\left[2\delta_0(g(x))\,2\delta_0(g(y))\right] \quad \text{for } L \geq 2. \tag{S34}$$

*Note that Equation (S34) can be derived from Remark D.1, and that Equation (S33) is also easy to show.*

We now want to explore the implications of our findings in this section for $f_{\mathrm{NTK}}$, which we introduced at the end of Section C.2. Equation (S20) yields

$$\lim_{m \to \infty} f_{\mathrm{NTK}}^m(x) = \lim_{m \to \infty} \Theta_m(x, \mathcal{X}) \Theta_m(\mathcal{X}, \mathcal{X})^{-1} \mathcal{Y} =: f_{\mathrm{NTK}}^\infty(x).$$

Note that $\Theta_m(\mathcal{X}, \mathcal{X}) := \Theta_m^{(L)}(\mathcal{X}, \mathcal{X})$ is invertible for sufficiently large $m$, as the matrix will be dominated by the diagonal for large $m$. Using this and the fact $\Theta_m(x, \mathcal{X}) \xrightarrow{m \to \infty} \Theta_\infty(x, \mathcal{X})$ if $x \neq x_i$ for all $i = 1, \ldots, d$, we obtain for such $x$:

$$f_{\mathrm{NTK}}^m(x) = \Theta_m(x, \mathcal{X}) \Theta_m(\mathcal{X}, \mathcal{X})^{-1} \mathcal{Y} \xrightarrow{m \to \infty} 0$$

However, if $x = x_i \in \mathcal{X}$ and denoting the $i$-th basic vector by $e_i \in \mathbb{R}^d$ we get:

$$f_{\mathrm{NTK}}^m(x_i) = \Theta_m(x_i, \mathcal{X}) \Theta_m(\mathcal{X}, \mathcal{X})^{-1} \mathcal{Y} = e_i^\intercal \mathcal{Y} = y_i$$

Therefore, $f_{\mathrm{NTK}}^m$ converges pointwise to a function that is zero almost everywhere, but interpolates exactly at the data points. The singular kernel $\Theta_\infty^{(L)}$, which we can see as the analytic NTK for the sign function, is for that reason not suitable for regression. We will deal with this problem in the following section.

### D.2  From singular kernel to Nadaraya-Watson estimator

Radhakrishnan et al. [2023] considered the limit of infinite depth, $\lim_{L \to \infty} \Theta^{(L)}$, and a singular kernel emerged similar to the previous section. It was shown that the resulting estimator using this singular kernel behaves like a Nadaraya-Watson estimator when classification tasks are considered instead of regression tasks. We will adapt the ideas of Radhakrishnan et al. [2023] to show similar results. To consider a classification task, we assume that $\mathcal{Y} \in \{-1, 1\}^d$. The network is then trained with data points $(\mathcal{X}, \mathcal{Y})$ as before, but we apply the sign function at the end to obtain the final classifier. As Radhakrishnan et al. [2023], we assume that we are operating in the infinite-width limit after infinite training. So, we are interested in

$$\lim_{m \to \infty} \mathrm{sign}\left(\Theta_m(x, \mathcal{X}) \Theta_m(\mathcal{X}, \mathcal{X})^{-1} \mathcal{Y}\right). \tag{S35}$$

We will see that the resulting classifier is equal to

$$\mathrm{sign}\left(\Theta_\infty(x, \mathcal{X}) \mathcal{Y}\right) = \mathrm{sign}\left(\sum_{i=1}^d \Theta(x, x_i) y_i\right).$$

The form of this classifier is similar to a Nadaraya-Watson estimator. To be precise, for a singular kernel $k$ and data points $(\mathcal{X}, \mathcal{Y})$, the Nadaraya-Watson estimator is defined as

$$c(x) := \frac{\sum_{i=1}^d k(x, x_i) y_i}{\sum_{i=1}^d k(x, x_i)} \quad \text{for all } x \neq x_i, 1 \leq i \leq d.$$

Since $k(x, x_i) \to \infty$ as $x \to x_i$, it holds that $c(x) \to y_i$ as $x \to x_i$. Thus, the continuous extension of $c$ to $\mathbb{R}^{n_0}$ interpolates the data points.

We now begin to further evaluate $\Sigma_\infty^{(L)}$ and $\dot{\Sigma}_\infty^{(L)}$ to analyze Equation (S35):

$$\Sigma_\infty^{(1)}(x, y) = \frac{\sigma_w^2}{n_0} \langle x, y \rangle + \sigma_b^2, \tag{S36}$$

$$\Sigma_\infty^{(L)}(x, x) \overset{(S29)}{=} \frac{2\sigma_w^2}{\pi} \arcsin(1) + \sigma_b^2 = \sigma_w^2 + \sigma_b^2 \quad \text{for all} \quad L \geq 2, \tag{S37}$$

$$\Sigma_\infty^{(2)}(x, y) \overset{(S29)}{=} \frac{2\sigma_w^2}{\pi} \arcsin\left(\frac{\frac{\sigma_w^2}{n_0}\langle x, y\rangle + \sigma_b^2}{\sqrt{\frac{\sigma_w^2}{n_0}\|x\|^2 + \sigma_b^2}\sqrt{\frac{\sigma_w^2}{n_0}\|y\|^2 + \sigma_b^2}}\right) + \sigma_b^2, \quad \text{and} \tag{S38}$$

$$\Sigma_\infty^{(L+1)}(x, y) \overset{(S29)}{=} \frac{2\sigma_w^2}{\pi} \arcsin\left(\frac{\Sigma_\infty^{(L)}(x, y)}{\sqrt{\Sigma_\infty^{(L)}(x, x)}\sqrt{\Sigma_\infty^{(L)}(y, y)}}\right) + \sigma_b^2$$

$$\overset{(S37)}{=} \frac{2\sigma_w^2}{\pi} \arcsin\left(\frac{\Sigma_\infty^{(L)}(x, y)}{\sigma_w^2 + \sigma_b^2}\right) + \sigma_b^2 \quad \text{for all} \quad L \geq 2. \tag{S39}$$

Regarding the assumptions made for Equation (S29), note that that $\Sigma_\infty^{(L)}(x,x) \neq 0$ if $L \geq 2$ and $\Sigma_\infty^{(1)}(x,x) \neq 0$ if $x \neq 0$ or $\sigma_b^2 > 0$.

After Equation (S31), we mentioned exceptions to the existence of $\dot{\Sigma}_\infty^{(L)}(x,y)$ for $x \neq y$ that were postponed at that time. For $\dot{\Sigma}_m^{(2)}$ with $x \neq y$ we can see that

$$\dot{\Sigma}_m^{(2)}(x,y)$$

$$\stackrel{(S30)}{=} \frac{2\sigma_w^2}{\pi} \left( \left( \frac{\sigma_w^2}{n_0}\|x\|^2 + \sigma_b^2 + \frac{1}{2m^2} \right) \left( \frac{\sigma_w^2}{n_0}\|y\|^2 + \sigma_b^2 + \frac{1}{2m^2} \right) - \left( \frac{\sigma_w^2}{n_0}\langle x,y\rangle + \sigma_b^2 \right)^2 \right)^{-1/2}$$

$$= \frac{2\sigma_w^2}{\pi} \left( \frac{\sigma_w^4}{n_0^2} \left( \|x\|^2\|y\|^2 - \langle x,y\rangle^2 \right) + \frac{\sigma_w^2\sigma_b^2}{n_0} \left( \|x\|^2 + \|y\|^2 - 2\langle x,y\rangle \right) + \frac{1}{2m^2}A_m \right)^{-1/2}$$

$$\xrightarrow{m\to\infty} \frac{2\sigma_w^2}{\pi} \left( \frac{\sigma_w^4}{n_0^2} \left( \|x\|^2\|y\|^2 - \langle x,y\rangle^2 \right) + \frac{\sigma_w^2\sigma_b^2}{n_0}\|x-y\|^2 \right)^{-1/2} =: \dot{\Sigma}_\infty^{(2)}(x,y), \tag{S40}$$

assuming that either $\sigma_b^2 > 0$ or that $x,y$ are not parallel, i.e., $|\langle x,y\rangle| \neq \|x\|\|y\|$. The term $\frac{1}{2m^2}A_m$ is given by

$$\frac{1}{2m^2}A_m = \frac{1}{2m^2} \left( \left( \frac{\sigma_w^2}{n_0}\|x\|^2 + \sigma_b^2 \right) + \left( \frac{\sigma_w^2}{n_0}\|y\|^2 + \sigma_b^2 \right) + \frac{1}{2m^2} \right) \xrightarrow{m\to\infty} 0.$$

For $L \geq 3$ the expression simplifies:

$$\dot{\Sigma}_m^{(L)}(x,y)$$

$$\stackrel{(S30)}{=} \frac{2\sigma_w^2}{\pi} \left( \left( \Sigma_m^{(L-1)}(x,x) + \frac{1}{2m^2} \right) \left( \Sigma_m^{(L-1)}(y,y) + \frac{1}{2m^2} \right) - \Sigma_m^{(L-1)}(x,y)^2 \right)^{-1/2}$$

$$= \frac{2\sigma_w^2}{\pi} \left( \left( \Sigma_m^{(L-1)}(x,x) \cdot \Sigma_m^{(L-1)}(y,y) - \Sigma_m^{(L-1)}(x,y)^2 \right) + \frac{1}{2m^2}B_m \right)^{-1/2}$$

$$\xrightarrow{m\to\infty} \frac{2\sigma_w^2}{\pi} \left( \Sigma_\infty^{(L-1)}(x,x) \cdot \Sigma_\infty^{(L-1)}(y,y) - \Sigma_\infty^{(L-1)}(x,y)^2 \right)$$

$$\stackrel{(S37)}{=} \frac{2\sigma_w^2}{\pi} \left( (\sigma_w^2 + \sigma_b^2)^2 - \Sigma_\infty^{(L-1)}(x,y)^2 \right)^{-1/2} =: \dot{\Sigma}_\infty^{(L)}(x,y), \tag{S41}$$

with the term $\frac{1}{2m^2}B_m$ given by

$$\frac{1}{2m^2}B_m = \frac{1}{2m^2} \left( \Sigma_m^{(L-1)}(x,x) + \Sigma_m^{(L-1)}(y,y) + \frac{1}{2m^2} \right) \xrightarrow{m\to\infty} 0.$$

From Equation (S39) and Equation (S41) we see that

$$\frac{\mathrm{d}}{\mathrm{d}\left( \Sigma_\infty^{(L-1)}(x,y) \right)} \Sigma_\infty^{(L)}(x,y) = \dot{\Sigma}_\infty^{(L)}(x,y).$$

Although we did not use dual activation functions to denote the NTK as [Jacot et al., 2018, Section A.4], we still find that the property $(\hat{\sigma})' = \widehat{(\sigma')}$ applies, i.e., the derivative of the dual activation function is the dual of the derivative of the activation function. Here $\hat{\sigma}$ denotes the dual activation function of $\sigma$.

In the case $x = y$ we get

$$\dot{\Sigma}_m^{(2)}(x,x) \stackrel{(S32)}{\sim} \frac{2\sigma_w^2}{\pi}m \left( \frac{\sigma_w^2}{n_0}\|x\|^2 + \sigma_b^2 \right)^{-1/2}, \quad \text{and using (S37)} \tag{S42}$$

$$\dot{\Sigma}_m^{(L)}(x,x) \stackrel{(S32)}{\sim} \frac{2\sigma_w^2}{\pi}m \left( \sigma_w^2 + \sigma_b^2 \right)^{-1/2} \quad \text{for } L \geq 3. \tag{S43}$$

We summarize the above calculations in the following lemma:

**Lemma D.3.** *For $m, L \in \mathbb{N}$ let $\Sigma_m^{(L)}$ and $\dot{\Sigma}_m^{(L)}$ be as in Theorem C.2 for activation function $\mathrm{erf}_m$. It then holds for any $x \neq y$ with $x, y \neq 0$:*

$$\Sigma_\infty^{(1)}(x,y) = \lim_{m \to \infty} \Sigma_m^{(1)}(x,y) \overset{(S36)}{=} \frac{\sigma_w^2}{n_0}\langle x,y \rangle + \sigma_b^2,$$

$$\Sigma_\infty^{(2)}(x,y) = \lim_{m \to \infty} \Sigma_m^{(2)}(x,y) \overset{(S38)}{=} \frac{2\sigma_w^2}{\pi} \arcsin\left( \frac{\frac{\sigma_w^2}{n_0}\langle x,y \rangle + \sigma_b^2}{\sqrt{\frac{\sigma_w^2}{n_0}\|x\|^2 + \sigma_b^2}\sqrt{\frac{\sigma_w^2}{n_0}\|y\|^2 + \sigma_b^2}} \right) + \sigma_b^2,$$

$$\Sigma_\infty^{(L)}(x,y) = \lim_{m \to \infty} \Sigma_m^{(L)}(x,y) \overset{(S39)}{=} \frac{2\sigma_w^2}{\pi} \arcsin\left( \frac{\Sigma_\infty^{(L-1)}(x,y)}{\sigma_w^2 + \sigma_b^2} \right) + \sigma_b^2 \quad \textit{for all} \quad L \geq 3,$$

$$\Sigma_\infty^{(L)}(x,x) = \lim_{m \to \infty} \Sigma_m^{(L)}(x,x) \overset{(S37)}{=} \sigma_w^2 + \sigma_b^2 \quad \textit{for all} \quad L \geq 2,$$

*and, assuming that $x, y$ are not parallel or that $\sigma_b^2 > 0$,*

$$\dot{\Sigma}_\infty^{(2)}(x,y) = \lim_{m \to \infty} \dot{\Sigma}_m^{(2)}(x,y) \overset{(S40)}{=} \frac{2\sigma_w^2}{\pi}\left( \frac{\sigma_w^4}{n_0^2}\left( \|x\|^2\|y\|^2 - \langle x,y\rangle^2 \right) + \frac{\sigma_w^2\sigma_b^2}{n_0}\|x - y\|^2 \right)^{-\frac{1}{2}}$$

$$= \frac{2\sigma_w^2}{\pi}\left( \Sigma_\infty^{(1)}(x,x)\,\Sigma_\infty^{(1)}(y,y)) - \Sigma_\infty^{(1)}(x,y)^2 \right)^{-\frac{1}{2}},$$

$$\dot{\Sigma}_\infty^{(L)}(x,y) = \lim_{m \to \infty} \dot{\Sigma}_m^{(L)}(x,y) \overset{(S41)}{=} \frac{2\sigma_w^2}{\pi}\left( (\sigma_w^2 + \sigma_b^2)^2 - \Sigma_\infty^{(L-1)}(x,y)^2 \right)^{-\frac{1}{2}} \quad \textit{for all } L \geq 3,$$

$$\dot{\Sigma}_m^{(2)}(x,x) \overset{(S42)}{\sim} \frac{2\sigma_w^2}{\pi} m \left( \frac{\sigma_w^2}{n_0}\|x\|^2 + \sigma_b^2 \right)^{-\frac{1}{2}},$$

$$\dot{\Sigma}_m^{(L)}(x,x) \overset{(S43)}{\sim} \frac{2\sigma_w^2}{\pi} m \left( \sigma_w^2 + \sigma_b^2 \right)^{-\frac{1}{2}} \quad \textit{for } L \geq 3.$$

**Remark D.3** (Addendum to Remark C.5). *In Remark C.5 we discussed the topology of the space of the Gaussian processes to which ANNs with continuous activation functions converge in the infinite-width limit. The product $\sigma$-algebra restricts us to a countable input set, so it is not possible to check for properties such as continuity or even differentiability. While Theorem C.4 is stated only for continuous activation functions with linear envelope property, we will see in Theorem E.3 that the convergence also holds in the (weak) infinite-width limit even for step-like activation functions. For the sign function as activation function, the covariance function of this Gaussian process is then given by $\Sigma_\infty^{(L)}$. Since we know explicitly what this covariance function looks like, we can examine the sample-continuity of the process. Note that to get the full picture, one would still have to show functional convergence of the network to this process, as Bracale et al. [2021] have done.*

*$\Sigma_\infty^{(L)}$ is isotropic when restricted to a sphere, as will be discussed in the next section. In the case of isotropic covariance functions, $k(x,y) = k(\|x - y\|)$, the simplest way to show sample-continuity using the Kolmogorov-Chentsov criterion is to show Lipschitz continuity of $k$ at zero. This can be seen from the proof of Lemma 4.3 by Lang and Schwab [2015]. In our case, the covariance function is basically given by a composition of arcsin functions. Lipschitz continuity of $k$ at zero is therefore equivalent to Lipschitz continuity of the arcsin function at 1, which does not hold. In conclusion, it is not possible to show sample-continuity of the Gaussian process given by $\Sigma_\infty^{(L)}$ in the established way using the Kolmogorov-Chentsov criterion.*

*Clearly, the kernel can be rewritten in terms of the arccos function. Cho and Saul [2009] analyzed arc-cosine kernels in the context of deep learning in detail.*

**Corollary D.3.** *For $m, L \in \mathbb{N}$ let $\Theta_m^{(L)}$ as in Theorem C.2 for activation function $\mathrm{erf}_m$. If $x \neq y$ and either $x, y$ not parallel or $\sigma_b^2 > 0$, then the limit*

$$\Theta_\infty^{(L)}(x,y) = \lim_{m \to \infty} \Theta_m^{(L)}(x,y)$$

*exists. Furthermore, it holds, asymptotically as $m \to \infty$,*

$$\Theta_m^{(L)}(x,x) \sim \frac{2\sigma_w^2}{\pi}\left( \frac{\sigma_w^2}{n_0}\|x\|^2 + \sigma_b^2 \right)^{\frac{1}{2}}\left( \frac{2\sigma_w^2}{\pi}\left( \sigma_w^2 + \sigma_b^2 \right)^{-\frac{1}{2}} \right)^{L-2} m^{L-1} \quad \textit{for} \quad L \geq 2.$$

*Proof.* The first statement directly follows from Lemma D.3 and the definition of $\Theta_m^{(L)}$. The recursive definition can be resolved to the following formula:

$$\Theta_m^{(L)}(x,y) = \sum_{k=1}^{L} \Sigma_m^{(k)}(x,y) \cdot \prod_{l=k}^{L-1} \dot{\Sigma}_m^{(l+1)}(x,y).$$

With

$$K(z) := \frac{2\sigma_w^2}{\pi} \left( \frac{\sigma_w^2}{n_0} z + \sigma_b^2 \right)^{-\frac{1}{2}},$$

we get from Lemma D.3 that $\dot{\Sigma}_m^{(2)}(x,x) \sim K\left(\|x\|^2\right) \cdot m$ and $\dot{\Sigma}_m^{(L)}(x,x) \sim K(n_0) \cdot m$ for $L \geq 3$. This implies $\prod_{l=1}^{L-1} \dot{\Sigma}_m^{(l+1)}(x,x) \sim K(\|x\|^2) K(n_0)^{L-2} \cdot m^{L-1}$ and $\prod_{l=k}^{L-1} \dot{\Sigma}_m^{(l+1)}(x,x) \sim K(n_0)^{L-k} \cdot m^{L-k}$ for any $k \geq 2$. For $L \geq 2$, we get that

$$\Theta_m^{(L)}(x,x) \sim \Sigma_\infty^{(1)}(x,y) K\left(\|x\|^2\right) K(n_0)^{L-2} \cdot m^{L-1} + \sum_{k=2}^{L} \Sigma_\infty^{k}(x,x) \cdot K(n_0)^{L-k} \cdot m^{L-k}$$

$$\sim \left( \frac{\sigma_w^2}{n_0} \|x\|^2 + \sigma_b^2 \right) \frac{2\sigma_w^2}{\pi} \left( \frac{\sigma_w^2}{n_0} \|x\|^2 + \sigma_b^2 \right)^{-\frac{1}{2}} K(n_0)^{L-2} m^{L-1}$$

$$= \frac{2\sigma_w^2}{\pi} \left( \frac{\sigma_w^2}{n_0} \|x\|^2 + \sigma_b^2 \right)^{\frac{1}{2}} \left( \frac{2\sigma_w^2}{\pi} \left( \sigma_w^2 + \sigma_b^2 \right)^{-\frac{1}{2}} \right)^{L-1} m^{L-2}.$$

$\square$

**Theorem D.4** (Inspired by Lemma 5 of Radhakrishnan et al. [2023]). *Let $\sigma_b^2 > 0$ or let all $x_i \in \mathbb{R}^{n_0}$ be pairwise non-parallel. Let $L \geq 2$ and $x_i \in \mathcal{S}_R^{n_0-1}$ for all $i = 1, \dots, d$, where $\mathcal{S}_R^{n_0-1} \subseteq \mathbb{R}^{n_0}$ is the sphere of radius $R$. Then, with*

$$c^{(L)}(x) := \lim_{m \to \infty} \text{sign}\left( \Theta_m^{(L)}(x, \mathcal{X}) \Theta_m^{(L)}(\mathcal{X}, \mathcal{X})^{-1} \mathcal{Y} \right),$$

*and assuming that $\Theta_\infty^{(L)}(x, \mathcal{X})\mathcal{Y} \neq 0$ for almost all $x \in \mathcal{S}_R^{n_0-1}$, it holds*

$$c^{(L)}(x) = \text{sign}\left( \Theta_\infty^{(L)}(x, \mathcal{X})\mathcal{Y} \right) \quad \textit{a.e. on } \mathcal{S}_R^{n_0-1}.$$

*Proof.* First note that almost all $x \in \mathcal{S}_R^{n_0-1}$ are not parallel to any $x_i \in \mathcal{X}$. We denote $\Theta_m^{(L)}(\mathcal{X}, \mathcal{X}) =: \Theta_m$, $m \in \mathbb{N} \cup \{\infty\}$, for convenience. Let $a_m > 0$ be a positive constant that we will choose later. It then holds for almost all $x \in \mathcal{S}_R^{n_0-1}$

$$\text{sign}\left( \Theta_m^{(L)}(x, \mathcal{X}) \Theta_m^{-1} \mathcal{Y} \right) = \text{sign}\left( a_m \Theta_m^{(L)}(x, \mathcal{X}) \Theta_m^{-1} \mathcal{Y} \right)$$

$$= \text{sign}\left( \underbrace{\left[ a_m \Theta_m^{(L)}(x, \mathcal{X}) \Theta_m^{-1} \mathcal{Y} - \Theta_m^{(L)}(x, \mathcal{X})\mathcal{Y} \right]}_{=:A_m} + \underbrace{\left[ \Theta_m^{(L)}(x, \mathcal{X})\mathcal{Y} - \Theta_\infty^{(L)}(x, \mathcal{X})\mathcal{Y} \right]}_{=:B_m} \right.$$

$$\left. + \Theta_\infty^{(L)}(x, \mathcal{X})\mathcal{Y} \right).$$

We now show that $A_m$ and $B_m$ go to zero as $m \to \infty$ for a suitable choice of $a_m$. First, note that

$$|B_m| \leq \left\| \Theta_m^{(L)}(x, \mathcal{X}) - \Theta_\infty^{(L)}(x, \mathcal{X}) \right\|_2 \|\mathcal{Y}\|_2 \xrightarrow{m \to \infty} 0,$$

since $\|\mathcal{Y}\|_2 < \infty$ and by Corollary D.3 it holds that $\Theta_m^{(L)}(x, x_i) \to \Theta_\infty^{(L)}(x, x_i)$ for all $x_i \in \mathcal{X}$ as $m \to \infty$. Second, we have

$$|A_m| \leq \left\| a_m \Theta_m^{(L)}(x, \mathcal{X}) \Theta_m^{-1} - \Theta_m^{(L)}(x, \mathcal{X}) \right\|_2 \|\mathcal{Y}\|_2 \leq \left\| \Theta_m^{(L)}(x, \mathcal{X}) \right\|_2 \left\| a_m \Theta_m^{-1} - \text{I}_d \right\|_2 \|\mathcal{Y}\|_2.$$

(S44)

Again by Corollary D.3 it holds that $\|\Theta_m^{(L)}(x,\mathcal{X})\|_2 \to \|\Theta_\infty^{(L)}(x,\mathcal{X})\|_2$ as $m \to \infty$. Furthermore, for all $1 \le i \le d$ we have $\Theta_m^{(L)}(x_i, x_i) \sim C(R) \cdot m^{L-1}$ for some constant $C(R)$ which depends on $R = \|x_i\|$. Since it holds $\Theta_m^{(L)}(x_j, x_i) \to \Theta_\infty^{(L)}(x_j, x_i)$ as $m \to \infty$ for any $i \neq j$, we choose $a_m = C(R) \cdot m^{L-1}$ and conclude

$$a_m^{-1}\Theta_m^{(L)}(x_i, x_j) \xrightarrow{m\to\infty} \begin{cases} 1 & \text{if } i = j \\ 0 & \text{if } i \neq j. \end{cases}$$

This implies $a_m^{-1}\Theta_m \to \mathrm{I}_d$ as $m \to \infty$. In particular, $\{\mathrm{I}_d\} \cup \{a_m^{-1}\Theta_m \mid m \in \mathbb{N}\}$ is a compact set with respect to $\|\cdot\|_2$. Since $D \mapsto D^{-1}$ is a continuous function, $\{\mathrm{I}_d\} \cup \{a_m\Theta_m^{-1} \mid m \in \mathbb{N}\}$ is bounded. We get

$$\left\|a_m\Theta_m^{-1} - \mathrm{I}_d\right\|_2 \le \left\|a_m\Theta^{-1}\right\|_2 \left\|\mathrm{I}_d - a_m^{-1}\Theta_m\right\|_2 \xrightarrow{m\to\infty} 0,$$

and thus

$$|A_m| \overset{(S44)}{\le} \left\|\Theta_m^{(L)}(x,\mathcal{X})\right\|_2 \left\|a_m\Theta_m^{-1} - \mathrm{I}_d\right\|_2 \|\mathcal{Y}\|_2 \xrightarrow{m\to\infty} 0.$$

Recall that $\Theta_\infty^{(L)}(x,\mathcal{X})\mathcal{Y} \neq 0$, which now concludes the proof:

$$\lim_{m\to\infty} \mathrm{sign}\left(\Theta_m^{(L)}(x,\mathcal{X})\Theta_m^{-1}\mathcal{Y}\right) = \lim_{m\to\infty} \mathrm{sign}\left(A_m + B_m + \Theta_\infty^{(L)}(x,\mathcal{X})\mathcal{Y}\right)$$
$$= \mathrm{sign}\left(\Theta_\infty^{(L)}(x,\mathcal{X})\mathcal{Y}\right).$$

$\square$

**Remark D.4.** *One can generalize the above theorem by dropping the restriction to $\mathcal{S}_R^{n_0-1}$. By Lemma D.3 it holds $\Theta_m^{(L)}(x_i, x_i) \sim \sqrt{\frac{\sigma_w^2}{n_0}\|x_i\|^2 + \sigma_b^2} \cdot C \cdot m^{L-1}$. For some constant $C$ independent of $L$ and $\|x_i\|$. If we now define $a_m := C \cdot m^{L-1}$, we get*

$$a_m^{-1}\Theta_m^{(L)}(\mathcal{X},\mathcal{X}) \xrightarrow{m\to\infty} \mathrm{diag}\left\{\left(\sqrt{\sigma_w^2\|x_i\|^2/n_0 + \sigma_b^2}\right)_{1 \le i \le d}\right\} =: D,$$

*where $\mathrm{diag}\{v\}$ of a vector $v$ is a square matrix with diagonal $v$. As in the proof above, and using the continuity of the inverse map $D \mapsto D^{-1}$, this implies*

$$\lim_{m\to\infty} \mathrm{sign}\left(\Theta_m^{(L)}(x,\mathcal{X})\Theta_m^{-1}\mathcal{Y}\right) = \mathrm{sign}\left(\Theta_\infty^{(L)}(x,\mathcal{X})D^{-1}\mathcal{Y}\right) = \mathrm{sign}\left(\tilde{\Theta}_\infty^{(L)}(x,\mathcal{X})\mathcal{Y}\right),$$

*if we define*

$$\tilde{\Theta}_\infty^{(L)}(x,y) := \left(\sigma_w^2\|x\|^2/n_0 + \sigma_b^2\right)^{-1/2} \left(\sigma_w^2\|y\|^2/n_0 + \sigma_b^2\right)^{-1/2} \Theta_\infty^{(L)}(x,y).$$

*In conclusion, dropping the restriction leads to a similar form for our classifier, but with a modified kernel.*

Theorem D.4 shows that the classifier

$$\lim_{m\to\infty} \mathrm{sign}\left(\Theta_m^{(L)}(x,\mathcal{X})\Theta_m^{-1}\mathcal{Y}\right) = \lim_{m\to\infty} \mathrm{sign}\left(f_{\mathrm{NTK}}^m(x)\right)$$

has an easily interpretable form that is close to the Nadaraya-Watson estimator with singular kernel $\Theta_\infty^{(L)}$. This is despite the fact that the pointwise limit of $f_{\mathrm{NTK}}^m$ is trivial, i.e., regression is no longer possible

### D.3 Checking for Bayes optimality

In addition to the ideas used in the above section, Radhakrishnan et al. [2023] proved that the resulting estimator is *Bayes optimal* under certain conditions. This property is also referred to as *consistency*. Let the probability distribution $\mathbb{P}_{\mathrm{data}}$ on our data space be given by a random variable $(X, Y) \in \mathbb{R}^{n_0} \times \{-1, 1\}$ and let

$$c(x) = \arg\max_{\tilde{y}\in\{-1,1\}} \mathbb{P}(Y = \tilde{y} \mid X = x)$$

be the *Bayes classifier* with respect to this distribution. Denote $\mathcal{X}_d = (x_1, \ldots, x_d)$ and $\mathcal{Y}_d = (y_1, \ldots, y_d)$ for $(x_i, y_i)$ drawn independently from $\mathbb{P}_{\mathrm{data}}$. We then define Bayes optimality as follows:

**Definition D.1** (Bayes optimality). *Let $c_d(\,\cdot\,) = c_d(\,\cdot\,; \mathcal{X}_d, \mathcal{Y})$ be classifiers for $d \in \mathbb{N}$ and estimators of $c(\,\cdot\,)$. We then say that $(c_d)_{d \in \mathbb{N}}$ is Bayes optimal, if it is a consistent estimator of $c$, i.e., for all $\varepsilon > 0$ and $X$-almost all $x$*

$$\lim_{d \to \infty} \mathbb{P}\big[|c_d(x) - c(x)| > \varepsilon\big] = 0.$$

First, we summarize the results of Radhakrishnan et al. [2023] and consider $\Theta^{(L)}$. If the singular limiting kernel for $L \to \infty$ behaves like a singular kernel of the form $k(x,y) = \|x - y\|^{-\alpha}$, then the classifier for $L \to \infty$ will be of the form

$$\text{sign}\left(\frac{\sum_{i=1}^{d} y_i / \|x - x_i\|^{\alpha}}{\sum_{i=1}^{d} 1 / \|x - x_i\|^{\alpha}}\right),$$

assuming $\sum_{i=1}^{d} 1 / \|x - x_i\|^{\alpha} > 0$. This classifier satisfies Bayes optimality for $\alpha = n_0$ by Devroye et al. [1998]. Radhakrishnan et al. [2023] generalized the results of Devroye et al. [1998], expressed $\alpha$ in terms of the activation function and its derivative, and chose them in such a way as to achieve $\alpha = n_0$. Going back to our setup, we want to see if we can write

$$\lim_{m \to \infty} \Theta_m^{(L)}(x, y) = \frac{R(\|x - y\|)}{\|x - y\|^{\alpha}},$$

for some constant $\alpha$ and for some function $R \colon \mathbb{R}_+ \to \mathbb{R}$ bounded away from zero as $\|x - y\| \to 0$. We start by proving that $\Theta_m^{(L)}(x, y) = G(\|x - y\|)$ for some function $G$. Recall that we have to restrict ourselves to a sphere by Theorem D.4. For simplicity, we restrict ourselves to the unit sphere $\mathcal{S}^{n_0 - 1}$. Then, it holds $\langle x, x \rangle = 1$ for all $x \in \mathcal{S}^{n_0 - 1}$, which gives us

$$\|x - y\|^2 = \langle x - y, x - y \rangle = \langle x, x \rangle - 2\langle x, y \rangle + \langle y, y \rangle = 2(1 - \langle x, y \rangle).$$

In the following, we will substitute $z = \langle x, y \rangle$. Taking $\|x - y\| \to 0$ is then equivalent to taking $z \to 1$. We can conclude from Lemma D.3 that we can indeed write $\Sigma_\infty^{(L)}(x, y) = \Sigma_\infty^{(L)}(z)$, $\dot{\Sigma}_\infty^{(L)}(x, y) = \dot{\Sigma}_\infty^{(L)}(z)$, and hence $\Theta_\infty^{(L)}(x, y) = \Theta_\infty^{(L)}(z)$ for all $L \geq 1$. Note that $\Sigma_\infty^{(L)}(1) = \sigma_w^2 + \sigma_b^2$ for $L \geq 2$ and $\Sigma_\infty^{(L)}(1) = \sigma_w^2/n_0 + \sigma_b^2$. Next, we consider $\dot{\Sigma}_\infty^{(2)}(z)$ for $z \to 1$ using Lemma D.3

$$\dot{\Sigma}_\infty^{(2)}(z) = \frac{2\sigma_w^2}{\pi}\left(\frac{\sigma_w^4}{n_0^2}(1 - z^2) + \frac{\sigma_w^2 \sigma_b^2}{n_0} \cdot 2(1 - z)\right)^{-\frac{1}{2}}$$

$$= \frac{2\sigma_w^2}{\pi}\left(\frac{\sigma_w^4}{n_0^2}(1 - z)(1 + z) + \frac{2\sigma_w^2 \sigma_b^2}{n_0}(1 - z)\right)^{-\frac{1}{2}}$$

$$\overset{z \to 1}{\sim} \frac{2\sigma_w^2}{\pi}\left(\frac{2\sigma_w^4}{n_0^2}(1 - z) + \frac{2\sigma_w^2 \sigma_b^2}{n_0}(1 - z)\right)^{-\frac{1}{2}} = \frac{2\sigma_w^2}{\pi}\left(\frac{2\sigma_w^2}{n_0}\left(\frac{\sigma_w^2}{n_0} + \sigma_b^2\right)\right)^{-\frac{1}{2}}(1 - z)^{-\frac{1}{2}}$$

$$= \frac{n_0}{\pi}\left(\frac{2\sigma_w^2}{n_0}\right)^{\frac{1}{2}}\left(\frac{\sigma_w^2}{n_0} + \sigma_b^2\right)^{-\frac{1}{2}}(1 - z)^{-\frac{1}{2}}. \tag{S45}$$

For $L \geq 3$, we can observe that

$$\dot{\Sigma}_\infty^{(L)}(z) = \frac{2\sigma_w^2}{\pi}\left(\left(\sigma_w^2 + \sigma_b^2\right)^2 - \Sigma_\infty^{(L-1)}(z)^2\right)^{-\frac{1}{2}}$$

$$= \frac{2\sigma_w^2}{\pi}\left(\left(\sigma_w^2 + \sigma_b^2 - \Sigma_\infty^{(L-1)}(z)\right)\left(\sigma_w^2 + \sigma_b^2 + \Sigma^{(L-1)}\infty(z)\right)\right)^{-\frac{1}{2}}$$

$$\overset{z \to 1}{\sim} \frac{2\sigma_w^2}{\pi}\left(2\left(\sigma_w^2 + \sigma_b^2\right)\right)^{-\frac{1}{2}}\left(\left(\sigma_w^2 + \sigma_b^2\right) - \Sigma_\infty^{(L-1)}(z)\right)^{-\frac{1}{2}}. \tag{S46}$$

We will now prove a lemma that allows us to analyze the behavior of $\dot{\Sigma}_\infty^{(L)}(z)$ as $z \to 1$.

**Lemma D.5.** *It holds:*

$$\lim_{z \to 1} \frac{(1 - z)^{1/2}}{1 - \frac{2}{\pi}\arcsin(z)} = \frac{\pi}{2\sqrt{2}}.$$

*Proof.* Since the numerator and the denominator both go to zero as $z \to 1$, we apply l'Hôpital's rule and differentiate both. This yields

$$\frac{\mathrm{d}}{\mathrm{d}z}\left[(1-z)^{\frac{1}{2}}\right] = -\frac{1}{2}(1-z)^{-\frac{1}{2}} \quad \text{and} \quad \frac{\mathrm{d}}{\mathrm{d}z}\left[1 - \frac{2}{\pi}\arcsin(z)\right] = -\frac{2}{\pi}\left(1-z^2\right)^{-\frac{1}{2}}.$$

Thus,

$$\lim_{z\to 1}\frac{(1-z)^{1/2}}{1 - \frac{2}{\pi}\arcsin(z)} = \lim_{z\to 1}\frac{\frac{1}{2}\sqrt{1-z}\sqrt{1+z}}{\frac{2}{\pi}\sqrt{1-z}} = \frac{\pi}{4}\sqrt{2} = \frac{\pi}{2\sqrt{2}},$$

concluding the proof. $\qquad\square$

For $L = 3$ it now holds

$$\dot\Sigma_\infty^{(3)}(z) \overset{(S46)}{\sim} \frac{2\sigma_w^2}{\pi}\left(2\left(\sigma_w^2 + \sigma_b^2\right)\right)^{-\frac{1}{2}}\left(\left(\sigma_w^2 + \sigma_b^2\right) - \Sigma_\infty^{(2)}(z)\right)^{-\frac{1}{2}}$$

$$= \frac{2\sigma_w^2}{\pi}\left(2\left(\sigma_w^2 + \sigma_b^2\right)\right)^{-\frac{1}{2}}\sigma_w^{-1}\left(1 - \frac{2}{\pi}\arcsin\left(\frac{\sigma_w^2 z/n_0 + \sigma_b^2}{\sigma_w^2/n_0 + \sigma_b^2}\right)\right)^{-\frac{1}{2}}$$

$$\overset{(\star)}{\sim} \frac{2\sigma_w^2}{\pi}\left(2\left(\sigma_w^2 + \sigma_b^2\right)\right)^{-\frac{1}{2}}\sigma_w^{-1}\sqrt{\frac{\pi}{2\sqrt{2}}}\left(1 - \frac{\sigma_w^2 z/n_0 + \sigma_b^2}{\sigma_w^2/n_0 + \sigma_b^2}\right)^{-\frac{1}{4}}$$

$$= \frac{2\sigma_w^2}{\pi}\left(2\left(\sigma_w^2 + \sigma_b^2\right)\right)^{-\frac{1}{2}}\sigma_w^{-1}\sqrt{\frac{\pi}{2\sqrt{2}}}\left(\frac{\sigma_w^2/n_0 \cdot (1-z)}{\sigma_w^2/n_0 + \sigma_b^2}\right)^{-\frac{1}{4}}$$

$$= \sqrt{\frac{2\sigma_w^2}{\pi}}\left(2\sqrt{2}\left(\sigma_w^2 + \sigma_b^2\right)\right)^{-\frac{1}{2}}\left(\frac{\sigma_w^2/n_0}{\sigma_w^2/n_0 + \sigma_b^2}\right)^{-\frac{1}{4}}(1-z)^{-\frac{1}{4}},$$

where we used Lemma D.5 at $(\star)$. For $L \geq 4$ it holds

$$\dot\Sigma_\infty^{(L)}(z) \overset{(S46)}{\sim} \frac{2\sigma_w^2}{\pi}\left(2\left(\sigma_w^2 + \sigma_b^2\right)\right)^{-\frac{1}{2}}\left(\left(\sigma_w^2 + \sigma_b^2\right) - \Sigma_\infty^{(L-1)}(z)\right)^{-\frac{1}{2}}$$

$$\overset{L-1\geq 3}{=} \frac{2\sigma_w^2}{\pi}\left(2\left(\sigma_w^2 + \sigma_b^2\right)\right)^{-\frac{1}{2}}\sigma_w^{-1}\left(1 - \frac{2}{\pi}\arcsin\left(\frac{\Sigma_\infty^{(L-2)}(z)}{\sigma_w^2 + \sigma_b^2}\right)\right)^{-\frac{1}{2}}$$

$$\overset{(\star)}{\sim} \frac{2\sigma_w^2}{\pi}\left(2\left(\sigma_w^2 + \sigma_b^2\right)\right)^{-\frac{1}{2}}\sigma_w^{-1}\sqrt{\frac{\pi}{2\sqrt{2}}}\left(1 - \frac{\Sigma_\infty^{(L-2)}(z)}{\sigma_w^2 + \sigma_b^2}\right)^{-\frac{1}{4}}$$

$$= \frac{2\sigma_w^2}{\pi}\left(2\left(\sigma_w^2 + \sigma_b^2\right)\right)^{-\frac{1}{2}}\sigma_w^{-1}\sqrt{\frac{\pi}{2\sqrt{2}}}\left(\sigma_w^2 + \sigma_b^2\right)^{\frac{1}{4}}\left(\sigma_w^2 + \sigma_b^2 - \Sigma_\infty^{(L-2)}(z)\right)^{-\frac{1}{4}}$$

$$\overset{(S46)}{\sim} \sqrt{\frac{2\sigma_w^2}{\pi}}\left(2\left(\sigma_w^2 + \sigma_b^2\right)\right)^{-\frac{1}{4}}\sigma_w^{-1}\sqrt{\frac{\pi}{2\sqrt{2}}}\left(\sigma_w^2 + \sigma_b^2\right)^{\frac{1}{4}}\sqrt{\dot\Sigma_\infty^{(L-1)}(z)}$$

$$= \sqrt{\dot\Sigma_\infty^{(L-1)}(z)/2},$$

again using Lemma D.5 at $(\star)$. This implies for arbitrary $L \geq 2$

$$\dot\Sigma_\infty^{(L)}(z) \sim K(L) \cdot (1-z)^{-1/2^{L-1}},$$

for some constant $K(L)$ depending on $L$. Recall the formula for the NTK, that was used in the proof of Corollary D.3

$$\Theta_\infty^{(L)}(z) = \sum_{k=1}^{L}\Sigma_\infty^{(k)}(z) \cdot \prod_{l=k}^{L-1}\dot\Sigma_\infty^{(l+1)}(z) \sim \sum_{k=1}^{L}\Sigma_\infty^{(k)}(z) \cdot \prod_{l=k}^{L-1}K(l+1) \cdot (1-z)^{-1/2^l}$$

$$\sim \Sigma_\infty^{(1)}(1)\prod_{l=1}^{L-1}K(l+1) \cdot (1-z)^{-1/2^l} = K(L) \cdot (1-z)^{-(1-1/2^{L-1})}$$

$$= K'(L) \cdot \|x - y\|^{-(2-1/2^{L-2})} =: K'(L) \cdot \|x - y\|^{-\alpha(L)},$$

for some constants $K(L), K'(L)$. It is therefore to possible find a function $R$ that is bounded away from zero near $\|x - y\| \to 0$, such that

$$\Theta_\infty^{(L)}(x, y) = K'(L) \cdot \frac{R(\|x - y\|)}{\|x - y\|^{\alpha(L)}}.$$

However, we have that $\alpha(1) = 1$ and $\alpha(L) \uparrow 2$ as $L \to \infty$. So it is only possible to choose $L$ such that $\alpha(L) = n_0$, if $n_0 = 1$. But this is a trivial case, since we have restricted ourselves to the unit sphere. In conclusion, we cannot prove that $c^{(L)}$ is a Bayes optimal classifier for any choice of $L$. Chapter 4 of Devroye et al. [1998] suggests that, in fact, the estimator will not be universally consistent.

## E   The NTK for surrogate gradient learning

In this chapter we explore *surrogate gradient learning*, introduced in Section 1, by connecting it to the NTK. Recall that the standard gradient flow dynamics of the parameters are given by

$$\frac{\mathrm{d}}{\mathrm{d}t}\theta_t = -\eta \, \nabla_\theta \mathcal{L}(f(\mathcal{X}; \theta_t); \mathcal{Y}) = -\eta \, J_\theta f(\mathcal{X}; \theta_t)^\intercal \, \nabla_{f(\mathcal{X};\theta_t)} \mathcal{L}(f(\mathcal{X}; \theta_t); \mathcal{Y}). \tag{S8}$$

As mentioned several times before, the Jacobian matrix $J_\theta$ will vanish for all parameters except those in the last layer if we consider the sign activation function due to its zero derivative. The idea of surrogate gradient learning is to circumvent the zero derivative of the sign function by replacing the derivative with a surrogate derivative [Neftci et al., 2019]. We can replace the derivative in two main ways:

- The activation function in the full network can be replaced by a differentiable surrogate activation function. In particular, we obtain a non-vanishing surrogate derivative of the activation function and thus non-vanishing network gradients. Let $g$ denote the network with surrogate activation function. Therefore, we can train the weights according to

$$\frac{\mathrm{d}}{\mathrm{d}t}\theta_t = -\eta \, J_\theta g(\mathcal{X}; \theta_t)^\intercal \, \nabla_{g(\mathcal{X};\theta_t)} \mathcal{L}(g(\mathcal{X}; \theta_t); \mathcal{Y}),$$

  or consider the loss with respect to $f$ and train according to

$$\frac{\mathrm{d}}{\mathrm{d}t}\theta_t = -\eta \, J_\theta g(\mathcal{X}; \theta_t)^\intercal \, \nabla_{f(\mathcal{X};\theta_t)} \mathcal{L}(f(\mathcal{X}; \theta_t); \mathcal{Y}). \tag{S47}$$

- Instead of replacing the activation function, we can only replace the derivative of the activation function $\dot\sigma$ with a surrogate derivative $\tilde\sigma$ in Equation (S8). Let $J_{\sigma,\tilde\sigma}$ be the quasi-Jacobian matrix as in Definition C.4 with activation function $\sigma$ and surrogate derivative $\tilde\sigma$. Then the training is given by

$$\frac{\mathrm{d}}{\mathrm{d}t}\theta_t \quad = -\eta \, J^{\sigma,\tilde\sigma}(\mathcal{X}; \theta_t)^\intercal \, \nabla_{f(\mathcal{X};\theta_t)} \mathcal{L}(f(\mathcal{X}; \theta_t); \mathcal{Y})$$

$$\implies \frac{\mathrm{d}}{\mathrm{d}t} f(x; \theta_t) = -\eta \, J_\theta f(x; \theta_t) J^{\sigma,\tilde\sigma}(\mathcal{X}; \theta_t)^\intercal \, \nabla_{f(\mathcal{X};\theta_t)} \mathcal{L}(f(\mathcal{X}; \theta_t); \mathcal{Y}) \tag{S48}$$

$$= -\eta \, J^{\sigma,\dot\sigma}(x; \theta_t) J^{\sigma,\tilde\sigma}(\mathcal{X}; \theta_t)^\intercal \, \nabla_{f(\mathcal{X};\theta_t)} \mathcal{L}(f(\mathcal{X}; \theta_t); \mathcal{Y})$$

$$= -\eta \, \hat{I}^{(L)}(x, \mathcal{X}; \theta_t) \nabla_{f(\mathcal{X};\theta_t)} \mathcal{L}(f(\mathcal{X}; \theta_t); \mathcal{Y}), \tag{S49}$$

  with $\hat{I}^{(L)}$ as in Definition C.5 for $\sigma_1 = \sigma$, $\tilde\sigma_1 = \dot\sigma$, $\sigma_2 = \sigma$ and $\tilde\sigma_2 = \tilde\sigma$. For Equations (S48) and (S49) we assume that $\dot\sigma$ exists and is non-vanishing, but we deliberately train with a surrogate gradient. For example, we can again consider $\mathrm{erf}_m$ as the activation function. Its derivative explodes at zero and vanishes everywhere else as $m \to \infty$. The hope is that $\lim_{n_1,\ldots n_{L-1} \to \infty} \hat{I}_m^{(L)} = I_m^{(L)}$ exists and $\lim_{m \to \infty} I_m^{(L)}$ is not a singular kernel as before.

In this chapter we will deal with the second approach, because $J_{\sigma,\tilde\sigma}(x; \theta_t) =: G(\sigma; \tilde\sigma; x; \theta_t)$ is closer to $J_\theta f(x; \theta_t) = G(\sigma; \dot\sigma; x; \theta_t)$ than $J_\theta g(x; \theta_t) = G(\eta; \tilde\sigma; x; \theta_t)$ as a formula if $J_\theta f(x; \theta_t)$ exists and is non-vanishing. Here, $\eta$ denotes the surrogate activation function with derivative $\tilde\sigma$. To do this, we provide asymmetric generalizations of Theorem C.1, Theorem C.2, and Theorem C.3. These generalizations would, in principle, even allow us to compare the two approaches.

## E.1 Asymmetric generalization of the neural tangent kernel

For this section we adopt the so-called *linear envelope property* from [Matthews et al., 2018, Definition 1] to ensure that all expectations exist in the following theorems:

$$\exists\, m, c \geq 0 \quad \forall\, u \in \mathbb{R}: \; |\sigma(u)| \leq c + m|u| \tag{S50}$$

First, we consider networks under the weak infinite-width limit, and are thus interested in taking the number of hidden neurons to infinity sequentially. This is done inductively while using the central limit theorem and the weak law of large numbers. In order to do this rigorously, we state and prove two lemmata.

The first lemma is stated in terms of Gaussian measures on Hilbert spaces. An introduction to Gaussian measures on Hilbert spaces can be found in Chapter 1 of Da Prato [2006]. A rigorous derivation and definition of convergence in distribution on arbitrary metric spaces is given by Heyer [2009, Chapter 1.2]. This includes a definition of weak convergence in terms of continuous and bounded functions (Remark 1.2.5 (b)), a version of the Portemanteau theorem (Theorem 1.2.7), and the fact that convergence in probability implies convergence in distribution (Application 1.2.15).

**Lemma E.1.** *Let $H_i^m$ and $Z_i$, $i, m \in \mathbb{N}$, be random variables with values in a separable Hilbert space $\mathcal{H}$. Furthermore, let $Z_i$ be independent and identically distributed with finite mean and covariance operator $V$. If $(H_1^m, \ldots, H_k^m) \xrightarrow{\mathcal{D}} (Z_1, \ldots, Z_k)$ as $m \to \infty$ for all $k \in \mathbb{N}$, then there exists $k \colon \mathbb{N} \to \mathbb{N}$ such that $k(m) \to \infty$ monotonically as $m \to \infty$ and*

$$\frac{1}{\sqrt{k(m)}} \sum_{i=1}^{k(m)} H_i^m \xrightarrow[m \to \infty]{\mathcal{D}} Z, \tag{S51}$$

*for an $\mathcal{H}$-valued Gaussian random variable $Z$ with mean and covariance operator like $Z_1$. Similarly, there exists $k' \colon \mathbb{N} \to \mathbb{N}$ such that $k'(m) \to \infty$ monotonically as $m \to \infty$ and*

$$\frac{1}{k'(m)} \sum_{i=1}^{k'(m)} H_i^m \xrightarrow[m \to \infty]{\mathcal{D}} \mathbb{E}[Z]. \tag{S52}$$

*Proof.* Since $\mathcal{H}$ is separable and complete, convergence in distribution and convergence with respect to the Prokhorov metric $d$ (also known as the Lévy-Prokhorov metric) are equivalent [Billingsley, 1999, Theorem 6.8]. By the central limit theorem for separable Hilbert spaces [Zalesskiĭ et al., 1991], this implies

$$\lim_{k \to \infty} d\left( \frac{1}{\sqrt{k}} \sum_{i=1}^{k} Z_i, Z \right) = 0. \tag{S53}$$

In addition, the assumption together with the continuous mapping theorem gives that

$$\lim_{m \to \infty} d\left( \frac{1}{\sqrt{k}} \sum_{i=1}^{k} H_i^m, \frac{1}{\sqrt{k}} \sum_{i=1}^{k} Z_i \right) = 0 \quad \text{for all } k \in \mathbb{N}.$$

In particular, for any $k \in \mathbb{N}$, there exists some $m_k \in \mathbb{N}$ such that

$$d\left( \frac{1}{\sqrt{k}} \sum_{i=1}^{k} H_i^m, \frac{1}{\sqrt{k}} \sum_{i=1}^{k} Z_i \right) \leq \frac{1}{k} \quad \text{for all } m \geq m_k. \tag{S54}$$

We now want to choose $k(m)$ as large as possible for any $m$, but small enough to ensure Inequality (S54), i.e., $m \geq m_{k(m)}$. So we define

$$k(m) := \sup\{k \mid m \geq m_k\}.$$

First note that $\{k \mid m \geq m_k\} \neq \varnothing$, if $m \geq m_1$. The map $k \colon \mathbb{N} \to \mathbb{N}$ is therefore well-defined, as we consider $m \to \infty$. Similarly, we can find a $m \geq m_k$ for any given $k$. This yields

$$\lim_{m \to \infty} k(m) = \lim_{m \to \infty} \sup\{k \mid m \geq m_k\} = \infty.$$

By definition of $k(m)$, it holds $m \geq m_{k(m)}$ for all $m \in \mathbb{N}$ and thus

$$d\left(\frac{1}{\sqrt{k(m)}}\sum_{i=1}^{k(m)} H_i^m, \frac{1}{\sqrt{k(m)}}\sum_{i=1}^{k(m)} Z_i\right) \leq \frac{1}{k(m)} \quad \text{for all } m \in \mathbb{N}. \tag{S55}$$

Together with Equation (S53) this yields the claim, (S51):

$$d\left(\frac{1}{\sqrt{k(m)}}\sum_{i=1}^{k(m)} H_i^m, Z\right)$$

$$\leq \; d\left(\frac{1}{\sqrt{k(m)}}\sum_{i=1}^{k(m)} H_i^m, \frac{1}{\sqrt{k(m)}}\sum_{i=1}^{k(m)} Z_i\right) + d\left(\frac{1}{\sqrt{k(m)}}\sum_{i=1}^{k(m)} Z_i, Z\right)$$

$$\overset{(S55)}{\leq} \; \frac{1}{k(m)} + d\left(\frac{1}{\sqrt{k(m)}}\sum_{i=1}^{k(m)} Z_i, Z\right) \xrightarrow{m\to\infty} 0.$$

For the second claim, (S52), one can follow the same procedure but use the law of large numbers for Banach spaces instead of the central limit theorem. Suitable results are given by Ledoux and Talagrand [1991, Corollary 7.10] and Hoffmann-Jørgensen and Pisier [1976, Theorem 2.1]. Note that even the strong law of large numbers holds, but the weak law is sufficient. $\quad\square$

In the second lemma, some properties about convergence in distribution and convergence in probability are stated.

**Lemma E.2** (Theorem 2.7 from van der Vaart [1998], modified and (iv) added)**.** *Let $X_n, X$ and $Y_n$ be random vectors. Then*

  (i) $X_n \xrightarrow{\mathcal{P}} X$ *implies* $X_n \xrightarrow{\mathcal{D}} X$;

  (ii) $X_n \xrightarrow{\mathcal{P}} c$ *for a constant $c$ if and only if* $X_n \xrightarrow{\mathcal{D}} c$;

  (iii) *if* $X_n \xrightarrow{\mathcal{D}} X$ *and* $Y_n \xrightarrow{\mathcal{P}} c$ *for a constant $c$, then* $(X_n, Y_n) \xrightarrow{\mathcal{D}} (X, c)$;

  (iv) *if* $X_n \xrightarrow{\mathcal{D}} X$ *and $W$ is a random vector independent of $(X_n)_{n\in\mathbb{N}}$, then* $(X_n, W) \xrightarrow{\mathcal{D}} (X, W)$.

*Proof.* **(i) – (iii).** The proofs are given by van der Vaart [1998].

**(iv).** Let $f$ be a bounded and continuous function. Then,

$$\lim_{n\to\infty} \mathbb{E}[f(X_n, W)] = \lim_{n\to\infty} \int \mathbb{E}\left[f(X_n, W) \mid W\right](x)\,\mathrm{d}\mathbb{P}(w)$$

$$\overset{(\star)}{=} \lim_{n\to\infty} \int \mathbb{E}\left[f(X_n, W(w))\right]\mathrm{d}\mathbb{P}(w) = \int \lim_{n\to\infty} \mathbb{E}\left[f(X_n, W(w))\right]\mathrm{d}\mathbb{P}(w)$$

$$= \int \mathbb{E}\left[f(X, W(w))\right]\mathrm{d}\mathbb{P}(w) = \int \mathbb{E}\left[f(X, W) \mid W\right](w)\,\mathrm{d}\mathbb{P}(w) = \mathbb{E}[f(X, W)],$$

where we used the independence of $W$ and $(X_n)_{n\in\mathbb{N}}$ in Equation $(\star)$ and the boundedness of $f$ for the interchange of limit and integration. This proves convergence in distribution. $\quad\square$

**Remark E.1.** *The above theorem can be generalized to metric spaces. One can easily check that the proofs in [van der Vaart, 1998, Theorem 2.7] also work for metric spaces using the Portemanteau theorem provided by Heyer [2009, Theorem 1.2.7]. However, it is necessary to derive some more equivalent characterizations of convergence in distribution, which are given and used by van der Vaart [1998] but are missing in the work of Heyer [2009].*

**Theorem E.3** (Generalized version of Proposition 1 by Jacot et al. [2018])**.** *For activation functions $\sigma_1$ and $\sigma_2$ with property (S50), which are continuous except for finitely many jump points, let $f_1(\,\cdot\,; \theta)$ and $f_2(\,\cdot\,; \theta)$ be network functions with hidden layers $h_1^{(l)}(\,\cdot\,; \theta)$, $h_2^{(l)}(\,\cdot\,; \theta)$, for $1 \leq l \leq L$, respectively*

*as in Definition C.1 and with shared weights $\theta$. Then $(f_1(\,\cdot\,;\theta), f_2(\,\cdot\,;\theta))$ converges in distribution to a multidimensional Gaussian process $(X_j^{(L)}, Y_j^{(L)})_{j=1,\dots,n_L}$ as $(n_l)_{1\le l\le L-1} \to \infty$ weakly for any fixed countable input set $(z_i)_{i=1}^\infty$. The Gaussian process is defined by $X_j^{(L)} \overset{\text{iid}}{\sim} \mathcal{N}\left(0, \Sigma_1^{(L)}\right)$, $Y_j^{(L)} \overset{\text{iid}}{\sim} \mathcal{N}\left(0, \Sigma_2^{(L)}\right)$, where we have for $x, x' \in \mathbb{R}^{n_0}$*

$$\Sigma_1^{(1)}(x, x') = \Sigma_2^{(1)}(x, x') = \frac{\sigma_w^2}{n_0}\langle x, x'\rangle + \sigma_b^2 \tag{S56}$$

$$\Sigma_k^{(L)}(x, x') = \sigma_w^2 \, \mathbb{E}_{g\sim\mathcal{N}\left(0,\Sigma_k^{(L-1)}\right)}[\sigma_k(g(x))\,\sigma_k(g(x'))] + \sigma_b^2 \quad \text{for} \quad L \ge 2,\; k \in \{1, 2\}. \tag{S57}$$

*Furthermore, $X_i^{(L)}$ and $Y_j^{(L)}$ are independent if $i \ne j$ and*

$$\mathbb{E}\left[X_i^{(L)}(x)\, Y_i^{(L)}(x')\right] = \begin{cases} \frac{\sigma_w^2}{n_0}\langle x, x'\rangle + \sigma_b^2 = \Sigma_1^{(1)}(x, x') =: \Sigma_{1,2}^{(1)}(x, x') & \text{for } L = 1, \\ \sigma_w^2\, \mathbb{E}[\sigma_1(Z_1)\,\sigma_2(Z_2)] + \sigma_b^2 =: \Sigma_{1,2}^{(L)}(x, x') & \text{for } L \ge 2, \end{cases} \tag{S58}$$

*where $(Z_1, Z_2) \sim \mathcal{N}\left(0, \begin{pmatrix} \Sigma_1^{(L-1)}(x,x) & \Sigma_{1,2}^{(L-1)}(x,x') \\ \Sigma_{1,2}^{(L-1)}(x,x') & \Sigma_2^{(L-1)}(x',x') \end{pmatrix}\right)$.*

*Proof.* We write $h^{(l)}(x) = h^{(l)}(x;\theta)$. We prove the theorem by induction, as in the proof of Proposition 1 of Jacot et al. [2018], but expand on the technical details.

**$L = 1$.** By definition, we have

$$h_1^{(1)}(x) = h_2^{(1)}(x) = \frac{\sigma_w}{\sqrt{n_0}}W^{(1)}x + \sigma_b b^{(1)}.$$

This implies that $h_{k_1,i}^{(1)}(x;\theta)$ and $h_{k_2,j}^{(1)}(x';\theta)$, the $i$-th and $j$-th component of $h_{k_1}^{(1)}(x;\theta)$ and $h_{k_2}^{(1)}(x';\theta)$ respectively, are independent for any $k_1, k_2 \in \{1,2\}$, $x, x' \in \mathbb{R}^{n_0}$ and $i \ne j$. To prove that $(h_1^{(1)}(\,\cdot\,;\theta), h_2^{(1)}(\,\cdot\,;\theta))$ is a Gaussian process, it is thus sufficient to show that vectors of the form $\big( h_{1,i}^{(1)}(x_1) \; \dots \; h_{1,i}^{(1)}(x_n) \; h_{2,i}^{(1)}(x_1) \; \dots \; h_{2,i}^{(1)}(x_n) \big)^\mathsf{T}$ have a multivariate Gaussian distribution for any $x_1, \dots, x_n \in (z_i)_{i=1}^\infty$. It holds

$$h_{1,i}^{(1)} = \frac{\sigma_w}{\sqrt{n_0}}W_{i\cdot}^{(1)}x + \sigma_b b_i^{(1)} = (\sigma_w x^\mathsf{T}/n_0 \quad \sigma_b)\begin{pmatrix} W_{i\cdot}^{(1)\mathsf{T}} \\ b_i^{(1)} \end{pmatrix},$$

and therefore

$$\big(h_{1,i}^{(1)}(x_1) \quad \dots \quad h_{1,i}^{(1)}(x_n) \quad h_{2,i}^{(1)}(x_1) \quad \dots \quad h_{2,i}^{(1)}(x_n)\big)^\mathsf{T} = \begin{pmatrix} \sigma_w \mathcal{X}^\mathsf{T}/n_0 & \sigma_b \mathbb{1}_d \\ \sigma_w \mathcal{X}^\mathsf{T}/n_0 & \sigma_b \mathbb{1}_d \end{pmatrix}\begin{pmatrix} W_{i\cdot}^{(1)\mathsf{T}} \\ b_i^{(1)} \end{pmatrix},$$

with $\mathbb{1}_d$ a column vector of ones with length $d$. Now since

$$\begin{pmatrix} W_{i\cdot}^{(1)\mathsf{T}} \\ b_i^{(1)} \end{pmatrix} \sim \mathcal{N}(0, \mathrm{I}_{n_0+1}),$$

it holds

$$\big( h_{1,i}^{(1)}(x_1) \; \dots \; h_{1,i}^{(1)}(x_n)\, h_{2,i}^{(1)}(x_1) \; \dots \; h_{2,i}^{(1)}(x_n) \big)^\mathsf{T} \sim \mathcal{N}\left(0, \begin{pmatrix} \sigma_w \mathcal{X}^\mathsf{T}/n_0 & \sigma_b \mathbb{1}_d \\ \sigma_w \mathcal{X}^\mathsf{T}/n_0 & \sigma_b \mathbb{1}_d \end{pmatrix}\begin{pmatrix} \sigma_w \mathcal{X}^\mathsf{T}/n_0 & \sigma_b \mathbb{1}_d \\ \sigma_w \mathcal{X}^\mathsf{T}/n_0 & \sigma_b \mathbb{1}_d \end{pmatrix}^\mathsf{T}\right).$$

Therefore, the vector has a multivariate Gaussian distribution with the covariances required for Equation (S56) and the first case of Equations (S58).

**$L \to L + 1$.** We assume that the convergence holds for depth $L$. This means that there exists some $r \in \mathcal{R}_L$ such that, for given constant width $n_0$, any width $n_L$, and widths $n_l = r_l(m)$, $1 \le l < L$, it holds

$$\left(h_1^{(L)}(\cdot), h_2^{(L)}(\cdot)\right) \xrightarrow[m\to\infty]{\mathcal{D}} \left(X_j^{(L)}, X_j^{(L)}\right)_{j=1,\dots,n_L}.$$

To be precise, there is no such $r$ in the case $L = 1 \to L + 1$. However, this only makes the proof simpler and one can still follow the same steps as for $L \ge 2$.

By the continuous mapping theorem [Billingsley, 1999, Theorem 2.7] it holds

$$\left(\sigma_1\left(h_1^{(L)}(\cdot)\right),\sigma_2\left(h_2^{(L)}(\cdot)\right)\right)\xrightarrow[m\to\infty]{\mathcal{D}}\left(\sigma_1\left(X_j^{(L)}\right),\sigma_2\left(Y_j^{(L)}\right)\right)_{j=1,\ldots,n_L}. \tag{S59}$$

The theorem is applicable despite the finitely many jump points, since $(X^{(L)},Y^{(L)})$ assumes the values of the jump points with zero probability.

We now need to find an increasing width function $r_L\colon\mathbb{N}\to\mathbb{N}$ such that, if we additionally set $n_L=r_L(m)$, it holds for any fixed $n_{L+1}$

$$\left(h_1^{(L+1)}(\cdot),h_2^{(L+1)}(\cdot)\right)\xrightarrow[m\to\infty]{\mathcal{D}}\left(X_j^{(L)},X_j^{(L)}\right)_{j=1,\ldots,n_{L+1}}.$$

Note that by Remark C.5 we consider the product $\sigma$-algebra. Therefore, to show convergence in distribution for the whole process, it is sufficient to show convergence in distribution for the marginal distributions. By definition, we have for $k\in\{1,2\}$ that

$$h_k^{(L+1)}(x)=\frac{\sigma_w}{\sqrt{n_L}}W^{(L+1)}\sigma_k\left(h_k^{(L)}(x)\right)+\sigma_b b^{(L+1)}=\frac{\sigma_w}{\sqrt{n_L}}\sum_{i=1}^{n_L}\sigma_k\left(h_{k,i}^{(L)}(x)\right)W_{\cdot i}^{(L+1)}+\sigma_b b^{(L)}.$$

The marginal vector for points $x_1,\ldots,x_n$ as before can thus be written as

$$\begin{pmatrix}h_1^{(L+1)}(x_1)\\ \vdots\\ h_1^{(L+1)}(x_n)\\ h_2^{(L+1)}(x_1)\\ \vdots\\ h_2^{(L+1)}(x_n)\end{pmatrix}=\frac{\sigma_w}{\sqrt{n_L}}\sum_{i=1}^{n_L}\begin{pmatrix}\sigma_1\left(h_{1,i}^{(L)}(x_1)\right)\mathrm{I}_{n_{L+1}}\\ \vdots\\ \sigma_1\left(h_{1,i}^{(L)}(x_n)\right)\mathrm{I}_{n_{L+1}}\\ \sigma_2\left(h_{2,i}^{(L)}(x_1)\right)\mathrm{I}_{n_{L+1}}\\ \vdots\\ \sigma_2\left(h_{2,i}^{(L)}(x_n)\right)\mathrm{I}_{n_{L+1}}\end{pmatrix}W_{\cdot i}^{(L+1)}+\sigma_b\begin{pmatrix}b^{(L+1)}\\ \vdots\\ b^{(L+1)}\\ b^{(L+1)}\\ \vdots\\ b^{(L+1)}\end{pmatrix}.$$

With the same arguments as before and using the continuous mapping theorem in combination with Lemma E.2 (iv) and the independence of $W^{(L+1)}$, it holds

$$\left[\begin{pmatrix}\sigma_1\left(h_{1,i}^{(L)}(x_1)\right)\mathrm{I}_{n_{L+1}}\\ \vdots\\ \sigma_1\left(h_{1,i}^{(L)}(x_n)\right)\mathrm{I}_{n_{L+1}}\\ \sigma_2\left(h_{2,i}^{(L)}(x_1)\right)\mathrm{I}_{n_{L+1}}\\ \vdots\\ \sigma_2\left(h_{2,i}^{(L)}(x_n)\right)\mathrm{I}_{n_{L+1}}\end{pmatrix}W_{\cdot i}^{(L+1)}\right]_{i=1}^{n_L}\xrightarrow[m\to\infty]{\mathcal{D}}\left[\begin{pmatrix}\sigma_1\left(X_i^{(L)}(x_1)\right)\mathrm{I}_{n_{L+1}}\\ \vdots\\ \sigma_1\left(X_i^{(L)}(x_n)\right)\mathrm{I}_{n_{L+1}}\\ \sigma_2\left(Y_i^{(L)}(x_1)\right)\mathrm{I}_{n_{L+1}}\\ \vdots\\ \sigma_2\left(Y_i^{(L)}(x_n)\right)\mathrm{I}_{n_{L+1}}\end{pmatrix}W_{\cdot i}^{(L+1)}\right]_{i=1}^{n_L}.$$

Since $(X_i,Y_i)$ and $(X_j,Y_j)$ are independent for $i\neq j$, the conditions of Lemma E.1 are satisfied. Now, there exists $k\colon\mathbb{N}\to\mathbb{N}$ such that $k(m)\to\infty$ monotonically as $m\to\infty$ and, when setting $n_L:=r_L(m):=k(m)$, it holds

$$\begin{pmatrix}h_1^{(L+1)}(x_1)\\ \vdots\\ h_1^{(L+1)}(x_n)\\ h_2^{(L+1)}(x_1)\\ \vdots\\ h_2^{(L+1)}(x_n)\end{pmatrix}\xrightarrow[m\to\infty]{\mathcal{D}}\sigma_w\begin{pmatrix}R_{x_1}^{(L+1)}\\ \vdots\\ R_{x_n}^{(L+1)}\\ S_{x_1}^{(L+1)}\\ \vdots\\ S_{x_n}^{(L+1)}\end{pmatrix}+\sigma_b\begin{pmatrix}b^{(L+1)}\\ \vdots\\ b^{(L+1)}\\ b^{(L+1)}\\ \vdots\\ b^{(L+1)}\end{pmatrix}\overset{(\star)}{=}\begin{pmatrix}X^{(L+1)}(x_1)\\ \vdots\\ X^{(L+1)}(x_n)\\ Y^{(L+1)}(x_1)\\ \vdots\\ Y^{(L+1)}(x_n)\end{pmatrix},$$

for a Gaussian random variable $\left(R_{x_1}^{(L+1)}\ldots,R_{x_n}^{(L+1)},S_{x_1}^{(L+1)},\ldots,S_{x_n}^{(L+1)}\right)\in\mathbb{R}^{2\cdot n_{L+1}\cdot d}$ with

$$\begin{pmatrix}R_{x_1}^{(L+1)}\\ \vdots\\ R_{x_n}^{(L+1)}\\ S_{x_1}^{(L+1)}\\ \vdots\\ S_{x_n}^{(L+1)}\end{pmatrix}\sim\begin{pmatrix}\sigma_1\left(X_1^{(L)}(x_1)\right)\mathrm{I}_{n_{L+1}}\\ \vdots\\ \sigma_1\left(X_1^{(L)}(x_n)\right)\mathrm{I}_{n_{L+1}}\\ \sigma_2\left(Y_1^{(L)}(x_1)\right)\mathrm{I}_{n_{L+1}}\\ \vdots\\ \sigma_2\left(Y_1^{(L)}(x_n)\right)\mathrm{I}_{n_{L+1}}\end{pmatrix}W_{\cdot 1}^{(L+1)}.$$

Before considering the covariances, we want to comment on $r_L$. First, note that this sequence may not initially be strictly increasing, but this can be circumvented by considering a strictly increasing subsequence. Second, $r_L$ could theoretically depend on $\mathcal{X}$. However, this can be resolved by not evaluating the pair $(h_1^{(L)}, h_2^{(L)})$ at certain data points, but doing the same calculation as above for $(h_1^{(L)}, h_2^{(L)})$. The starting point for this is Equation (S59). To apply Lemma E.1, note additionally that $(h_1^{(L)}, h_2^{(L)}) \in \bigotimes_{i=1}^{\infty} \mathbb{R}^{2 \cdot n_L}$, which is a separable Hilbert space because we are considering a countable input set. Above, we worked with the marginal distribution because this makes the following calculation of the covariances easier. Finally, the choice of $r_L$ should be independent of $n_{L+1}$. This follows from the independence of $W_{ji}^{(L+1)}$ and $W_{j'i}^{(L+1)}$ for $j \neq j'$.

To verify Equation $(\star)$, we check the covariances of the random vector. They are given by

$$\mathrm{Cov}\left[R_{x_i}^{(L+1)}, R_{x_j}^{(L+1)}\right] = \mathbb{E}\left[\sigma_1\left(X_1^{(L)}(x_i)\right) W_{\cdot 1}^{(L+1)} \left(W_{\cdot 1}^{(L+1)}\right)^{\mathsf{T}} \sigma_1\left(X_1^{(L)}(x_j)\right)\right]$$
$$= \mathbb{E}\left[\sigma_1\left(X_1^{(L)}(x_i)\right) \mathbb{E}\left[W_{\cdot 1}^{(L+1)} \left(W_{\cdot 1}^{(L+1)}\right)^{\mathsf{T}} \middle| X_1^{(L)}(x_i), X_1^{(L)}(x_j)\right] \sigma_1\left(X_1^{(L)}(x_j)\right)\right]$$
$$= \mathbb{E}\left[\sigma_1(X_1^{(L)}(x_i)) \, \mathrm{I}_{n_{L+1}} \, \sigma_1\left(X_1^{(L)}(x_j)\right)\right] = \mathbb{E}\left[\sigma_1\left(X_1^{(L)}(x_i)\right) \sigma_1\left(X_1^{(L)}(x_j)\right)\right] \mathrm{I}_{n_{L+1}}.$$

Similarly, we get that

$$\mathrm{Cov}\left[S_{x_i}^{(L+1)}, S_{x_j}^{(L+1)}\right] = \mathbb{E}\left[\sigma_2\left(Y_1^{(L)}(x_i)\right) \sigma_2\left(Y_1^{(L)}(x_j)\right)\right] \mathrm{I}_{n_{L+1}},$$

and together this implies using the independence of biases $b^{(L+1)}$ and weight matrices $W^{(L+1)}$

$$\mathrm{Cov}\left[\sigma_w R_{x_i,k}^{(L+1)} + \sigma_b b_k^{(L+1)}, \sigma_w R_{x_j,l}^{(L+1)} + \sigma_b b_l^{(L+1)}\right]$$
$$= \delta_{kl}\left(\sigma_w^2 \mathbb{E}\left[\sigma_1\left(X_1^{(L)}(x_i)\right) \sigma_1\left(X_1^{(L)}(x_j)\right)\right] + \sigma_b^2\right) = \mathrm{Cov}\left[X_k^{(L+1)}(x_i), X_l^{(L+1)}(x_j)\right],$$
$$\mathrm{Cov}\left[\sigma_w S_{x_i,k}^{(L+1)} + \sigma_b b_k^{(L+1)}, \sigma_w S_{x_j,l}^{(L+1)} + \sigma_b b_l^{(L+1)}\right]$$
$$= \delta_{kl}\left(\sigma_w^2 \mathbb{E}\left[\sigma_2\left(Y_1^{(L)}(x_i)\right) \sigma_2\left(Y_1^{(L)}(x_j)\right)\right] + \sigma_b^2\right) = \mathrm{Cov}\left[Y_k^{(L+1)}(x_i), Y_l^{(L+1)}(x_j)\right].$$

We therefore proved Equation (S57). For the second case of (S58), we see that

$$\mathrm{Cov}\left[R_{x_i}^{(L+1)}, S_{x_j}^{(L+1)}\right] = \mathbb{E}\left[\sigma_1\left(X_1^{(L)}(x_i)\right) W_{\cdot 1}^{(L+1)} \left(W_{\cdot 1}^{(L+1)}\right)^{\mathsf{T}} \sigma_2\left(Y_1^{(L)}(x_j)\right)\right]$$
$$= \mathbb{E}\left[\sigma_1\left(X_1^{(L)}(x_i)\right) \mathbb{E}\left[W_{\cdot 1}^{(L+1)} \left(W_{\cdot 1}^{(L+1)}\right)^{\mathsf{T}} \middle| X_1^{(L)}(x_i), Y_1^{(L)}(x_j)\right] \sigma_2\left(Y_1^{(L)}(x_j)\right)\right]$$
$$= \mathbb{E}\left[\sigma_1\left(X_1^{(L)}(x_i)\right) \sigma_2\left(Y_1^{(L)}(x_j)\right)\right] \mathrm{I}_{n_{L+1}},$$

with, by induction hypothesis,

$$\left(X_1^{(L)}(x_i), Y_1^{(L)}(x_j)\right) \sim \mathcal{N}\left(0, \begin{pmatrix} \Sigma_1^{(L)}(x_i, x_i) & \Sigma_{1,2}^{(L)}(x_i, x_j) \\ \Sigma_{1,2}^{(L)}(x_i, x_j) & \Sigma_2^{(L)}(x_j, x_j) \end{pmatrix}\right).$$

This finished the proof, since it now holds

$$\mathrm{Cov}\left[\sigma_w R_{x_i,k}^{(L+1)} + \sigma_b b_k^{(L+1)}, \sigma_w S_{x_j,l}^{(L+1)} + \sigma_b b_l^{(L+1)}\right]$$
$$= \delta_{kl}\left(\sigma_w^2 \mathbb{E}\left[\sigma_1\left(X_1^{(L)}(x_i)\right) \sigma_2\left(Y_1^{(L)}(x_j)\right)\right] + \sigma_b^2\right) = \mathrm{Cov}\left[X_k^{(L+1)}(x_i), Y_l^{(L+1)}(x_j)\right],$$

$\square$

**Remark E.2.** *In the preceding proof, we checked the marginal distributions of arbitrary size in order to give a complete proof of the convergence to a Gaussian process. However, the covariances can be derived by considering only a pair of data points $(x_1, x_2)$, which drastically simplifies the notation. Also, we only need the distributions of pairs for the next theorems.*

**Theorem E.4** (Generalized version of Theorem 1 by Jacot et al. [2018]). *For activation functions $\sigma_1$, $\sigma_2$ and so-called surrogate derivatives $\tilde\sigma_1$, $\tilde\sigma_2$ such that $\sigma_1, \sigma_2, \tilde\sigma_1$, and $\tilde\sigma_2$ are continuous except for finitely many jump points with property (S50), let $f_1(\,\cdot\,;\theta)$ and $f_2(\,\cdot\,;\theta)$ be network functions with hidden layers $h_1^{(l)}(\,\cdot\,;\theta)$, $h_2^{(l)}(\,\cdot\,;\theta)$, $1 \le l \le L$, respectively as in Definition C.1 with shared weights $\theta$. Denote the empirical generalized neural tangent kernel*

$$\hat{I}^{(L)}(x, x') = J^{(L),\sigma_1,\tilde\sigma_1}(x;\theta)\, J^{(L),\sigma_2,\tilde\sigma_2}(x';\theta)^\intercal \quad for\ x, x' \in \mathbb{R}^{n_0},$$

*as in Definition C.5. Then, for any $x, x' \in \mathbb{R}^{n_0}$ and $1 \le i, j \le n_L$, it holds*

$$\hat{I}_{ij}^{(L)}(x, x') \xrightarrow{\mathcal{P}} \delta_{ij} I^{(L)}(x, x'),$$

*as $n_1, \ldots, n_{L-1} \to \infty$ weakly. We call $I^{(L)}$ the analytic generalized neural tangent kernel, which is recursively given by*

$$I^{(1)}(x, x') = \Sigma_{1,2}^{(1)}(x, x') \tag{S60}$$

$$I^{(L)}(x, x') = \Sigma_{1,2}^{(L)}(x, x') + I^{(L-1)}(x, x') \cdot \tilde\Sigma_{1,2}^{(L)}(x, x') \quad for\ L \ge 2, \tag{S61}$$

*with $\Sigma_{1,2}^{(L)}$ as in Theorem E.3 and*

$$\tilde\Sigma_{1,2}^{(L)}(x, x') = \sigma_w^2\, \mathbb{E}[\tilde\sigma_1(Z_1)\, \tilde\sigma_2(Z_2)] \quad for\ L \ge 2,$$

*where $(Z_1, Z_2) \sim \mathcal{N}\left(0, \begin{pmatrix} \Sigma_1^{(L-1)}(x,x) & \Sigma_{1,2}^{(L-1)}(x,x') \\ \Sigma_{1,2}^{(L-1)}(x,x') & \Sigma_2^{(L-1)}(x',x') \end{pmatrix}\right).$*

*Proof.* We prove the theorem by induction over $L$, as in the proof of Theorem 1 by Jacot et al. [2018]. We denote $J^{(L),k}(z) = J^{(L),\sigma_k,\tilde\sigma_k}(z;\theta)$ for $k \in \{1, 2\}, z \in \mathbb{R}^{n_0}$.

**L = 1.** For $1 \le i, j \le n_1$ it holds

$$\hat{I}_{ij}^{(1)}(x, x') = J_{i\cdot}^{(1),1}(x)\, J_{j\cdot}^{(1),2}(x')^\intercal = \sum_{\theta' \in \theta^{(1)}} J_{i\,\theta'}^{(1),1}(x)\, J_{j\,\theta'}^{(1),2}(x')$$

$$\overset{(S12)}{=} \sum_{\substack{1 \le k \le n_1 \\ 1 \le l \le n_0}} \delta_{ki} \frac{\sigma_w}{\sqrt{n_0}} x_l\, \delta_{kj} \frac{\sigma_w}{\sqrt{n_0}} x_l' + \sum_{1 \le k \le n_1} \delta_{ki}\sigma_b\, \delta_{kj}\sigma_b$$

$$= \frac{\sigma_w^2}{n_0}\delta_{ij} \sum_{1 \le k \le n_0} x_l x_l' + \sigma_b^2\delta_{ij} = \delta_{ij}\left(\frac{\sigma_w^2}{n_0}\langle x, x'\rangle + \sigma_b^2\right) = \delta_{ij} I^{(1)}(x, x').$$

This proves Equation (S60).

**L → L + 1.** Now we assume that the statement is true for $L$ and need to prove it for $L + 1$. Instead of considering an explicit $r \in \mathcal{R}_L$ as in the proof of Theorem E.3 and taking $m \to \infty$, we will just write "$n_1, \ldots, n_{L-1} \to \infty$ weakly".

In the induction step, we would like to use Theorem E.3. However, it is not obvious that this is possible. In the setting of Definition C.7, let $\mathcal{S}_1$ and $\mathcal{S}_2$ be two statements that hold weakly. In our setting, these are the induction hypothesis and Theorem E.3 for depth $L$. Then there exist $s, t \in \mathcal{R}_L$ such that $\mathcal{S}_1(s)$ and $\mathcal{S}_2(t)$ are true. It is not clear that there exists some $r \in \mathcal{R}_L$ such that $\mathcal{S}_1(r)$ and $\mathcal{S}_2(r)$ are true. It would be natural to define $r$ by $r_l(m) := \max\{s_l(m), t_l(m)\}$ for all $1 \le l < L$ and $m \in \mathbb{N}$, but it is still unclear that this implies $\mathcal{S}_1(r)$ and $\mathcal{S}_2(r)$. Instead, one can consider the combined statement "$\mathcal{S}_1$ and $\mathcal{S}_2$" as a new statement $\mathcal{S}$. In our case, for any depth $L$, we would need to find a $r \in \mathcal{R}_L$ such that the statements in Theorem E.3 and Theorem E.4 are both true, which can be done. Since we used the first part of Lemma E.1 to prove Theorem E.3 and will use the second part of Lemma E.1 for this proof, we would have to define $r_L(m) := \min\{k(m), k'(m)\}$. For simplicity, we will assume that the $r \in \mathcal{R}_L$ we get by the induction hypothesis also gives us the convergence statement of Theorem E.3.

Using the definition of the generalized NTK and the quasi-Jacobian matrices, we obtain for $1 \leq i, j \leq n_{L+1}$

$$I_{ij}^{(L+1)}(x, x') = J_{i\cdot}^{(L+1),1}(x) J_{j\cdot}^{(L+1),2}(x')^{\mathsf{T}} = \sum_{\theta' \in \theta^{(1:\, L+1)}} J_{i\,\theta'}^{(1),1}(x) J_{j\,\theta'}^{(1),2}(x')$$

$$= \sum_{\substack{1 \leq k \leq n_{L+1} \\ 1 \leq l \leq n_L}} \delta_{ki} \frac{\sigma_w}{\sqrt{n_L}} \sigma_1\left(h_{1,l}^{(L)}(x)\right) \delta_{kj} \frac{\sigma_w}{\sqrt{n_L}} \sigma_2\left(h_{2,l}^{(L)}(x')\right) + \sum_{1 \leq k \leq n_{L+1}} \delta_{ki}\sigma_b\, \delta_{kj}\sigma_b$$

$$+ \sum_{\theta' \in \theta^{(1:\, L)}} \frac{\sigma_w^2}{n_L} \left[\sum_{m=1}^{n_L} W_{i,m}^{(L+1)} \tilde\sigma_1\left(h_{1,m}^{(L)}(x)\right) J_{m\,\theta'}^{(L),1}(x)\right] \left[\sum_{r=1}^{n_L} W_{i,r}^{(L+1)} \tilde\sigma_2\left(h_{2,r}^{(L)}(x')\right) J_{r\,\theta'}^{(L),2}(x')\right]$$

$$= \delta_{ij}\left(\frac{\sigma_w^2}{n_L}\sum_{l=1}^{n_L}\sigma_1\left(h_{1,l}^{(L)}(x)\right)\sigma_2\left(h_{2,l}^{(L)}(x')\right) + \sigma_b^2\right) \tag{S62}$$

$$+ \frac{\sigma_w^2}{n_L}\sum_{m,r=1}^{n_L} W_{i,m}^{(L+1)} W_{j,r}^{(L+1)} \tilde\sigma_1\left(h_{1,m}^{(L)}(x)\right)\tilde\sigma_2\left(h_{2,r}^{(L)}(x')\right) \cdot \sum_{\theta' \in \theta^{(1:\, L)}} J_{m\,\theta'}^{(L),1}(x) J_{r\,\theta'}^{(L),2}(x') \tag{S63}$$

We want to apply the second part of Lemma E.1. We will consider the terms (S62) and (S63) separately. First note that

$$\left(\sigma_1\left(h_{1,l}^{(L)}(x)\right), \sigma_2\left(h_{2,l}^{(L)}(x')\right)\right) \xrightarrow{\mathcal{D}} \left(\sigma_1\left(X_l^{(L)}(x)\right), \sigma_2\left(Y_l^{(L)}(x')\right)\right), \tag{S64}$$

as $n_1, \ldots, n_{L-1} \to \infty$ weakly by Theorem E.3 and the continuous mapping theorem with

$$\left(X_l^{(L)}(x), Y_l^{(L)}(x')\right) \overset{\text{iid}}{\sim} \mathcal{N}\left(0, \begin{pmatrix} \Sigma_{1,1}^{(L)}(x, x) & \Sigma_{1,2}^{(L)}(x, x') \\ \Sigma_{1,2}^{(L)}(x, x') & \Sigma_{2,2}^{(L)}(x', x') \end{pmatrix}\right).$$

Again, we used that the values of the jump points are assumed with zero probability. Thus, again by the continuous mapping theorem and by the second part of Lemma E.1, it holds

$$\frac{\sigma_w^2}{n_L}\sum_{l=1}^{n_L}\sigma_1\left(h_{1,l}^{(L)}(x)\right)\sigma_2\left(h_{2,l}^{(L)}(x')\right) + \sigma_b^2 \xrightarrow[\substack{n_1,\ldots,n_L \to \infty \\ \text{weakly}}]{\mathcal{D}} \mathbb{E}\left[\sigma_1\left(X_1^{(L)}(x)\right)\sigma_2\left(Y_1^{(L)}(x')\right)\right] + \sigma_b^2.$$

Here, as in the proof of Theorem E.3, $n_L \to \infty$ is given by $n_L := r_L(m) := k'(m)$, which is in turn is given by Lemma E.1. Note also that the limit is a constant, which implies convergence in probability according to Lemma E.2 (ii). In conclusion, we obtain

$$\delta_{ij}\left(\frac{\sigma_w^2}{n_L}\sum_{l=1}^{n_L}\sigma_1\left(h_{1,l}^{(L)}(x)\right)\sigma_2\left(h_{2,l}^{(L)}(x')\right) + \sigma_b^2\right) \xrightarrow[\substack{n_1,\ldots,n_L \to \infty \\ \text{weakly}}]{\mathcal{P}} \delta_{ij}\, \Sigma_{1,2}^{(L+1)}(x, x'). \tag{S65}$$

For term (S63), we can apply the induction hypothesis to obtain

$$\sum_{\theta' \in \theta^{(1:\, L)}} J_{\theta',m}^{(L),1}(x) J_{\theta',r}^{(L),2}(y) = \hat{I}_{m,r}^{(L)}(x, y) \xrightarrow[\substack{n_1,\ldots,n_{L-1} \to \infty \\ \text{weakly}}]{\mathcal{P}} \delta_{mr} I^{(L)}(x, y). \tag{S66}$$

Also, we can again use the convergence given by (S64), but with $\sigma_1, \sigma_2$ replaced by $\tilde\sigma_1, \tilde\sigma_2$ respectively. This gives using Lemma E.2 (iv)

$$\left(\tilde\sigma_1\left(h_1^{(L)}(\cdot)\right), \tilde\sigma_1\left(h_2^{(L)}(\cdot)\right), W^{(L+1)}\right) \xrightarrow[\substack{n_1,\ldots,n_{L-1} \\ \text{weakly}}]{\mathcal{D}} \left(\tilde\sigma_1\left(X^{(L)}\right), \tilde\sigma_1\left(Y^{(L)}\right), W^{(L+1)}\right).$$

Together with (S66) and Lemma E.2 (iii) this implies

$$\left(\tilde\sigma_1\left(h_1^{(L)}(\cdot)\right), \tilde\sigma_1\left(h_2^{(L)}(\cdot)\right), W^{(L+1)}, \hat{I}_{mr}^{(L)}(x, x')\right)$$

$$\xrightarrow[\substack{n_1,\ldots,n_{L-1} \\ \text{weakly}}]{\mathcal{D}} \left(\tilde\sigma_1\left(X^{(L)}\right), \tilde\sigma_1\left(Y^{(L)}\right), W^{(L+1)}, \delta_{mr} I^{(L)}(x, x')\right).$$

For the summands in (S63) we therefore have by the continuous mapping theorem

$$\sum_{r=1}^{n_L} W_{i,m}^{(L+1)} W_{j,r}^{(L+1)} \tilde{\sigma}_1\left(h_{1,m}^{(L)}(x)\right) \tilde{\sigma}_2\left(h_{2,r}^{(L)}(x')\right) \cdot \sum_{\theta' \in \theta^{(1:L)}} J_{\theta',m}^{(L),1}(x) J_{\theta',r}^{(L),2}(x')$$

$$\xrightarrow[\substack{n_1,\ldots,n_{L-1} \to \\ \text{weakly}}]{\mathcal{D}} \sum_{r=1}^{n_L} W_{i,m}^{(L+1)} W_{j,r}^{(L+1)} \tilde{\sigma}_1\left(X_m^{(L)}(x)\right) \tilde{\sigma}_2\left(Y_r^{(L)}(x')\right) \cdot \delta_{mr} I^{(L)}(x,x')$$

$$= W_{i,m}^{(L+1)} W_{j,m}^{(L+1)} \tilde{\sigma}_1\left(X_m^{(L)}(x)\right) \tilde{\sigma}_2\left(Y_m^{(L)}(x')\right) \cdot I^{(L)}(x,x')$$

These terms are independent for different $1 \le m \le n_L$. Therefore, the second part of Lemma E.1 can be applied as before. This yields

$$\frac{\sigma_w^2}{n_L} \sum_{r=1}^{n_L} W_{i,m}^{(L+1)} W_{j,r}^{(L+1)} \tilde{\sigma}_1\left(h_{1,m}^{(L)}(x)\right) \tilde{\sigma}_2\left(h_{2,r}^{(L)}(x')\right) \cdot \sum_{\theta' \in \theta^{(1:L)}} J_{\theta',m}^{(L),1}(x) J_{\theta',r}^{(L),2}(x')$$

$$\xrightarrow[\substack{n_1,\ldots,n_L \to \infty \\ \text{weakly}}]{\mathcal{D}} I^{(L)}(x,x') \cdot \sigma_w^2 \, \mathbb{E}\left[W_{i,1}^{(L+1)} W_{j,1}^{(L+1)} \tilde{\sigma}_1\left(X_1^{L)}(x)\right) \tilde{\sigma}_2\left(Y_1^{(L)}(x')\right)\right]$$

$$= I^{(L)}(x,x') \cdot \delta_{ij} \sigma_w^2 \, \mathbb{E}\left[\tilde{\sigma}_1\left(X_1^{L)}(x)\right) \tilde{\sigma}_2\left(Y_1^{(L)}(x')\right)\right] = \delta_{ij} I^{(L)}(x,x') \cdot \tilde{\Sigma}_{1,2}^{(L)}(x,x').$$

Together with (S65) this yields Equation (S61) and concludes the proof. $\qquad\square$

The two theorems above are complemented by the convergence of the generalized NTK in the infinite-width limit during training, which we will prove below. The theorem is a generalization of Theorem G.2 from Lee et al. [2019]. There, the convergence of the NTK at initialization as $(n_l)_{l=1}^{L-1} \gtrsim n$ is assumed. More precisely, they refer to a result of Yang [2019a]. In consequence, we have to assume the same for the next theorem. The proof of Yang [2019a] could also be generalizable to our case. Alternatively, one could also try to generalize the statement of Jacot et al. [2018] on stability during training in the weak infinite-width limit.

**Theorem E.5** (Based on Theorem G.2 from Lee et al. [2019] ). *Let $\sigma$ be a Lipschitz continuous and differentiable activation function. Let the derivative of the activation function $\dot{\sigma}$ and a so-called surrogate derivative $\tilde{\sigma}$ be Lipschitz continuous and bounded. Let $f_t(\,\cdot\,;\theta)$ be the corresponding network function with depth $L$ initialized as in Definition C.1 and trained with MSE loss and surrogate gradient learning, i.e., according to Equation (S49) with surrogate derivative $\tilde{\sigma}$. The hidden layers are denoted by $h_l^{(l)}(\,\cdot\,;\theta)$, for $1 \le l < L$, as in Definition C.1. Assume that the generalized NTK converges in probability to the analytic generalized NTK of Theorem E.4 as $(n_l)_{l=1}^{L-1} \gtrsim n$,*

$$\left(J^{(L),\sigma,\dot{\sigma}}\right) \left(J^{(L),\sigma,\tilde{\sigma}}\right)^{\mathsf{T}} = \hat{I}^{(L)} \xrightarrow[n\to\infty]{\mathcal{P}} I^{(L)} \otimes \mathrm{I}_{n_L}.$$

*Furthermore, assume that the smallest and largest eigenvalue of the symmetrization of $I^{(L)}(\mathcal{X},\mathcal{X})$,*

$$S^{(L)} := \frac{1}{2}\left(I^{(L)}(\mathcal{X},\mathcal{X}) + I^{(L)}(\mathcal{X},\mathcal{X})^{\mathsf{T}}\right),$$

*are given by $0 < \lambda_{\min} \le \lambda_{\max} < \infty$ and that the learning rate is given by $\eta > 0$. Then, for any $\delta > 0$ there exist $R > 0, N \in \mathbb{N}$ and $K > 1$ such that for every $n \ge N$, the following holds with probability at least $1 - \delta$ over random initialization*

$$\sup_{t \in [0,\infty)} \left\|\hat{I}^{(L)}(\mathcal{X},\mathcal{X}) - I^{(L)}(\mathcal{X},\mathcal{X})\right\|_F \le \frac{6K^3 R}{\lambda_{\min}} n^{-\frac{1}{2}},$$

*where $\|\cdot\|_F$ denotes the Frobenius norm.*

*Proof.* We follow the proofs in Chapter G of Lee et al. [2019]. The analogous statement is given by Theorem G.2 of Lee et al. [2019]. Since we are considering the infinite-width limit $(n_l)_{l=1}^{L-1} \gtrsim n$, we will assume that the width of the hidden layers are given by $n \in \mathbb{N}$. The more general case can easily be proved using the same steps. Recall that

$$\frac{\mathrm{d}}{\mathrm{d}t} f(x;\theta_t) = -\eta \, \hat{I}^{(L)}(x,\mathcal{X};\theta_t) \nabla_{f(\mathcal{X};\theta_t)} \mathcal{L}(f(\mathcal{X};\theta_t); \mathcal{Y}). \tag{S49}$$

The loss function is the MSE loss by assumption. This yields

$$\nabla_{f(\mathcal{X};\theta_t)}\mathcal{L}(f(\mathcal{X};\theta_t);\mathcal{Y}) = \nabla_{f(\mathcal{X};\theta_t)}\frac{1}{2}\|f(\mathcal{X};\theta_t) - \mathcal{Y}\|_2^2 = f(\mathcal{X};\theta_t) - \mathcal{Y} =: g(\theta_t).$$

The change of weights over time is hence given by

$$\frac{\mathrm{d}}{\mathrm{d}t}\theta_t \stackrel{(S48)}{=} -\eta\, J^{(L),\sigma,\tilde{\sigma}}(\mathcal{X};\theta_t)^\intercal\, g(\theta_t)$$

By Lemma E.6 below, for any $\delta_1 > 0$ there exists a $K > 0$ such that for any $C > 0$ it holds with probability at least $1 - \delta_1$

$$\left\|J^{(L),\sigma,\tilde{\sigma}}(\mathcal{X};\theta_t)^\intercal\right\|_F \leq K \quad \text{for all } \theta_t \in B\left(\theta_0, C\right).$$

Together, this implies

$$\frac{\mathrm{d}}{\mathrm{d}t}\|\theta_t - \theta_0\|_2 \stackrel{(\star)}{\leq} \left\|\frac{\mathrm{d}}{\mathrm{d}t}\theta_t\right\|_2 = \eta\left\|J^{(L),\sigma,\tilde{\sigma}}(\mathcal{X};\theta_t)^\intercal\, g(\theta_t)\right\|_2$$

$$\leq \eta\left\|J^{(L),\sigma,\tilde{\sigma}}(\mathcal{X};\theta_t)^\intercal\right\|_F \|g(\theta_t)\|_2 \leq \eta\, K\, \|g(\theta_t)\|_2, \tag{S67}$$

for all $\theta_t \in B\left(\theta_0, C\right)$ with probability at least $1 - \delta_1$. Inequality $(\star)$ is a consequence of the Cauchy-Schwarz inequality.

To estimate the term $\|g(\theta_t)\|_2$ in Inequality (S67), we use Grönwall's inequality in the differential form, e.g., Lemma 1.1.1 from Qin [2016]). First, note that as a consequence of the proof of Lemma E.6, for all $\delta_2 > 0$ there exist $R_0 > 0$ and $n_0 \in \mathbb{N}$ such that with probability at least $1 - \delta_2$ it holds for all $n \geq n_0$

$$\|g(\theta_0)\|_2 \leq R_0. \tag{S68}$$

We now set $C := 3KR_0/\lambda_{\min}$. Next, observe that

$$\frac{\mathrm{d}}{\mathrm{d}t}g(\theta_t) = \frac{\mathrm{d}}{\mathrm{d}t}f(\mathcal{X};\theta_t) \stackrel{(S49)}{=} -\eta\, J^{(L),\sigma,\dot{\sigma}}(\mathcal{X};\theta_t)\, J^{(L),\sigma,\tilde{\sigma}}(\mathcal{X};\theta_t)^\intercal\, g(\theta_t)$$

$$\implies \frac{\mathrm{d}}{\mathrm{d}t}\|g(\theta_t)\|_2^2 = \frac{\mathrm{d}}{\mathrm{d}t}\langle g(\theta_t), g(\theta_t)\rangle = 2\left\langle g(\theta_t), \frac{\mathrm{d}}{\mathrm{d}t}g(\theta_t)\right\rangle$$

$$= -2\eta\left\langle g(\theta_t), J^{(L),\sigma,\dot{\sigma}}(\mathcal{X};\theta_t)\, J^{(L),\sigma,\tilde{\sigma}}(\mathcal{X};\theta_t)^\intercal\, g(\theta_t)\right\rangle$$

$$= -2\eta\left\langle g(\theta_t), \hat{I}_t^{(L)}(\mathcal{X},\mathcal{X})\, g(\theta_t)\right\rangle = -2\eta\left\langle g(\theta_t), S_t^{(L)}g(\theta_t)\right\rangle$$

$$\stackrel{(\star\star)}{\leq} -2\eta\left\langle g(\theta_t), \frac{1}{3}\lambda_{\min}\mathrm{I}\, g(\theta_t)\right\rangle = -\frac{2}{3}\eta\lambda_{\min}\|g(\theta_t)\|_2^2, \tag{S69}$$

with $S_t^{(L)} := \frac{1}{2}\left(\hat{I}_t^{(L)}(\mathcal{X},\mathcal{X}) + \hat{I}_t^{(L)}(\mathcal{X},\mathcal{X})^\intercal\right)$. To show $(\star\star)$, we will prove that for any $0 \neq z \in \mathbb{R}^{n_L \cdot n}$

$$\left\langle z, \left(S_t^{(L)} - \frac{1}{3}\lambda_{\min}\mathrm{I}\right)z\right\rangle \geq 0. \tag{S70}$$

First, note that by assumption for all $z \neq 0$

$$\left\langle z, \left(S^{(L)} - \lambda_{\min}\mathrm{I}\right)z\right\rangle \geq 0.$$

Since the generalized NTK converges at initialization in probability, we can assume that with probability at least $1 - \delta_3$ for any $\delta_3 > 0$ that

$$\left\|\hat{I}_0^{(L)}(\mathcal{X},\mathcal{X}) - I^{(L)}(\mathcal{X},\mathcal{X})\right\|_2 \leq \frac{1}{3}\lambda_{\min},$$

for any $n \geq n_1 \in \mathbb{N}$. For the symmetrizations, this yields

$$\left\|S_0^{(L)} - S^{(L)}\right\|_2 \leq \frac{1}{2}\left\|\hat{I}_0^{(L)}(\mathcal{X},\mathcal{X}) - I^{(L)}(\mathcal{X},\mathcal{X})\right\|_2 + \frac{1}{2}\left\|\hat{I}_0^{(L)}(\mathcal{X},\mathcal{X})^\intercal - I^{(L)}(\mathcal{X},\mathcal{X})^\intercal\right\|_2$$

$$= \left\|\hat{I}_0^{(L)}(\mathcal{X},\mathcal{X}) - I^{(L)}(\mathcal{X},\mathcal{X})\right\|_2 \leq \frac{1}{3}\lambda_{\min}. \tag{S71}$$

Furthermore, for $\theta_t \in B(\theta_0, C)$ it holds that

$$
\left\| \hat{I}_0^{(L)}(\mathcal{X}, \mathcal{X}) - \hat{I}_t^{(L)}(\mathcal{X}, \mathcal{X}) \right\|_2
$$
$$
= \left\| J^{(L),\sigma,\dot{\sigma}}(\mathcal{X}; \theta_0) \, J^{(L),\sigma,\tilde{\sigma}}(\mathcal{X}; \theta_0)^{\mathsf{T}} - J^{(L),\sigma,\dot{\sigma}}(\mathcal{X}; \theta_t) \, J^{(L),\sigma,\tilde{\sigma}}(\mathcal{X}; \theta_t)^{\mathsf{T}} \right\|_2
$$
$$
\leq \left\| J^{(L),\sigma,\dot{\sigma}}(\mathcal{X}; \theta_0) - J^{(L),\sigma,\dot{\sigma}}(\mathcal{X}; \theta_t) \right\|_2 \left\| J^{(L),\sigma,\tilde{\sigma}}(\mathcal{X}; \theta_0) \right\|_2
$$
$$
+ \left\| J^{(L),\sigma,\dot{\sigma}}(\mathcal{X}; \theta_t) \right\|_2 \left\| J^{(L),\sigma,\tilde{\sigma}}(\mathcal{X}; \theta_0) - J^{(L),\sigma,\tilde{\sigma}}(\mathcal{X}; \theta_t) \right\|_2
$$
$$
\leq \frac{2(K')^2}{\sqrt{n}} \left\| \theta_t - \theta_0 \right\|_2 \leq \frac{6(K')^2 K R_0}{\sqrt{n}}, \tag{S72}
$$

with probability at least $1 - \delta_1$ using Lemma E.6 as before. With the same calculations as for Inequality (S71), this implies for the symmetrizations that

$$
\left\| S_0^{(L)} - S_t^{(L)} \right\|_2 \leq \left\| \hat{I}_0^{(L)}(\mathcal{X}, \mathcal{X}) - \hat{I}_t^{(L)}(\mathcal{X}, \mathcal{X}) \right\|_2 \leq \frac{6(K')^2 K R_0}{\sqrt{n}} \leq \frac{1}{3}\lambda_{\min}, \tag{S73}
$$

for all $n \geq N := \max\left\{ m_0, m_1, \left( \frac{\lambda_{\min}}{18(K')^2 K R_0} \right)^2 \right\}$. Now Inequalities (S70), (S71) and (S73) give us

$$
\left\langle z, S_t^{(L)} z \right\rangle = \left\langle z, \left( S^{(L)} + S_0^{(L)} - S^{(L)} + S_t^{(L)} - S_0^{(L)} \right) z \right\rangle
$$
$$
= \left\langle z, S^{(L)} z \right\rangle + \left\langle z, \left( S_0^{(L)} - S^{(L)} \right) z \right\rangle + \left\langle z, \left( S_t^{(L)} - S_0^{(L)} \right) z \right\rangle
$$
$$
\overset{(S70)}{\geq} \lambda_{\min} \|z\|^2 - \left| \left\langle z, \left( S_0^{(L)} - S^{(L)} \right) z \right\rangle \right| - \left| \left\langle z, \left( S_t^{(L)} - S_0^{(L)} \right) z \right\rangle \right|
$$
$$
\overset{(S71)+(S73)}{\geq} \lambda_{\min} \|z\|^2 - \frac{1}{3}\lambda_{\min} \|z\|^2 - \frac{1}{3}\lambda_{\min} \|z\|^2 = \frac{1}{3}\left\langle z, \frac{1}{3}\lambda_{\min} I \, z \right\rangle,
$$

and thus imply Inequality (S70). To summarize, it holds with probability at least $1 - \delta_2 - \delta_3$ for any $n \geq N$ and $\theta_t \in B(\theta_0, C)$ that

$$
\|g(\theta_0)\|_2^2 \overset{(S68)}{\leq} R_0^2 \quad \text{and} \quad \frac{\mathrm{d}}{\mathrm{d}t} \|g(\theta_t)\|_2^2 \overset{(S69)}{\leq} -\frac{2}{3}\eta\lambda_{\min} \|g(\theta_t)\|_2^2.
$$

Grönwall's inequality now implies

$$
\|g(\theta_t)\|_2^2 \leq e^{-\frac{2}{3}\eta\lambda_{\min}t} \|g(\theta_0)\|_2^2 \leq e^{-\frac{2}{3}\eta\lambda_{\min}t} R_0^2.
$$

We can now return to Inequality (S67) to obtain with probability at least $1 - \delta_1 - \delta_2 - \delta_3$

$$
\frac{\mathrm{d}}{\mathrm{d}t} \|\theta_t - \theta_0\|_2 \overset{(S67)}{\leq} \eta K \|g(\theta_t)\|_2 \leq \eta K R_0 \, e^{-\frac{1}{3}\eta\lambda_{\min}t}.
$$

Integrating the inequality on both sides yields for all $\theta_t \in B(\theta_0, C)$

$$
\|\theta_t - \theta_0\|_2 \leq \frac{3 K R_0}{\lambda_{\min}} \left( 1 - e^{-\frac{1}{3}\eta\lambda_{\min}t} \right) < \frac{3 K R_0}{\lambda_{\min}} = C. \tag{S74}
$$

Let $t_1 := \inf\{t \colon \|\theta_t - \theta_0\|_2 < C\}$. Now, if $t_1 < \infty$, it holds

$$
C = \lim_{t \uparrow t_1} \|\theta_t - \theta_0\|_2 \overset{(S74)}{<} C.
$$

This is a contradiction, so we conclude that $t_1 = \infty$. In particular, Inequality (S74) holds for all $t > 0$. We can now repeat the calculations for Inequality (S72) with the Frobenius norm to finally obtain

$$
\left\| \hat{I}_0^{(L)}(\mathcal{X}, \mathcal{X}) - \hat{I}_t^{(L)}(\mathcal{X}, \mathcal{X}) \right\|_F \leq \frac{2K^2}{\sqrt{n}} \|\theta_t - \theta_0\|_2 \leq \frac{6K^3 R_0}{\lambda_{\min}} n^{-\frac{1}{2}}.
$$

Therefore, if we choose $\delta_1 = \delta_2 = \delta_3 = \frac{1}{3}\delta$, the desired inequality holds for any $n \geq N$ with probability at least $1 - \delta$. $\qquad\square$

**Lemma E.6** (Based on Lemma 1 and Lemma 2 from Lee et al. [2019]). *In the setting of Theorem E.5, for any $\delta > 0$ there exists a $K > 0$ such that for any $C > 0$, with probability at least $1 - \delta$ it holds*

$$\left\| J^{(L),\sigma,\hat{\sigma}}(\mathcal{X};\theta) - J^{(L),\sigma,\hat{\sigma}}(\mathcal{X};\tilde{\theta}) \right\|_F \leq \frac{K}{\sqrt{n}} \left\| \theta - \tilde{\theta} \right\|_2 \quad and$$

$$\left\| J^{(L),\sigma,\hat{\sigma}}(\mathcal{X};\theta) \right\|_F \leq K,$$

*for all $\theta, \tilde{\theta} \in B(\theta_0, C)$ and $\hat{\sigma} \in \{\dot{\sigma}, \tilde{\sigma}\}$. Due to the equivalence of matrix norms, the same inequalities hold for the norm $\|\cdot\|_2$ and some constant $K' > 0$.*

*Proof.* For $\hat{\sigma} = \dot{\sigma}$ and a different but equivalent parameterization of the network parameters [Lee et al., 2019, Chapter F], the proof can be found in Section G.2 of Lee et al. [2019]. The surrogate derivative $\tilde{\sigma}$ is bounded and Lipschitz continuous. Also, $J^{(L),\sigma,\tilde{\sigma}}(\mathcal{X};\theta)$ and $J^{(L),\sigma,\dot{\sigma}}(\mathcal{X};\theta) = J_\theta f(\mathcal{X};\theta)$ share the same recursive formula. Therefore, the proof, which builds on the so-called *Gaussian conditioning technique* [Yang, 2019a, Section E.1], should also hold for the surrogate derivative. $\square$

**Remark E.3** (The analytic generalized NTK for surrogate gradient learning). *Since in Theorem E.5 we consider only one activation function $\sigma$ instead of two different ones $\sigma_1, \sigma_2$ as in Theorem E.3 and E.4, it holds*

$$\Sigma_{1,2}^{(L)} = \Sigma_1^{(L)} = \Sigma_2^{(L)} = \Sigma^{(L)}.$$

*The analytic generalized NTK in the setting of Theorem E.4 is thus given by*

$$I^{(1)} = \Sigma^{(1)} \quad and \quad I^{(L+1)} = \Sigma^{(L+1)} + I^{(L)} \cdot \tilde{\Sigma}_{1,2}^{(L+1)}.$$

*We therefore still obtain an asymmetric kernel due to the contribution of $\tilde{\Sigma}_{1,2}^{(L)}$.*

**Remark E.4** (Positive definiteness of the generalized NTK). *In Theorem E.5 we require that the matrix $I^{(L)}(\mathcal{X}, \mathcal{X})$ be positive definite. Equivalently, its symmetrization,*

$$S^{(L)} = \frac{1}{2} \left( I^{(L)}(\mathcal{X}, \mathcal{X}) + I^{(L)}(\mathcal{X}, \mathcal{X})^\intercal \right),$$

*should be positive definite. For applications this can be checked numerically. More generally, it would be interesting to know whether the symmetric kernel given by*

$$S^{(L)}(x, x') = \frac{1}{2} \left( I^{(L)}(x, x') + I^{(L)}(x', x) \right)$$

*is positive definite. One could try to approach this question inductively. However, this leads to problems in the induction step. We want to present a different ansatz, which reduces the question to a question about $\tilde{\Sigma}_{1,2}^{(L)}$. We use the closed form of the analytic NTK and the symmetry of $\Sigma^{(l)}$ for all $l \in \mathbb{N}$ to see that*

$$S^{(L)}(x, x') = \frac{1}{2} \left( I^{(L)}(x, x') + I^{(L)}(x', x) \right)$$

$$= \frac{1}{2} \left( \left( \sum_{k=1}^{L} \Sigma^{(k)}(x, x') \cdot \prod_{l=k}^{L-1} \tilde{\Sigma}_{1,2}^{(l+1)}(x, x') \right) + \left( \sum_{k=1}^{L} \Sigma^{(k)}(x', x) \cdot \prod_{l=k}^{L-1} \tilde{\Sigma}_{1,2}^{(l+1)}(x', x) \right) \right)$$

$$= \sum_{k=1}^{L} \Sigma^{(k)}(x, x') \cdot \frac{1}{2} \left( \left( \prod_{l=k}^{L-1} \tilde{\Sigma}_{1,2}^{(l+1)}(x, x') \right) + \left( \prod_{l=k}^{L-1} \tilde{\Sigma}_{1,2}^{(l+1)}(x', x) \right) \right).$$

*This defines a symmetric positive definite kernel if $\Sigma^{(L)}$ is positive definite (see Jacot et al. [2018, Section A.4] for comparison) and the symmetrized kernels,*

$$\frac{1}{2} \left( \left( \prod_{l=k}^{L-1} \tilde{\Sigma}_{1,2}^{(l+1)}(x, x') \right) + \left( \prod_{l=k}^{L-1} \tilde{\Sigma}_{1,2}^{(l+1)}(x', x) \right) \right),$$

*are positive semi-definite for all $k = 1, \ldots, L - 1$.*

## E.2 The analytic NTK for surrogate gradient learning with sign activation function

We would like to proceed as in Section C.2.1 and replace the empirical generalized NTK in Equation (S49), the equation defining surrogate gradient learning, with the analytic generalized NTK obtained by Theorem E.5. Since the activation functions considered in the above section are Lipschitz continuous with bounded and Lipschitz continuous derivative, we will again approximate the sign function and its distributional derivative with the error function as in Section D. The results will not depend on the approximation of the weak derivative of the sign function. This approach can only lead to a useful result if the resulting kernel is not singular. We will check this in the following.

We choose activation function $\mathrm{erf}_m(z) = \mathrm{erf}(m \cdot z)$, $m \in \mathbb{N}$, with surrogate derivative $\tilde{\sigma}(z)$. We will also consider the special case $\tilde{\sigma}(z) = \dot{\mathrm{erf}}(z)$. As discussed in Remark E.3 and with the final results of Section D.1, this immediately yields

$$\Sigma_\infty^{(1)}(x,y) := \lim_{m \to \infty} \Sigma_m^{(1)}(x,y) \stackrel{(S36)}{=} \frac{\sigma_w^2}{n_0} \langle x,y \rangle + \sigma_b^2 \quad \text{and}$$

$$\Sigma_\infty^{(L+1)}(x,y) := \lim_{m \to \infty} \Sigma_m^{(L+1)}(x,y) \stackrel{(S29)}{=} \frac{2\sigma_w^2}{\pi} \arcsin\left(\frac{\Sigma_\infty^{(L)}(x,y)}{\sqrt{\Sigma_\infty^{(L)}(x,x)}\sqrt{\Sigma_\infty^{(L)}(y,y)}}\right) + \sigma_b^2.$$

$$\text{(S75)}$$

Here, we have to assume that $\sigma_b^2 > 0$ or $x, y \neq 0$ to ensure that $\Sigma_\infty^{(1)}(x,x), \Sigma_\infty^{(1)}(y,y) \neq 0$. This has already been discussed after Equation (S39).

### E.2.1 The derivative of the error function as surrogate derivative

Next, we want to calculate $\tilde{\Sigma}_{1,2;\infty}^{(L)}(x,y) := \lim_{m \to \infty} \tilde{\Sigma}_{1,2;m}^{(L)}(x,y)$. We will first consider the case $\tilde{\sigma} = \dot{\mathrm{erf}}$, for which we can use the already established tools. In particular, we will discuss the differences to the results of Section D.

It holds

$$\tilde{\Sigma}_{1,2;m}^{(L)}(x,y) = \sigma_w^2 \, \mathbb{E}[\dot{\mathrm{erf}}_m(Z_1^m)\,\dot{\mathrm{erf}}(Z_2^m)] \quad \text{for } L \geq 2,$$

where

$$(Z_1^m, Z_2^m) \sim \mathcal{N}\left(0, \begin{pmatrix} \Sigma_{1;m}^{(L-1)}(x,x) & \Sigma_{1,2;m}^{(L-1)}(x,y) \\ \Sigma_{1,2;m}^{(L-1)}(x,y) & \Sigma_{2;m}^{(L-1)}(y,y) \end{pmatrix}\right)$$

$$= \mathcal{N}\left(0, \begin{pmatrix} \Sigma_m^{(L-1)}(x,x) & \Sigma_m^{(L-1)}(x,y) \\ \Sigma_m^{(L-1)}(x,y) & \Sigma_m^{(L-1)}(y,y) \end{pmatrix}\right) = \mathcal{N}\left(0, \Sigma_{m;x,y}^{(L-1)}\right),$$

again using Remark E.3 to simplify the covariance matrix. The notation $\Sigma_{1,2;m}^{(L-1)}$, which comes from Theorem E.3 in combination with the scaling variable $m$ of the activation function, should not be confused with $\Sigma_{m;x,y}^{(L-1)}$, which is a shorthand notation for the Gram matrix $\Sigma_m^{(L-1)}(\{x,y\},\{x,y\})$.

We denote $e_1 = \left(\begin{smallmatrix} 1 \\ 0 \end{smallmatrix}\right)$, $e_2 = \left(\begin{smallmatrix} 0 \\ 1 \end{smallmatrix}\right)$ and $U = \left(\begin{smallmatrix} Z_1^m \\ Z_2^m \end{smallmatrix}\right)$. This yields

$$\tilde{\Sigma}_{1,2;m}^{(L)}(x,y) = \sigma_w^2\, \mathbb{E}[\dot{\mathrm{erf}}_m(Z_1^m)\, \dot{\mathrm{erf}}(Z_2^m)] = \sigma_w^2\, \mathbb{E}\left[ m\, \dot{\mathrm{erf}}\left((m \cdot e_1)^\intercal U\right)\, \dot{\mathrm{erf}}\left(e_2^\intercal U\right)\right]$$

$$\overset{(\star)}{=} \frac{4\sigma_w^2}{\pi} m \left( \left(1 + m^2 \cdot 2e_1^\intercal \Sigma_{m;x,y}^{(L-1)})e_1\right) \left(1 + 2e_2^\intercal \Sigma_{m;x,y}^{(L-1)})e_2\right) - \left(2m \cdot e_1^\intercal \Sigma_{m;x,y}^{(L-1)} e_2\right)^2 \right)^{-\frac{1}{2}}$$

$$= \frac{2\sigma_w^2}{\pi} \left( \frac{1}{4m^2} \left( \left(1 + 2m^2 \Sigma_m^{(L-1)}(x,x)\right) \left(1 + 2\Sigma_m^{(L-1)}(y,y)\right) - 4m^2 \Sigma_m^{(L-1)}(x,y)^2 \right) \right)^{-\frac{1}{2}}$$

$$= \frac{2\sigma_w^2}{\pi} \left( \left( \frac{1}{2m^2} + \Sigma_m^{(L-1)}(x,x)\right) \left( \frac{1}{2} + \Sigma_m^{(L-1)}(y,y)\right) - \Sigma_m^{(L-1)}(x,y)^2 \right)^{-\frac{1}{2}}$$

$$\xrightarrow{m \to \infty} \frac{2\sigma_w^2}{\pi} \left( \Sigma_\infty^{(L-1)}(x,x) \left( \frac{1}{2} + \Sigma_\infty^{(L-1)}(y,y)\right) - \Sigma_\infty^{(L-1)}(x,y)^2 \right)^{-\frac{1}{2}}$$

$$= \frac{2\sigma_w^2}{\pi} \left( \left| \Sigma_{\infty;x,y}^{(L-1)} \right| + \frac{1}{2} \Sigma_\infty^{(L-1)}(x,x) \right)^{-\frac{1}{2}}$$

$$= \begin{cases} \frac{2\sigma_w^2}{\pi} \left( \left| \Sigma_{\infty;x,y}^{(L-1)} \right| + \frac{\sigma_w^2 \|x\|^2}{2n_0} + \frac{\sigma_b^2}{2} \right)^{-\frac{1}{2}} & \text{for } L = 2, \\[2mm] \frac{2\sigma_w^2}{\pi} \left( \left| \Sigma_{\infty;x,y}^{(L-1)} \right| + \frac{\sigma_w^2 + \sigma_b^2}{2} \right)^{-\frac{1}{2}} & \text{for } L \geq 3. \end{cases}$$

$$=: \tilde{\Sigma}_{1,2;\infty}^{(L)}(x,y), \tag{S76}$$

using Lemma D.1 in Equation $(\star)$. For the penultimate equality we used Equation (S36) and Equation (S37). Compared to Equation (S41),

$$\dot{\Sigma}_m^{(L)}(x,y) \xrightarrow{m \to \infty} \frac{2\sigma_w^2}{\pi} \left( \Sigma_\infty^{(L-1)}(x,x) \cdot \Sigma_\infty^{(L-1)}(y,y) - \Sigma_\infty^{(L-1)}(x,y)^2 \right) = \frac{2\sigma_w^2}{\pi} \left| \Sigma_{\infty;x,y}^{(L-1)} \right|^{-\frac{1}{2}}, \tag{S41}$$

which in fact holds for both $L = 2$ and $L \geq 3$, an additional term appeared in Equation (S76). It always holds $\sigma_w^2 + \sigma_b^2 > 0$ and it holds $\sigma_w^2 \|x\|^2/n_0 + \sigma_b^2 > 0$ if $\sigma_b^2 > 0$ or $x \neq 0$. As discussed earlier, we always assume that $x \neq 0$ is satisfied. It follows that this asymmetric NTK is not singular, since

$$\tilde{\Sigma}_{1,2;\infty}^{(L)}(x,x) = \frac{2\sigma_w^2}{\pi} \left( \left| \Sigma_{\infty;x,x}^{(L-1)} \right| + \frac{1}{2} \Sigma_\infty^{(L-1)}(x,x) \right)^{-\frac{1}{2}}$$

$$= \frac{2\sigma_w^2}{\pi} \left( 0 + \frac{1}{2} \Sigma_\infty^{(L-1)}(x,x) \right)^{-\frac{1}{2}} = \sqrt{2}\, \frac{2\sigma_w^2}{\pi} \left( \Sigma_\infty^{(L-1)}(x,x) \right)^{-\frac{1}{2}} \in \mathbb{R}. \tag{S77}$$

Note that this is reminiscent of the constant factor depending on $x$ in the asymptotics of $\dot{\Sigma}_m^{(L)}(x,x)$ as $m \to \infty$, given by (S42) and (S43),

$$\dot{\Sigma}_m^{(L)}(x,x) \sim \frac{2\sigma_w^2}{\pi} m \left( \Sigma_\infty^{(L-1)}(x,x) \right)^{-\frac{1}{2}}.$$

According to Remark E.3 it holds

$$I_m^{(L)}(x,y) = \Sigma_m^{(L)}(x,y) + I_m^{(L-1)}(x,y) \cdot \tilde{\Sigma}_{1,2;m}(x,y).$$

Since $\Sigma_m^{(L)}(x,y)$ and $\tilde{\Sigma}_{1,2;m}^{(L)}$ are continuous functions of the entries of the matrix $\Sigma_{m;x,y}^{(L-1)}$, it follows by induction using Equations (S75) and (S76) that the limit of the analytic NTK is well defined for $m \to \infty$. We can write

$$I_\infty^{(L)}(x,y) = \lim_{m \to \infty} I_m^{(L)}(x,y).$$

Thus, by approximating the sign function with error functions, we found that the analytic NTK for surrogate gradient learning with sign function and error function as surrogate derivative is well-defined and non-singular as a kernel on $\mathbb{R}^{n_0} \times \mathbb{R}^{n_0}$. Furthermore, comparing this NTK with the

NTK we derived in Section D, the term $\Sigma_\infty^{(L)}$ does not change. This is a direct consequence of not replacing the activation function with a surrogate activation function. In this sense, we inherit more properties with this approach than by replacing not only the derivative but the entire activation function including its derivative with a surrogate. Comparing Equation (S41) with Equation (S76), we see that $\tilde{\Sigma}_{1,2;\infty}^{(L)}(x,y)$ can be obtained from $\dot{\Sigma}_\infty^{(L)}(x,y)$ by adding a regularizing term depending on $x$, $\Sigma_\infty^{(L-1)}(x,x)/2$.

### E.2.2 General surrogate derivative

Now we turn to the general case with a general surrogate derivative $\tilde{\sigma}$. Similar to before, for $m \in \mathbb{N} \cup \{\infty\}$ we denote

$$(Z_1^m, Z_2^m) \sim \mathcal{N}\left(0, \begin{pmatrix} \Sigma_{1;m}^{(L-1)}(x,x) & \Sigma_{1,2;m}^{(L-1)}(x,y) \\ \Sigma_{1,2;m}^{(L-1)}(x,y) & \Sigma_{2;m}^{(L-1)}(y,y) \end{pmatrix}\right)$$
$$= \mathcal{N}\left(0, \begin{pmatrix} \Sigma_m^{(L-1)}(x,x) & \Sigma_m^{(L-1)}(x,y) \\ \Sigma_m^{(L-1)}(x,y) & \Sigma_m^{(L-1)}(y,y) \end{pmatrix}\right) = \mathcal{N}\left(0, \Sigma_{m;x,y}^{(L-1)}\right).$$

With $u = \begin{pmatrix} z_1 \\ z_2 \end{pmatrix}$ and for invertible $\Sigma = \Sigma_{\infty;x,y}^{(L-1)}$, it holds for $L \geq 2$

$$\tilde{\Sigma}_{1,2;\infty}^{(L)}(x,y) = \lim_{m\to\infty} \tilde{\Sigma}_{1,2;m}^{(L)}(x,y) = \lim_{m\to\infty} \sigma_w^2 \, \mathbb{E}\left[\dot{\mathrm{erf}}_m(Z_1^m)\,\tilde{\sigma}(Z_2^m)\right]$$

$$\overset{(\star)}{=} \sigma_w^2 \, \mathbb{E}\left[2\delta_0(Z_1^\infty)\,\tilde{\sigma}(Z_2^\infty)\right] = 2\sigma_w^2 \int_{\mathbb{R}^2} \frac{1}{2\pi} |\Sigma|^{-\frac{1}{2}} \delta_0(z_1) \cdot \tilde{\sigma}(z_2) \cdot e^{-\frac{1}{2}u^\mathsf{T}\Sigma^{-1}u}\, \mathrm{d}u$$

$$= \sigma_w^2 \sqrt{\frac{2}{\pi}} \, \Sigma_{1,1}^{-\frac{1}{2}} \int_{\mathbb{R}} \frac{1}{\sqrt{2\pi}} \sqrt{\frac{\Sigma_{1,1}}{|\Sigma|}} \cdot \tilde{\sigma}(z_2) \cdot e^{-\frac{1}{2}\frac{\Sigma_{1,1}}{|\Sigma|}z_2^2}\, \mathrm{d}z_2$$

$$= \sigma_w^2 \sqrt{\frac{2}{\pi}} \left(\Sigma_\infty^{(L-1)}(x,x)\right)^{-\frac{1}{2}} \mathbb{E}_{Y\sim\mathcal{N}\left(0,\left|\Sigma_{\infty;x,y}^{(L-1)}\right|/\Sigma_\infty^{(L-1)}(x,x)\right)}\left[\tilde{\sigma}(Y)\right], \tag{S78}$$

where Equation $(\star)$ seems natural, but requires further reasoning. We will prove the above rigorously in Lemma E.7. For now, we assume that the equality holds. Since $\tilde{\sigma}$ is bounded and continuous, this yields

$$\lim_{x\to y} \tilde{\Sigma}_{1,2;\infty}^{(L)}(x,y)$$

$$= \lim_{\left|\Sigma_{\infty;x,y}^{(L-1)}\right|\to 0} \sigma_w^2 \sqrt{\frac{2}{\pi}} \left(\Sigma_\infty^{(L-1)}(x,x)\right)^{-\frac{1}{2}} \mathbb{E}_{Y\sim\mathcal{N}\left(0,\left|\Sigma_{\infty;x,y}^{(L-1)}\right|/\Sigma_\infty^{(L-1)}(x,x)\right)}\left[\tilde{\sigma}(Y)\right]$$

$$= \sigma_w^2 \sqrt{\frac{2}{\pi}} \left(\Sigma_\infty^{(L-1)}(x,x)\right)^{-\frac{1}{2}} \tilde{\sigma}(0).$$

This agrees with the fact that

$$\tilde{\Sigma}_{1,2;\infty}^{(L)}(x,x) = \lim_{m\to\infty} \sigma_w^2 \, \mathbb{E}\left[\dot{\mathrm{erf}}_m(Z_1^m)\,\tilde{\sigma}(Z_1^m)\right] = \sigma_w^2 \, \mathbb{E}\left[2\delta_0(Z_1^\infty)\,\tilde{\sigma}(Z_1^\infty)\right] \tag{S79}$$

$$= 2\sigma_w^2 \frac{1}{\sqrt{2\pi}} \left(\Sigma_\infty^{(L-1)}(x,x)\right)^{-\frac{1}{2}} \tilde{\sigma}(0) = \sigma_w^2 \sqrt{\frac{2}{\pi}} \left(\Sigma_\infty^{(L-1)}(x,x)\right)^{-\frac{1}{2}} \tilde{\sigma}(0),$$

where we again assumed an equality very similar to Equation $(\star)$. It is easy to check that Equations (S76) and (S77) can be recovered by inserting $\tilde{\sigma} = \dot{\mathrm{erf}}$ into the derived formulas.

We now prove the missing part of Equation (S78).

**Lemma E.7.** *Let $\tilde{\sigma}$ be a bounded and continuous function and $((Z_1^m, Z_2^m))_{m\in\mathbb{N}}$ a sequence of random variables, $(Z_1^m, Z_2^m) \sim \mathcal{N}(0, \Sigma^m)$. If the covariance matrices are invertible and converge to an invertible matrix, $\Sigma^m \to \Sigma^\infty \in \mathbb{R}^{2\times 2}$ as $m \to \infty$, then it holds*

$$\lim_{m\to\infty} \mathbb{E}\left[\dot{\mathrm{erf}}_m(Z_1^m)\,\tilde{\sigma}(Z_2^m)\right] = \sqrt{\frac{2}{\pi}} \left(\Sigma_{1,1}^\infty\right)^{-\frac{1}{2}} \mathbb{E}_{Y\sim\mathcal{N}\left(0,\left|\Sigma^\infty\right|/\Sigma_{1,1}^\infty\right)}\left[\tilde{\sigma}(Y)\right]. \tag{S80}$$

If $Z_1^m = Z_2^m$ for all $m \in \mathbb{N}$ so that the covariance matrices are given by $\Sigma^m = \left( \begin{smallmatrix} \Sigma_1^m & \Sigma_1^m \\ \Sigma_1^m & \Sigma_1^m \end{smallmatrix} \right)$, and if $\Sigma_1^m \to \Sigma_1^\infty \neq 0$ as $m \to \infty$, then it holds

$$\lim_{m\to\infty} \mathbb{E}\left[ \dot{\mathrm{erf}}_m(Z_1^m)\, \tilde{\sigma}(Z_2^m) \right] = \sqrt{\frac{2}{\pi}} \, (\Sigma_1^\infty)^{-\frac{1}{2}} \, \tilde{\sigma}(0). \tag{S81}$$

*Proof.* We begin with the case of invertible covariance matrices. Again denoting $u = \left( \begin{smallmatrix} z_1 \\ z_2 \end{smallmatrix} \right)$, it holds by assumption

$$\mathbb{E}\left[ \dot{\mathrm{erf}}_m(Z_1^m)\, \tilde{\sigma}(Z_2^m) \right] = \int_{\mathbb{R}^2} \frac{1}{2\pi} |\Sigma^m|^{-\frac{1}{2}} \cdot \tilde{\sigma}(z_2) \cdot \frac{2m}{\sqrt{\pi}} e^{-m^2 z_1^2} \cdot e^{-\frac{1}{2} u^\mathsf{T} (\Sigma^m)^{-1} u} \, du$$

$$= \int_{\mathbb{R}^2} \frac{1}{2\pi} |\Sigma^m|^{-\frac{1}{2}} \cdot \tilde{\sigma}(z_2) \cdot \frac{2}{\sqrt{\pi}} \cdot e^{-\frac{1}{2}\left( \left( \begin{smallmatrix} z_1/m \\ z_2 \end{smallmatrix} \right)^\mathsf{T} (\Sigma^m)^{-1} \left( \begin{smallmatrix} z_1/m \\ z_2 \end{smallmatrix} \right) + 2 u^\mathsf{T} e_1 e_1^\mathsf{T} u \right)} \, du$$

$$= \int_{\mathbb{R}^2} \frac{1}{2\pi} |\Sigma^m|^{-\frac{1}{2}} \cdot \tilde{\sigma}(z_2) \cdot \frac{2}{\sqrt{\pi}} \cdot e^{-\frac{1}{2} u^\mathsf{T} B^m u} \, du, \tag{S82}$$

for

$$B^m := \frac{1}{|\Sigma^m|} \begin{pmatrix} \Sigma_{2,2}^m/m^2 & -\Sigma_{1,2}^m/m \\ -\Sigma_{1,2}^m/m & \Sigma_{1,1}^m \end{pmatrix} + \begin{pmatrix} 2 & 0 \\ 0 & 0 \end{pmatrix} \xrightarrow{m\to\infty} \begin{pmatrix} 2 & 0 \\ 0 & \Sigma_{1,1}^\infty/|\Sigma^\infty| \end{pmatrix} =: B^\infty. \tag{S83}$$

The determinant of $B^m$ is given by

$$|B^m| = \left( \frac{\Sigma_{2,2}^m}{m^2 |\Sigma^m|} + 2 \right) \frac{\Sigma_{1,1}^m}{|\Sigma^m|} - \frac{\left( -\Sigma_{1,2}^m \right)^2}{m^2 |\Sigma^m|^2} = \frac{1}{m^2 |\Sigma^m|} + 2 \frac{\Sigma_{1,1}^m}{|\Sigma^m|}.$$

This now yields

$$(S82) = \frac{2}{\sqrt{\pi}} |\Sigma^m|^{-\frac{1}{2}} \left| (B^m)^{-1} \right|^{\frac{1}{2}} \int_{\mathbb{R}^2} \frac{1}{2\pi} \left| (B^m)^{-1} \right|^{-\frac{1}{2}} \cdot \tilde{\sigma}(z_2) \cdot e^{-\frac{1}{2} u^\mathsf{T} B^m u} \, du$$

$$= \frac{2}{\sqrt{\pi}} |\Sigma^m|^{-\frac{1}{2}} |B^m|^{-\frac{1}{2}} \mathbb{E}_{(Y_1^m, Y_2^m) \sim \mathcal{N}\left(0, (B^m)^{-1}\right)} [\tilde{\sigma}(Y_2^m)].$$

The continuity of matrix inversion implies $(B^m)^{-1} \to (B^\infty)^{-1}$ as $m \to \infty$. The characteristic functions of finite-dimensional Gaussian random variables are fully defined by their means and covariance matrices. Thus, the convergence of the covariance matrices implies convergence of the characteristic functions, which in turn implies convergence in distribution. Since $\tilde{\sigma}$ is continuous and bounded and the determinant is continuous, we obtain

$$\frac{2}{\sqrt{\pi}} |\Sigma^m|^{-\frac{1}{2}} |B^m|^{-\frac{1}{2}} \mathbb{E}_{(Y_1^m, Y_2^m) \sim \mathcal{N}\left(0, (B^m)^{-1}\right)} [\tilde{\sigma}(Y_2^m)]$$

$$\xrightarrow{m\to\infty} \frac{2}{\sqrt{\pi}} |\Sigma^\infty|^{-\frac{1}{2}} |B^\infty|^{-\frac{1}{2}} \mathbb{E}_{(Y_1^\infty, Y_2^\infty) \sim \mathcal{N}\left(0, (B^\infty)^{-1}\right)} [\tilde{\sigma}(Y_2^\infty)]$$

$$\overset{(S83)}{=} \frac{2}{\sqrt{\pi}} |\Sigma^\infty|^{-\frac{1}{2}} \left( \frac{2\Sigma_{1,1}^\infty}{|\Sigma^\infty|} \right)^{-\frac{1}{2}} \mathbb{E}_{Y \sim \mathcal{N}\left(0, |\Sigma^\infty|/\Sigma_{1,1}^\infty\right)} [\tilde{\sigma}(Y)]$$

$$= \sqrt{\frac{2}{\pi}} \, (\Sigma_{1,1}^\infty)^{-\frac{1}{2}} \mathbb{E}_{Y \sim \mathcal{N}\left(0, |\Sigma^\infty|/\Sigma_{1,1}^\infty\right)} [\tilde{\sigma}(Y)].$$

This proves Equation (S80). In the case $Z_1^m = Z_2^m$, we have

$$
\mathbb{E}\left[\dot{\mathrm{erf}}_m(Z_1^m)\,\tilde{\sigma}(Z_2^m)\right] = \mathbb{E}_{Y\sim\mathcal{N}(0,\Sigma_1^m)}[\dot{\mathrm{erf}}_m(Y)\,\tilde{\sigma}(Y)]
$$

$$
= \int_{\mathbb{R}} \frac{1}{\sqrt{2\pi}}\frac{1}{\sqrt{\Sigma_1^m}}\frac{2}{\sqrt{\pi}}m\cdot e^{-m^2 y^2}\cdot\tilde{\sigma}(y)\cdot e^{-\frac{1}{2}\frac{y^2}{\Sigma_1^m}}\,\mathrm{d}y
$$

$$
= \int_{\mathbb{R}} \frac{1}{\sqrt{2\pi}}\frac{1}{\sqrt{\Sigma_1^m}}\frac{2}{\sqrt{\pi}}m\cdot\tilde{\sigma}(y)\cdot e^{-\frac{1}{2}y^2\left(1/\Sigma_1^m+2m^2\right)}\,\mathrm{d}y
$$

$$
= \frac{2}{\sqrt{\pi}}\frac{1}{\sqrt{\Sigma_1^m}}\frac{m}{\sqrt{1/\Sigma_1^m+2m^2}}\int_{\mathbb{R}}\frac{1}{\sqrt{2\pi}}(1/\Sigma_1^m+2m^2)^{\frac{1}{2}}\cdot\tilde{\sigma}(y)\cdot e^{-\frac{1}{2}y^2\left(1/\Sigma_1^m+2m^2\right)}\,\mathrm{d}y
$$

$$
= \frac{2}{\sqrt{\pi}}\frac{1}{\sqrt{1/m^2+2\Sigma_1^m}}\mathbb{E}_{Y\sim\mathcal{N}\left(0,(1/\Sigma_1^m+2m^2)^{-1}\right)}[\tilde{\sigma}(Y)]
$$

$$
\xrightarrow{m\to\infty} \frac{2}{\sqrt{\pi}}\frac{1}{\sqrt{2\Sigma_1^\infty}}\tilde{\sigma}(0) = \sqrt{\frac{2}{\pi}}(\Sigma_1^\infty)^{-\frac{1}{2}}\tilde{\sigma}(0),
$$

again using the boundedness and continuity of $\tilde{\sigma}$ in the last line. This proves Equation (S81). $\qquad\square$

With $\Sigma^m = \Sigma_{m;x,y}^{(L-1)}$ the lemma yields

$$
\tilde{\Sigma}_{1,2;\infty}^{(L)}(x,y) = \lim_{m\to\infty}\tilde{\Sigma}_{1,2;m}^{(L)}(x,y) = \lim_{m\to\infty}\sigma_w^2\,\mathbb{E}\left[\dot{\mathrm{erf}}_m(Z_1^m)\,\tilde{\sigma}(Z_2^m)\right]
$$

$$
= \sigma_w^2\sqrt{\frac{2}{\pi}}\left(\Sigma_\infty^{(L-1)}(x,x)\right)^{-\frac{1}{2}}\mathbb{E}_{Y\sim\mathcal{N}\left(0,\left|\Sigma_{\infty;x,y}^{(L-1)}\right|/\Sigma_\infty^{(L-1)}(x,x)\right)}[\tilde{\sigma}(Y)].
$$

In conclusion, we found that the analytic NTK for surrogate gradient learning with sign function is well-defined and non-singular for any bounded and Lipschitz continuous surrogate derivative. As in the special case of the error function, the term $\tilde{\Sigma}_{1,2;\infty}^{(L)}(x,y)$ can be expressed in terms of $\Sigma_\infty^{(L-1)}(x,x)$ and the determinant $\left|\Sigma_{\infty;x,y}^{(L-1)}\right|$.

