# OpenReview forum: "A generalized neural tangent kernel for surrogate gradient learning"
_NeurIPS.cc/2024/Conference — NeurIPS 2024 spotlight_

### Official Review · Reviewer_TYqA · 2024-07-01

**Soundness:** 3
**Presentation:** 3
**Contribution:** 3
**Rating:** 6
**Confidence:** 3

**Summary:**

The paper addresses the challenge of applying gradient-based training methods to neural networks with non-differentiable activation functions, such as binary and spiking neural networks. These networks use surrogate derivatives to enable gradient descent, but this approach lacks theoretical foundation. The authors propose a generalization of the NTK, called the surrogate gradient NTK (SG-NTK), to provide a rigorous theoretical basis for SGL.

**Strengths:**

1. Novel extension of NTK to non-differentiable activation functions using surrogate gradients using the results in [1,2,3,4].
2. Strong empirical validation through numerical experiments.

[1] Gaussian process behavior in wide deep neural networks
[2] Neural tangent kernel: Convergence and generalization in neural networks.
[3] Wide and deep neural networks achieve consistency for classification
[4] Wide neural networks of any depth evolve as linear models under gradient descent

**Weaknesses:**

1. The complexity of the theoretical framework may limit accessibility for practitioners
2. Additional experiments on real datasets (MNIST, CIFAR10, ImageNet) could further validate the approach.

**Questions:**

What is the application of the results? The activation function, the ReLU function is commonly used and I believe it can be solved in previous works. Are there any commonly used activation functions that can solved in your paper but not in previous works?

**Limitations:**

The results in this paper are only applicable to sufficiently wide networks with random initialization.

---

> ### Author Rebuttal · Authors · 2024-08-06
>
> Thank you for your review and for your helpful suggestions. We appreciate the focus on the applicability of our work.
>
> - While the mathematical details behind the NTK theory are generally complex and not particularly accessible to practitioners, the analytic NTK can be easily calculated for different data sets, network depths and activation functions, e.g., using the Neural tangents package (see citations in line 843). The analytic NTK then shows how the class of functions learned with gradient descent looks like, without actually having to learn a network. As we write in lines 22 to 23, this is possible whenever the activation function is differentiable or at least semi-differentiable. This also includes the case of the ReLU activation function. However, whenever gradient descent is not applicable, we cannot define the NTK anymore. This includes binary neural networks, time-discrete spiking neural networks and any activation function with jumps. Surrogate gradient learning is then commonly used by practitioners, see lines 102 to 120. Surrogate gradient learning introduces the surrogate derivative, which is an additional unknown (besides network depth and activation function) that changes the class of functions learned with SGL. So far, there is no theoretical tool like the SG-NTK to systematically choose the surrogate derivative and the network depth. In particular, the SG-NTK yields a prediction for the class of functions that are learned with SGL.
> - We agree that our approach would benefit from additional experiments. Moreover, as we write in lines 338 to 339, "[a more] rigorous analysis should be carried out on how the connection between SGL and the SG-NTK carries over to activation functions with jumps, as shown by our simulations".

---

### Official Review · Reviewer_Ljsu · 2024-07-04

**Soundness:** 4
**Presentation:** 3
**Contribution:** 3
**Rating:** 8
**Confidence:** 3

**Summary:**

The paper considers an extension of neural tangent kernel methods for the analysis of training of neural networks with non-differentiable activations with surrogate gradient learning.  The basic approach is the define a generalized NTK (the SG-NTK) based on the quasi-Jacobian matrix (that is, the Jacobian constructed using the surrogate gradients) rather than the (ill-defined) Jacobian based on the actual gradients of the activations.  This construct is shown to be deterministic (distributionally determined) in the infinite width limit.  For the sign activation (with erfm acting as surrogate gradient) both the NTK and SG-NTK are derived.  Finally some experimental results are given that show that the distribution of networks trained with SG-NTK matches reality.

**Strengths:**

- The paper is clearly presented.
- Motivation, aims etc are clear and compelling.
- The paper appears to fills an important gap in existing literature (admittedly I am relatively new to this area of research so there may be predecessors I am unaware of).
- The derivations given seem correct to the best of my understanding, though I did skim some of the proofs in the rather long appendix.
- The experimental results would appear to confirm that the SG-NTK matches with reality.

**Weaknesses:**

- it would perhaps be more useful to present figure 1 with a log y scale as the linear scale used, combined with the divergence at $\Delta \alpha = 0$, tends to flatten all features of the NTK, masking the differences between the NTKs away from the point of divergence.
- also regarding figure 1 is it feasible to plot the limiting case $m \to \infty$ with the point of divergence elided?

**Questions:**

See previous sections.

**Limitations:**

The authors have addressed limitations adequately and there appear to be no obvious negative societal impacts.

---

> ### Author Rebuttal · Authors · 2024-08-06
>
> Thank you for your encouraging and positive review and for your helpful suggestions.
>
> - We have refrained from using logarithmic scales to keep the plot as simple as possible and to facilitate the comparison between Figure 1 and Figure 2. However, we agree that a logarithmic y-scale helps to illustrate the divergence at $\Delta \alpha = 0$ and the convergence at $\Delta \alpha \not = 0$ in Figure 1, and provide an updated figure with a quasi-logarithmic y-axis using the inverse hyperbolic sine (see global response PDF, Figure R2).
> - Yes, we agree with this suggestion and add the analytic NTK for $m \to \infty$ as in Figure 2 (see global response PDF, Figure R2). Note that this singular kernel, $\Theta_\mathrm{sign}$, can never be fully plotted with a logarithmic y-axis, since $\Theta_\mathrm{sign}(\Delta \alpha) \to \infty$ as $\Delta \alpha \to 0$.

---

> > ### Comment · Reviewer_Ljsu · 2024-08-08
> >
> > Thank you for the response, I'm happy to keep my score of 8.

---

> > > ### Author Response · Authors · 2024-08-12
> > >
> > > Thank you for your quick response, we are glad that we were able to answer your questions.

---

### Official Review · Reviewer_L5ST · 2024-07-12

**Soundness:** 4
**Presentation:** 3
**Contribution:** 4
**Rating:** 8
**Confidence:** 3

**Summary:**

This paper explore the neural tangent kernel (NTK) with regards to surrogate gradient learning for non-differentiable activation functions. The authors show that the standard neural tangent kernel is not equipped to deal with such activation functions and causes the kernel function to become singular. They provide a derivation of a surrogate derivative based NTK that can handle such activations thus further generalizing the NTK for a larger subset of networks.

**Strengths:**

This paper provides a brand new derivation of the NTK for non-smooth ANN activations and thus is very original. The overall paper is of high quality and clearly developed. The results of this paper are of high significance and allow for new avenues for application of NTK analysis.

**Weaknesses:**

While the paper is well developed overall, there are a few spots that can benefit from additional clarity regarding notation:

Line 144 (Definition 2.1) - $r_l(m)$ refers to the number of neurons for a particular layer however $m$ is not directly described. My understanding is that it is referring to the output size dimension but this is not immediately clear.

Line 215 - I am assuming that $\delta(z)$ refers to the Dirac delta function (delta distribution).

Line 244 - Here we have $\delta_{ij}$ which I assume is *not* the Diract delta and instead a constant for a given kernel matrix entry.

Line 289 - Since we are talking about divergence, notationally it would be better to use $\Theta_{\text{erf}_m} \not\rightarrow \Theta\_{\text{sign}}$ or specify the formal definition of divergence *or* avoid it all together.

In regard to the experiments line 303 mentions that the NTK diverge as $m \rightarrow \infty$, however this does not seem to be clear from the plots especially since $m=20$ does not seem to be sufficient to show the divergence occurring. In addition, since we are discussing divergence, I believe the paper could benefit by including the average error values between the analytic and empirical kernels since graphs tend to do a poor job of illustrating this.

**Questions:**

Are $\delta(z)$ and $\delta_{ij}$ the same?

**Limitations:**

Everything is adequately addressed.

---

> ### Author Rebuttal · Authors · 2024-08-06
>
> Thank you for your positive and thorough review and for your helpful suggestions.
> - We apologise for any confusion caused by the use of the variable $m$ in lines 133 to 137 and will change the variable. The parameter $m$ is used in Definition 2.1 to be able to consider a limit by taking $m \to \infty$. It does not refer to any dimension or size of the network itself, but rather indexes the limit. This is described directly after the Definition 2.1 in lines 148 to 149: "Every element of $\mathcal{R}_L$ provides a way to take the widths of the hidden layers to infinity by setting $n_l = r_l(m)$ for any $1 \leq l < L$ and considering $m \to \infty$". Example: We have two hidden layers and hence layer widths $n_0, n_1, n_2, n_3$. Suppose that $n_0 = n_3 = 2$ are fixed, layer one grows quadratically, and layer two grows cubically. Then we have $r_1(m) = m^2$ and $r_2(m) = m^3$.
> - Yes, $\delta(z)$ denotes the delta distribution. We will add this information right after the first occurrence of the delta distribution.
> - Yes, $\delta_{ij}$ denotes the Kronecker delta, i.e., $\delta_{ij} = 1 $ if $i=j$ and $\delta_{ij} = 0$ otherwise. To avoid confusion with the delta distribution, we will add a clarification after the first occurrence of the Kronecker delta.
> - It is true that $\Theta_ {\mathrm{erf}_ m}$ converges to the singular kernel $\Theta_\mathrm{sign}$, $\Theta_ {\mathrm{erf}_ m} \to \Theta_\mathrm{sign}$, because this is how $\Theta_\mathrm{sign}$ is defined in lines 221 to 222. This means that $\Theta_ {\mathrm{erf}_ m}(x,y) \to \infty$ as $m\to\infty$ for $x = y$. To be more precise, we will instead write in line 289, "We numerically illustrate the divergence of the analytic NTK, $\Theta_ {\mathrm{erf}_ m}$, as $m \to \infty$, [...]".
> - It is unclear in which way the plots do not sufficiently illustrate the divergence. Comparing the y-axis and the size of the peak at $\Delta \alpha = 0$ for $m=1,5,20$, the effect of the divergence can be seen, as the peak grows from $<5$ for $m=2$ to $>100$ for $m=20$. Figures can never fully confirm or disprove convergences or divergences and we will weaken the formulation in lines 302 to 303, "the plots confirm that the analytic NTKs diverge". In addition, we introduce a common quasi-logarithmic y-axis using the inverse hyperbolic sine to better illustrate the convergence for all $\Delta\alpha \not= 0$ (see global response PDF, Figure R2), as suggested by Reviewer Ljsu. By comparing the analytic NTKs for finite $m$ with the singular kernel for $m \to \infty$ in Figure R2, we can indeed see that $m=20$ is large enough to illustrate the convergence to the singular kernel. Finally, we agree that average error values help to illustrate the convergence of the empirical kernels to the analytic kernels and provide them in Figure R3 and Figure R4 (see global response PDF).

---

> > ### Comment · Reviewer_L5ST · 2024-08-08
> >
> > Thank you for your rebuttal as well as your clarifications and additions to your manuscript. I read through the other reviews and your rebuttals as well. Given the additional changes and the clarity of your presentation, I am comfortable to bump up my score to an 8.

---

> > > ### Author Response · Authors · 2024-08-12
> > >
> > > Thank you for taking the time to consider the other reviews and rebuttals. We are glad that we were able to answer your questions and we appreciate the score improvement.

---

### Official Review · Reviewer_PShm · 2024-07-12

**Soundness:** 3
**Presentation:** 3
**Contribution:** 3
**Rating:** 8
**Confidence:** 3

**Summary:**

The paper adapts the neural tangent kernel framework to surrogate gradient learning (and so to learning in spiking neural networks).

**Strengths:**

- The paper generalizes NTK to (some algorithms for) spiking neural networks, which is an important scenario for neuroscience and neuromorphic computing
- Like the original NTK results, analyzing learning dynamics with this approach seems easy (i.e. it works like kernel regression, Eq. 7)
- The paper is well-written and self-contained

Overall, I think this paper is important for theoretical analysis of spiking neural networks, and contains novel results. I didn't closely follow the derivations in the appendix, but I'm familiar with the NTK literature/proofs, and the approach of this paper seems reasonable to me (+ it follows the original derivations to some extent).

**Weaknesses:**

The main weakness is the NTK approach itself: NTK is known to be a poor approximation to learning in deep networks in many cases, which limits what kind of conclusions we can make with NTK-driven theoretical analysis. However, this weakness is not specific to this paper.

**Questions:**

- I don’t see how definition 2.2 iii can work for strictly increasing width functions. Sequential limits mean some of the widths stay fixed, right?
- Fig.1 should have the black line at the back I think, otherwise it’s not clear if the peaks always overlap

**Limitations:**

The authors have addressed the limitations of the work.

---

> ### Author Rebuttal · Authors · 2024-08-06
>
> Thank you for your encouraging, thorough and detailed review and for your helpful suggestions.
>
> - **Short answer:** Indeed, by parameterizing all hidden layer widths with the parameter $m$ and considering $m \to \infty$, we cannot cover the sequential infinite-width limit as described in [1]. This is because the parameterization requires all hidden layer widths to be finite during the entire limit procedure, which is not the case for the sequential limit. The fact that the width functions are strictly increasing is not creating this problem. However, by following the inductive proof of the sequential limit, we construct a parameterized limit using the width functions (see next paragraph). A consequence of this proof technique is that we have no control over the rates at which the hidden layer widths diverge. However, this is not a problem if one wants to prove weak convergence as defined in Definition 2.2 (iii), see also Section E.1, Lemma E.1 and E.2. This is why we wrote "In practice, this means that the statement holds as $n_  1,\dots,n_L \to \infty$ sequentially". We agree that this wording is not very clear and we will add a more elaborate clarification.
> **Connection between the sequential infinite-width limit and our notion of weak convergence:** The simplest form of a sequential limit takes the form $\lim_{n_1 \to \infty} \left( \lim_{n_2 \to \infty} f(n_1,n_2) \right)$. Let us assume that $\lim_{n_2 \to \infty} f(n_1,n_2) = \hat{f}(n_1)$ exists for all $n_1$ and that $\lim_{n_1 \to \infty} \hat{f}(n_1) = a$ also exists. If $f$ is continuous, one can always find a parameterization $n_1(m), n_2(m)$, such that $\lim_{m \to\infty} f(n_1(m),n_2(m)) = a$. Now, the sequential limits in [1] correspond to the limit of the form $\lim_{n_1 \to \infty} \left( \lim_{n_2 \to \infty} f(n_1,n_2) \right)$. Following this analogy, we provide a way to find the parametrization $n_1(m), n_2(m)$ with Lemma E.1. This allows us to show convergence in the weak sense as defined in Definition 2.2 (iii).
> **Why parameterised limits are preferable:** There are two reasons why we think that our proposed way of unifying different infinite-width limits is relevant, even though it excludes the sequential infinite-width limit. First, in practice, the hidden layer widths can never be set to infinity, as required by the sequential infinite width limit. In this sense, it is more meaningful in practice to consider the weak convergence we have introduced instead of the sequential limits. Second, the infinite-width limits considered by [2] and [3] are both parameterized and provide elaborate ways of dealing with the infinite-width limit. Our definition shows how they relate to each other.
>
> - We have added grid lines in all revised plots (see global response PDF) to better show the horizontal alignment of the peaks. In addition, the quasi-logarithmic scaling in Figure R2 (see global response PDF) improves the visibility of the vertical alignment of the peaks. The black line is not sufficiently visible if it is placed at the back of the plot.
>
> [1] Jacot, Arthur, Franck Gabriel, and Clément Hongler. "Neural tangent kernel: Convergence and generalization in neural networks." Advances in neural information processing systems 31 (2018).
> [2] Lee, Jaehoon, et al. "Wide neural networks of any depth evolve as linear models under gradient descent." Advances in neural information processing systems 32 (2019).
> [3] Matthews, Alexander G. de G., et al. "Gaussian process behaviour in wide deep neural networks." arXiv preprint arXiv:1804.11271 (2018).

---

> > ### Comment · Reviewer_PShm · 2024-08-07
> >
> > Thank you for the answer! This response addressed my questions, so I'm keeping the score of 8. (The only note is that $f(n_1, n_2)$ would need to have additional conditions to guarantee exchangeability of limits, but that's a small point.)

---

> > > ### Author Response · Authors · 2024-08-12
> > >
> > > Thank you for your quick response. We agree that the continuity of $f$ does not guarantee the exchangeability of limits. A small note to avoid misunderstandings: We are interested in finding a parameterization that corresponds to a particular sequential limit, not in changing the order of that sequential limit.

---

### Author Rebuttal · Authors · 2024-08-06

We would like to thank all reviewers for their thorough reviews and helpful suggestions.

We provide four additional figures in the attached PDF, where we have implemented the suggestions of Reviewer PShm and Reviewer Ljsu. The figures address the question raised by Reviewer L5ST. In Figure R1, we have added grid lines to Figure 1 to illustrate that the peaks overlap horizontally. In Figure R2, we have added grid lines and an asinh-scaling for the y-axis (approximately linear for small absolute values and logarithmic for large absolute values) to Figure 1. Moreover, we have added a plot of the singular kernel, which we obtain from the analytic NTK as $m \to \infty$. Note that this allows for a nice comparison with Figure 2, where we have plotted the non-singular kernel, which we obtain from the analytic SG-NTK as $m \to \infty$. In Figure R3 and Figure R4, we have plotted the mean squared errors between the empirical and analytic kernels for Figure 1 and Figure 2 respectively. We can see the convergence of the empirical SG-NTKs in R4, in accordance with our theoretical results.

We answer the individual reviews in the order of reviewer comments.

---

### Comment · Area_Chair_dx7n · 2024-08-08

Given the enthusiasm for this paper from the reviewers, the question
of accepting this paper as a spotlight/oral comes up. To make this
decision, it is particularly important to consider the novelty and
general applicability of this paper. It would be very helpful if the
reviewers could comment whether they believe this paper is
novel/important enough to get a spotlight/oral, and why.

---

### Decision · Program_Chairs · 2024-09-25

**Decision:**

Accept (spotlight)

**Comment:**

The paper does a thorough analysis of the NTK regime in in the presence of surrogate gradients. The technique seem general enough that they might be relevant to other models. Most reviewers are excited about the result and consider the paper to be worth of a spotlight.